# Probing Classifiers are Unreliable for Concept Removal and Detection

**Abhinav Kumar**
Microsoft Research
t-abkumar@microsoft.com

**Chenhao Tan**
University of Chicago
chenhao@uchicago.edu

**Amit Sharma**
Microsoft Research
amshar@microsoft.com

## Abstract

Neural network models trained on text data have been found to encode undesirable linguistic or sensitive concepts in their representation. Removing such concepts is non-trivial because of a complex relationship between the concept, text input, and the learnt representation. Recent work has proposed post-hoc and adversarial methods to remove such unwanted concepts from a model's representation. Through an extensive theoretical and empirical analysis, we show that these methods can be counter-productive: they are unable to remove the concepts entirely, and in some cases may fail severely by destroying all task-relevant features. The reason is the methods' reliance on a *probing* classifier as a proxy for the concept. Even under the most favorable conditions for learning a probing classifier when a concept's relevant features in representation space alone can provide 100% accuracy, we prove that a probing classifier is likely to use non-concept features and thus post-hoc or adversarial methods will fail to remove the concept correctly. These theoretical implications are confirmed by experiments on models trained on synthetic, Multi-NLI, and Twitter datasets. For sensitive applications of concept removal such as fairness, we recommend caution against using these methods and propose a spuriousness metric to gauge the quality of the final classifier.

## 1 Introduction

Neural models in text classification have been shown to learn spuriously correlated features [16, 27] or embed sensitive attributes like gender or race [8, 7, 6] in their representation layer. Classifiers that use such sensitive or spurious concepts (henceforth *concepts*) raise concerns of model unfairness and out-of-distribution generalization failure [39, 2, 15]. Removing the influence of these concepts is non-trivial because the classifiers are based on hard-to-interpret deep neural networks. Moreover, since many concepts cannot be modified at the input tokens level, removal methods that work at the representation layer have been proposed: **1)** post-hoc removal [7, 49, 14] on a pre-trained model (e.g., null space projection [32]), and **2)** adversarial removal [15, 48, 12] by jointly training the main task classifier with an (adversarial) classifier for the concept.

In this paper, we theoretically show that both these classes of methods can be counter-productive in real-world settings where the main task label is often correlated with the concept. Examples include natural language inference (*main* task) where the presence of negation (spurious *concept*) may be correlated with the "contradicts" label; or tweet sentiment classification (*main* task) where the author's gender (sensitive *concept*) maybe correlated with the sentiment label. Our key result is based on the observation that both these methods internally use an auxiliary (or probing) classifier [1, 42] that aims to predict the spurious concept based on the representation learnt by the main classifier.

We show that an auxiliary classifier cannot be a reliable signal on whether the representation includes features that are causally derived from the concept. As previous work has argued [4, 44, 40, 2], if the representation features causally derived from the concept are not predictive enough, the probing classifier for the concept can be expected to rely on correlated features to obtain a higher accuracy.

36th Conference on Neural Information Processing Systems (NeurIPS 2022).

However, we show a stronger result: this behavior holds even when there is no potential accuracy gain and the concept's features are easily learnable. Specifically, even when the concept's causally-related features alone can provide 100% accuracy and are linearly separable with respect to a binary probing task label, the probing classifier may still learn non-zero weights for the correlated main-task relevant features. Based on this result, under some simplifying assumptions, we prove that both post-hoc and adversarial training methods can fail to remove the undesired concept, remove useful task-relevant features in addition to the undesired concept, or do both. As a severe failure mode, we show that post-hoc removal methods can lead to a *random-guess* main-task classifier by removing all task-relevant information from the representation.

Empirical results on four datasets—natural language inference, sentiment analysis, tweet-mention detection, and a synthetic task—confirm our claims. Across all datasets, as the correlation between the main task and the concept increases, post-hoc removal using null space projection removes a higher amount of the main-task features, eventually leading to a random-guess classifier. In particular, for a pre-trained classifier that does not use the concept at all, the method modifies the representation to yield a classifier that either uses the concept or has lower main-task accuracy, irrespective of the correlation between the main task and the concept. Similarly, for the adversarial removal method, we find that it does not remove all concept-related features. For most datasets, the concept features left within an classifier's representation are comparable to that for a standard main-task classifier.

Our theoretical analysis complements past empirical critiques of adversarial methods for concept removal [12]. More generally, we extend the literature on probing classifiers and their unreliability [4]. Adding to known limitations of explainability methods [19, 35] based on the accuracy of a probing classifier, our results show that recent causally-inspired methods like amnesic probing [13] are also flawed because they depend on access to a good quality concept classifier. Our contributions include:

- Theoretical analysis of null space and adversarial removal methods showing that they fail to remove an undesirable concept from a model's representation, even under favorable conditions.
- Empirical results on four datasets showing that the methods are unable to remove a spurious concept's features fully and end up unnecessarily removing task-relevant features.
- A practical spuriousness score for evaluating the output of concept removal methods.

## 2  Concept removal: Background and problem statement

For a classification task, let $(\boldsymbol{x}^i, y_m^i)_{i=1}^n$ be set of examples in the dataset $\mathcal{D}_m$, where $\boldsymbol{x}^i \in \boldsymbol{X}$ are the input features and $y_m^i \in Y_m$ the label. We call this the *main task* and label $y_m^i$ the main task label. The main task classifier can be written as $c_m(h(\boldsymbol{x}))$ where $h : \boldsymbol{X} \to \boldsymbol{Z}$ is an encoder mapping the input $\boldsymbol{x}$ to a latent representation $\boldsymbol{z} := h(\boldsymbol{x})$ and $c_m : \boldsymbol{Z} \to Y_m$ is the classifier on top of the representation $\boldsymbol{Z}$. Additionally, we are given labels for a spurious or sensitive concept, $y_p \in Y_p$, i.e., $(\boldsymbol{x}^i, y_p^i)_{i=1}^{n'}$ in a dataset $\mathcal{D}_p$, and our goal is to ensure that the representation $h(\boldsymbol{x})$ learnt by the main classifier does not include features causally derived from the concept. Below we define what it means to be "causally derived": the representation should not change under an intervention on concept.

**Definition 2.1.** *(**Concept-causal feature**) A feature $Z_j \in \boldsymbol{Z}$ (jth dimension of $h(\boldsymbol{x})$) at the representation layer is defined to be causally derived from a concept (concept-causal for short) if upon changing the value of the concept, the corresponding change in the input's value $\boldsymbol{x}$ will lead to a change in the feature's ($Z_j$) value.*

For simplicity, we assume that the non-concept-causal features are the *main task* features. Often, the main task and the concept label are correlated; hence the learnt representation $h(\boldsymbol{x})$ for the main task may include concept-causal features too. A concept removal algorithm is said to be successful if it produces a *clean* representation $h'(\boldsymbol{x})$ to be used by the main-classifier that has no concept-causal features and it does not corrupt or removes the main-task features. If the representation does not contain such features, the main classifier cannot use them [12]. In practice, it is okay if the concept-causal features are not completely removed, but our key criterion is that the removal process should not remove the correlated main task features.

**Existing concept removal methods.** When the text input can be changed based on changing the value of concept label, methods like data augmentation [23, 50, 41] have been proposed for concept removal. However, for most sensitive or spurious concepts, it is not possible to know the correct change to apply at the input level corresponding to a change in the concept's value.

Instead, methods based on the representation layer have been proposed. To determine whether features in a representation are causally derived by the concept, these methods train an auxiliary, probing classifier $c_p : \boldsymbol{Z} \rightarrow Y_p$ where $y_p \in Y_p$ is the label of the concept we want to remove from the latent space $\boldsymbol{z} \in \boldsymbol{Z}$. The accuracy of the classifier indicates the predictive information about the concept embedded in the representation. This probing classifier is then used to remove the sensitive concept from the latent representation which will ensure that the main-task classifier cannot use them. Two kinds of feature removal methods have been proposed: 1) *post-hoc* methods such as null space removal [32, 13, 24, 20], with removal after the main-task classifier is trained; 2) *adversarial* methods that jointly train the main task with the probing classifier as the adversary [15, 48, 33, 34].

For adversarial removal, recent empirical results cast doubt on the method's capability to fully remove the sensitive concept from the model's representation [12]. We extend those results with a rigorous theoretical analysis and provide experiments for both adversarial and post-hoc removal methods.

# 3   Attribute removal using probing classifier can be counter-productive

As mentioned above, both removal methods internally use a probing classifier as a proxy for the concept's features. In §3.1, we start off by showing that for any classification task be it probing or main-task classification, it is difficult to learn a *clean* classifier which doesn't use any spuriously correlated feature (Lemma 3.1 and Lemma B.1). Hence the key assumption driving the use of predictive classifiers within both removal methods is incorrect. Next in §3.2 and 3.3, we will show how these individual components' failure leads to the failure of both removal methods. Finally, in §3.4, we propose a practical *spuriousness score* to assess the output classifier from any of the removal methods. Throughout this section, we assume that both the main task label $y_m$ and probing task label $y_p$ are binary ($\in \{-1, 1\}$) and there is a basic, fixed encoder $h$ converting the text input to features in the representation space (e.g., a pre-trained model like BERT [10]).

## 3.1   Fundamental limits to learning a *clean* classifier: Probing and Main Classifier

Given $\boldsymbol{z} = h(\boldsymbol{x})$ and the concept label $y_p$, the goal of the probing task is to learn a classifier $c_p(\boldsymbol{z})$ such that it only uses the concept-causal features and the accuracy for $y_p$ is maximized. We assume that the main task and concept labels are correlated, so it can be beneficial to use main-task features to maximize accuracy for $y_p$. As argued in the probing literature [19, 4], if there are features in $\boldsymbol{z}$ outside concept-causal that help improve the accuracy of the classifier, a classifier trained on standard losses such as cross-entropy or max-margin is expected to use those features too. Below we show a stronger result: even when there is no accuracy benefit of using non concept-causal features, we find that a probing classifier may still use those features.

**Creating a favorable setup for the probing classifier.** Specifically, we create a setting that is the most favorable for a probing classifier to use only concept-causal features: **1)** no accuracy gain on using features outside of concept-causal because concept-causal features are linearly separable for concept labels , and **2)** disentangled representation so that no further representation learning is required. Yet we find that a trained probing classifier would use non-concept-causal features.

**Assumption 3.1** (Disentangled Latent Representation). *The latent representation $\boldsymbol{z}$ is disentangled and is of form $[\boldsymbol{z}_m, \boldsymbol{z}_p]$, where $\boldsymbol{z}_p \in \mathbb{R}^{d_p}$ are the concept-causal features and $\boldsymbol{z}_m \in \mathbb{R}^{d_m}$ are the main task features. Here $d_m$ and $d_p$ are the dimensions of $\boldsymbol{z}_m$ and $\boldsymbol{z}_p$ respectively.*

**Assumption 3.2** (Concept-causal Feature Linear Separability). *The concept-causal features ($\boldsymbol{z}_p$) of the latent representation ($\boldsymbol{z}$) are linearly separable/fully predictive for the concept labels $y_p$, i.e., $y_p^i \cdot (\hat{\boldsymbol{\epsilon}}_p \cdot \boldsymbol{z}_p^i + b_p) > 0, \forall(\boldsymbol{x}^i, y_p^i)$ in training dataset $\mathcal{D}_p$ for some $\hat{\boldsymbol{\epsilon}}_p \in \mathbb{R}^{d_p}$ and $b_p \in \mathbb{R}$.*

**The effect of spurious correlation between concept and label.** Now we are ready to state the key lemma which will show that if there is a *spurious correlation* between the main task and concept labels such that the main-task features $\boldsymbol{z}_m$ are predictive of the concept label for only a *few special* points, then the probing classifier $c_p(\boldsymbol{z})$ will use those features. We operationalize spurious correlation as,

**Assumption 3.3** (Spurious Correlation). *For a subset of training points $\mathcal{S} \subset \mathcal{D}_p$ in the training dataset for a probing classifier, $\boldsymbol{z}_m$ is linearly-separable with respect to concept label $y_p$, i.e., $y_p^i \cdot (\hat{\boldsymbol{\epsilon}}_m \cdot \boldsymbol{z}_m^i + b_m) > 0 \; \forall i \in \mathcal{S}$ , where $\hat{\boldsymbol{\epsilon}}_m \in \mathbb{R}^{d_m}$ and $b_m \in \mathbb{R}$.*

For simplicity, we assume that the encoder $h(\cdot)$ which maps the input $\boldsymbol{X}$ to latent representation $\boldsymbol{Z}$ is frozen or non-trainable. Following [29], we assume max-margin as training loss; under some

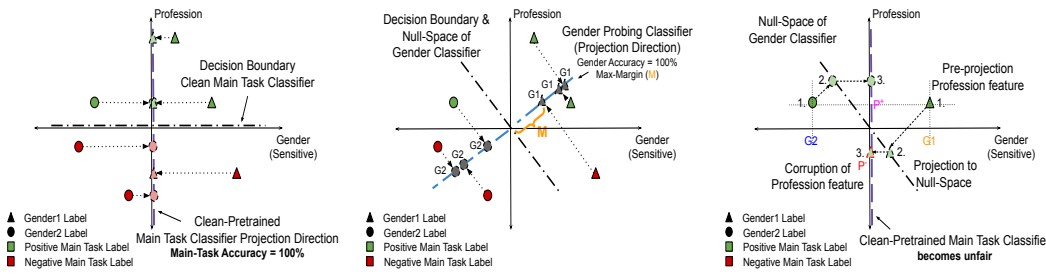

Figure 1: **Failure mode of null space removal.** Consider a main task (*Profession*) classifier where *Gender* is the spurious concept to be removed. Assume a 2-dimensional latent representation $z$, where one dimension corresponds to profession and the other to the gender feature. **(a)** A "clean" (fair) main task classifier that only uses the Profession feature, shown by its vertical projection direction, that is input to INLP for concept removal. Its decision boundary is orthogonal to the projection direction. **(b)** From Lemma 3.1, INLP trains a probing classifier for gender with a slanted projection direction (ideal gender projection direction would be horizontal). **(c)** For two points having the same profession but different gender features (marked *'1'*), projection to the null-space (*'2'*) has their profession feature reversed (*'3'*), thus making the fair pretrained classifier become unfair (also see §3.2).

mild conditions on separable data, a classifier trained using logistic/exponential loss converges to max-margin classifier given infinite training time [43, 21].

**Lemma 3.1.** *Let the latent representation be frozen and disentangled such that* $z = [z_m, z_p]$ *(Assm 3.1), and concept-causal features $z_p$ are fully predictive for the concept label $y_p$ (Assm 3.2). Let* $c_p^*(z) = w_p \cdot z_p$ *where* $w_p \in \mathbb{R}^{d_p}$ *be the desired* clean *linear classifier trained using the max-margin objective (§B.1) that only uses $z_p$ for its prediction. Let $z_m$ be the main task features, spuriously correlated s.t. $z_m$ are linearly-separable w.r.t. probing task label $y_p$ for the margin points of $c_p^*(z)$ (Assm 3.3). Then, assuming a zero-centered latent space ($b_p = 0$), a concept-probing classifier $c_p$ trained using the max-margin objective will use spurious features, i.e.,* $c_p(z) = w_p \cdot z_p + w_m \cdot z_m$ *where* $w_m \neq 0$ *and* $w_m \in \mathbb{R}^{d_m}$.

*Proof Sketch.* Starting from $c_p^*(z)$, we show that there always exists a perturbed classifier which uses the main task features and has a bigger margin than $c_p^*(z)$. Within some range of perturbation, for all margin points of $c_p^*$, using the main task features increases the margin by Assm 3.3, and does not reduce the margin for non-margin points s.t. it becomes the same as the margin of $c_p^*$. Proof in §B.3.

Our result shows that not just accuracy, even geometric skews in the dataset can yield an incorrect probing classifier. In §B.4 we prove that the assumptions for Lemma 3.1 are both sufficient and necessary for a classifier to use non-concept-features $z_m$ when $z_p$ is 1-dimensional. Lemma 3.1 generalizes a result from [29] by using fewer assumptions (we do not restrict $z_m$ to be binary, do not assume that $z_m$ and $z_p$ are conditionally independent given $y$, and do not assume monotonicity of classifier norm with dataset size). We present a similar result for the main task classifier: under spurious correlation of concept and main task labels, the main task classifier would use concept-causal features even when 100% accuracy can be achieved using only main task features (Lemma B.1, §B.2).

### 3.2 Failure mode of post-hoc removal methods: Null-space removal (INLP)

The null space method [32, 13], henceforth referred as *INLP*, removes a concept from latent space by projecting the latent space to a subspace that is not discriminative of that concept. First, it estimates the subspace in the latent space discriminative of the concept we want to remove by training a probing classifier $c_p : Z \rightarrow Y_p$, where $Y_p$ is the concept label. Then the projection is done onto the null-space of this probing classifier which is expected to be non-discriminative of the concept. For instance, [32] use a linear probing classifier $c_p(z)$ to ensure that the any linear classifier cannot recover the removed concept from modified latent representation $z'$ and hence the main task classifier ($c_m(z')$) becomes invariant to removed concept. Also, they recommend running this removal step for multiple iterations to ensure the unwanted concept is removed completely (details are in §C.1). Below we state the failure of the null-space method using $z^{i(k)}$ to denote the representation $z^i$ after $k$ steps of INLP.

**Theorem 3.2.** *Let $c_m(z)$ be a pre-trained main-task classifier where the latent representation $z = [z_m, z_p]$ satisfies Assm 3.1 and 3.2. Let $c_p(z)$ be the probing classifier used by INLP to remove the unwanted features $z_p$ from the latent representation. Under Assm 3.3, Lemma 3.1 is satisfied for the probing classifier $c_p(z)$ such that $c_p(z) = w_p \cdot z_p + w_m \cdot z_m$ and $w_m \neq 0$. Then,*

1. ***Damage in the first step of INLP.*** *The first step of linear-INLP will corrupt the main-task features and this corruption is non-invertible with subsequent projection steps of INLP.*

   (a) ***Mixing:*** *If $w_p \neq 0$, the main task $z_m$ and concept-causal features $z_p$ will get mixed such that $z^{i(1)} = [g(z_m^i, z_p^i), f(z_p^i, z_m^i)] \neq [g(z_m^i), f(z_p^i)]$ for some function "$f$" and "$g$". Thus, the latent representation is no longer disentangled and removal of concept-causal features will also lead to removal of main task features.*

   (b) ***Removal:*** *If $w_p = 0$, then the first projection step of INLP will do opposite of what is intended, i.e., damage the main task features $z_m$ (in case $z_m \in \mathbb{R}$, it will completely remove $z_m$) but have no effect on the concept-causal features $z_p$.*

2. ***Removal in the long term:*** *The L2-norm of the latent representation $z$ decreases with every projection step as long as the parameters of probing classifier ($w^k$) at a step "$k$" does not lie completely in the space spanned by parameters of previous probing classifiers, i.e., span($w^1, \ldots, w^{k-1}$), $z^{i(k-1)}$, $z^{i(0)}$ and $z^{i(0)}$ in direction of $w^k$ is not trivially zero. Thus, after sufficiently many steps, INLP can destroy all information in the representation s.t. $z^{i(\infty)} = [0, 0]$.*

*Proof Sketch.* From Lemma 3.1, in the first step, probing classifier for $z_p$ will use $z_m$ in addition to $z_p$. Consequently, the projection matrix for INLP based on the probing classifier will be incorrect, hence corrupting the main task features $z_m$ with $z_p$ **(1a)** or damage $z_m$ without any effect on $z_p$ **(1b)**. Next, we show that each step of the projection operation reduces the norm of latent representation $z$; thus the latent representation can go to **0** as the number of steps increases **(2)**. Proof in §C.

**Failure Mode:** Fig. 1a-1c demonstrate the *mixing* problem stated in Theorem 3.2, where a fair classifier becomes unfair after the first step of projection. Note that after first step the main task classifier's accuracy will drop because of this mixing of features, affecting INLP-based probing methods like Amnesic Probing [13] that interpret a drop in the main classifier's accuracy after INLP projection as evidence that the main classifier was using the sensitive concept.

### 3.3 Failure mode of adversarial removal methods

To remove the unwanted features $z_p$ from the latent representation, adversarial removal methods jointly train the main classifier $c_m : Z \rightarrow Y_m$ and the probing classifier $c_p : Z \rightarrow Y_p$ by specifying $c_p$'s loss as an adversarial loss. For details refer to §D.1.

As in Lemma 3.1, we assume that the encoder $h : X \rightarrow Z$ mapping the input to the latent representation $Z$ is frozen. To allow for the removal of the unwanted features $z_p$, we introduce additional representation layers after it. For simplicity in the proof, we assume a linear transformation to the latent representation $h_2 : Z \rightarrow \zeta$. This layer is followed by the linear main-task classifier $c_m : \zeta \rightarrow Y_m$, as before. The probing classifier $c_p : \zeta \rightarrow Y_p$ is trained adversarially to remove $z_p$ from the latent representation $\zeta$. Thus, the goal of the adversarial method can be stated as removing the information of $z_p$ from $\zeta$. Let the main-task classifier satisfy assumptions of the generalized version of Lemma 3.1 (Lemma B.1, §B.2). We also need an additional assumptions on the hard-to-classify margin points to ensure that main-task labels and concept labels are correlated on the margin points of a *clean* main-task classifier. Proof of the Theorem 3.3 stated below is in §D.

**Assumption 3.4** (Label Correlation on Margin Points)**.** *For the margin points of a* clean *classifier for the main task, the adversarial-probing labels $y_p$ and the main task labels $y_m$ are correlated, i.e., w.l.o.g., $y_m^i = y_p^i$ for all margins points of the clean main task classifier.*

**Theorem 3.3.** *Let the latent representation $z$ satisfy Assm 3.1 and be frozen, $h_2(z)$ be a linear transformation over $Z$ s.t. $h_2 : Z \rightarrow \zeta$, the main-task classifier be $c_m(\zeta) = w_{c_m} \cdot \zeta$, and the adversarial probing classifier be $c_p(\zeta) = w_{c_p} \cdot \zeta$. Let all the assumptions of Lemma B.1 be satisfied for main-classifier $c_m(\cdot)$ when using $z$ directly as input and Assm 3.2 be satisfied on $z$ w.r.t. the adversarial task. Let $h_2^*(z)$ be the desired encoder which is successful in removing $z_p$ from $\zeta$. Then there exists an undesired/incorrect encoder $h_2^\alpha(z)$ s.t. $h_2^\alpha(z)$ is dependent on $z_p$ and the main-task classifier $c_m(h_2^\alpha(z))$ has bigger margin than $c_m(h_2^*(z))$ and has,*

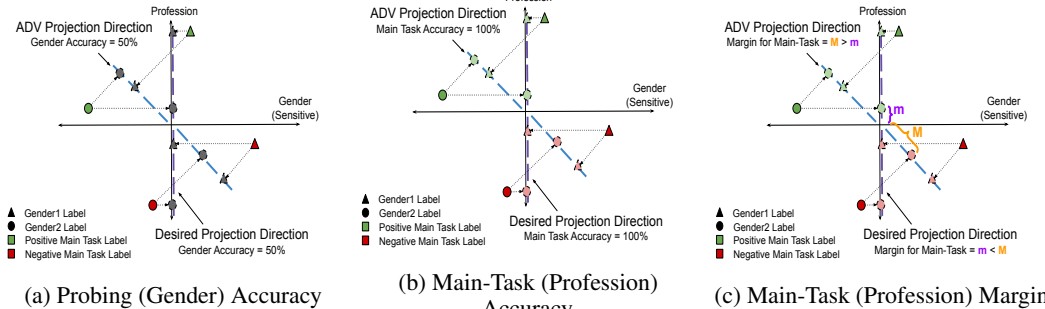

|  |  |
|---|---|
| (a) Probing (Gender) Accuracy | (b) Main-Task (Profession) Accuracy |

(c) Main-Task (Profession) Margin

Figure 2: **Failure mode of adversarial removal.** As in Fig. 1, the main task label is *Profession* and *Gender* is the spurious concept, each corresponding to one of the dimensions of the 2-dimensional feature representation $z$. Assume that the shared representation is a scalar value obtained by projecting the two features in some direction. The adversarial goal is to find a projection direction such that the concept (gender) classifier obtains a random-guess accuracy of $50\%$ but has good accuracy on the main task label (profession). **(a)** Two projection directions, shown by vertical and slanted lines, that yield *random-guess* $50\%$ accuracy on gender prediction, and **(b)** have the same $100\%$ accuracy for profession prediction. **(c)** However, the slanted projection direction has a bigger margin for the main task and will be preferred, thus leading to a final classifier that uses the gender concept (see §3.3).

1. $Accuracy(c_p(h_2^\alpha(\boldsymbol{z})), y_p) = Accuracy(c_p(h_2^*(\boldsymbol{z})), y_p)$; *when adversarial probing classifier $c_p(\cdot)$ is trained using any learning objective like max-margin or cross-entropy loss. Thus, the undesired encoder $h_2^\alpha(\boldsymbol{z})$ is indistinguishable from desired encoder $h_2^*(\boldsymbol{z})$ in terms of adversarial task prediction accuracy but better for main-task in terms of max-margin objective.*

2. $L_{h_2}\big(c_m(h_2^\alpha(\boldsymbol{z})), c_p(h_2^\alpha(\boldsymbol{z}))\big) < L_{h_2}\big(c_m(h_2^*(\boldsymbol{z})), c_p(h_2^*(\boldsymbol{z}))\big)$; *when Assm 3.4 is satisfied and concept-causal features $\boldsymbol{z}_p^M$ of any margin point $\boldsymbol{z}^M$ of $c_m(h_2^*(\boldsymbol{z}))$ are more predictive of the main task label than $\boldsymbol{z}_p^P$ of any margin point $\boldsymbol{z}^P$ of $c_p(h_2^*(\boldsymbol{z}))$ is predictive for the probing label (Assm D.1). Thus, undesired encoder $h_2^\alpha(\boldsymbol{z})$ is preferable over desired encoder $h_2^*(\boldsymbol{z})$ for both main and combined adversarial objective. Here $L_{h_2} = L(c_m(\cdot)) - L(c_p(\cdot))$ is the combined adversarial loss w.r.t. to $h_2$ and $L(c(\cdot))$ is the max-margin loss for a classifier "c" (see §D.1).*

*Proof Sketch.* **(1)** The proof is by construction. Using Lemma B.1, we show that there exists $h_2^\alpha$ s.t. $L(c_m(h_2^\alpha)) < L(c_m(h_2^*))$, and that accuracy of the probing classifier remains the same when using either encoder. **(2)** Compared to $h_2^*$, we show that the improvement in main task loss when using $\boldsymbol{z}_p$ features is larger than the improvement in the probing loss for $h_2^\alpha$, thus preferred by overall objective.

### 3.4 Implications for real-world data: A metric for quantifying degree of spuriousness

Our theoretical analysis shows that probing-based removal methods fail to make the main task classifier invariant to unwanted concepts. However, to verify whether the final classifier is using the concept or not, the theorem statements require knowledge of the concept's features $\boldsymbol{z}_p$. For practical usage, we propose a metric that quantifies the degree of failure or *spuriousness* for both the main and probing classifier. For simplicity, we define it assuming that both main and concept labels are binary.

Let $\mathcal{D}_{m,p}$ be the dataset where for every input $\boldsymbol{x}^i$ we have both the main task label $y_m$ and the concept label $y_p$. We define $2 \times 2$ groups, one for each combination of $(y_m, y_p)$. Without loss of generality, assume that the main-task label $y_m = 1$ is spuriously correlated with concept label $y_p = 1$ and similarly $y_m = 0$ is correlated with $y_p = 0$. Thus, $(y_m = 1, y_p = 1)$ and $(y_m = 0, y_p = 0)$ are the majority group $S_{maj}$ while groups $(y_m = 1, y_p = 0)$ and $(y_m = 0, y_p = 1)$ make up the minority group $S_{min}$. We expect the main classifier to exploit this correlation and hence perform badly on $S_{min}$ where the correlation breaks. Following [39], we posit that minority group accuracy i.e $Acc(S_{min})$ can be a good metric to evaluate the degree of *spuriousness*. We bound the metric by comparing it with the accuracy on $S_{min}$ of a "clean" classifier that does not use the concept features.

**Definition 3.1** (Spuriousness Score)**.** *Given a dataset, $\mathcal{D}_{m,p} = S_{min} \cup S_{maj}$ with binary task label and binary concept, let $Acc^f(S_{min})$ be the minority group accuracy of a given main task classifier ($f$) and $Acc^*(S_{min})$ be the minority group accuracy of a* clean *main task classifier that does not use the spurious concept. Then spuriousness score of $f$ is: $\psi(f) = |1 - Acc^f(S_{min})/Acc^*(S_{min})|$.*

To estimate $Acc^*(S_{min})$, we subsample the dataset such that $y_p$ takes a single value in the sample and train the main classifier on it, as in [35]. Here the probing label $y_p$ no longer is correlated with the main task label $y_m$. The spuriousness score of a *probing* classifier can be defined analogously to Def 3.1, by swapping the task and concept label (see Def E.1). For creating a clean probing classifier, we subsample the dataset such that $y_m$ takes a single value and train the probing classifier.

## 4  Experimental Results

Theorems 3.2 and 3.3 show the failure of concept removal methods under a simplified setup and max-margin loss. But current deep-learning models are not trained using max-margin objective and might not satisfy the required assumptions (Assm 3.1,3.2,3.3,3.4). Thus, we now verify the failure modes on three real-world datasets and one synthetic dataset, without making any restrictive assumptions. We use RoBERTa [25] as default encoder and fine-tune it over each real-world dataset. For Synthetic-Text dataset we use the sum of pre-trained GloVe embeddings [30] of words in a sentence as the default encoder. For details on the experimental setup, refer §E.

### 4.1  Datasets: Main task and spurious/sensitive concept

**Real-world data.** We use three datasets: MultiNLI [46], Twitter-PAN16 [31] and Twitter-AAE [6]. In MultiNLI, given two sentences—premise and hypothesis—the main task is to predict whether hypothesis *entails*, *contradicts* or is *neutral* with respect to premise. We simplify to a binary task of predicting whether a hypothesis *contradicts* the premise or not. Since negation words like *nobody,no,never* and *nothing* have been reported to be spuriously correlated with the *contradiction* label [16], we create a 'negation' concept denoting the presence of these words. The goal is to remove the negation concept from an NLI model's representation space. In Twitter-PAN16, the main task is to detect whether a tweet mentions another user or not, as in [12]. The dataset contains *gender* label for each tweet, which we consider as the sensitive concept to be removed from the main model's representation. In Twitter-AAE, again following [12], the main-task is to predict binary sentiment labels from a tweet's text. The tweets are associated with *race* of the author, the sensitive concept to be removed from the main model's representation.

**Synthetic-Text.** To understand the reasons for failure, we introduce a Synthetic-Text dataset where it is possible to change the input text based on a change in concept (thus implementing Def. 2.1). Here we can directly evaluate whether the concept is being used by the main-task classifier by intervening on the concept (adding or removing) and observing the change in model's prediction. The main-task is to predict whether a sentence contains a numbered word (e.g., *one, fifteen, etc.*). We introduce a spurious concept (length) by increasing the length of sentences that contain *numbered* words.

**Predictive correlation.** To assess robustness of removal methods, we create multiple datasets with different *predictive* correlation between the two labels $y_m$ and $y_p$. The predictive correlation ($\kappa$) is a practical measure for the *spurious correlation* defined in Assm 3.3, that does not require access to $z_m$ features. It measures how informative one label is for predicting the other, $\kappa = Pr(y_m \cdot y_p > 0) = \frac{\sum_{i=1}^{N} \mathbf{1}[y_m \cdot y_p > 0]}{N}$, where $N$ is the size of dataset and $\mathbf{1}[\cdot]$ is the indicator function that is 1 if the argument is true otherwise 0. Predictive-correlation lies in $\kappa \in [0.5, 1]$ where $\kappa = 0.5$ indicates no correlation and $\kappa = 1$ indicates that the attributes are fully correlated. For more details on the datasets and how we vary the predictive-correlation, refer to §E; and for additional results see §F.

**Measuring spuriousness of a classifier.** We use the *Spuriousness Score* (Def 3.1) to measure the degree of reliance of the main task classifier on the spurious concept, and vice-versa for the probing classifier. In addition, for the Synthetic-Text dataset, we use a metric $\Delta$Prob that exactly implements Def 2.1 for estimating a model's reliance on the spurious concept. Since we can modify the concept directly in input space for Synthetic-Text, $\Delta$Prob changes the parts of a input sentence corresponding to spurious concept and measures the change in the main task classifier's prediction probability. As a sanity check, on the Synthetic-Text dataset, $\Delta$Prob and Spuriousness Score are highly correlated (Pearson correlation=0.83, §G). For all real-world datasets, we use the Spuriousness Score.

### 4.2  Results: Null space removal

In general, for any model given as input to INLP, it may be difficult to verify whether INLP removed the correct features. Hence, we construct a benchmark where the input classifier is *clean*, i.e., it does not use the concept at all. We do so by training on a subset of data with one particular value of spurious concept label, as in [35]. Since the input classifier does not use the concept-causal features,

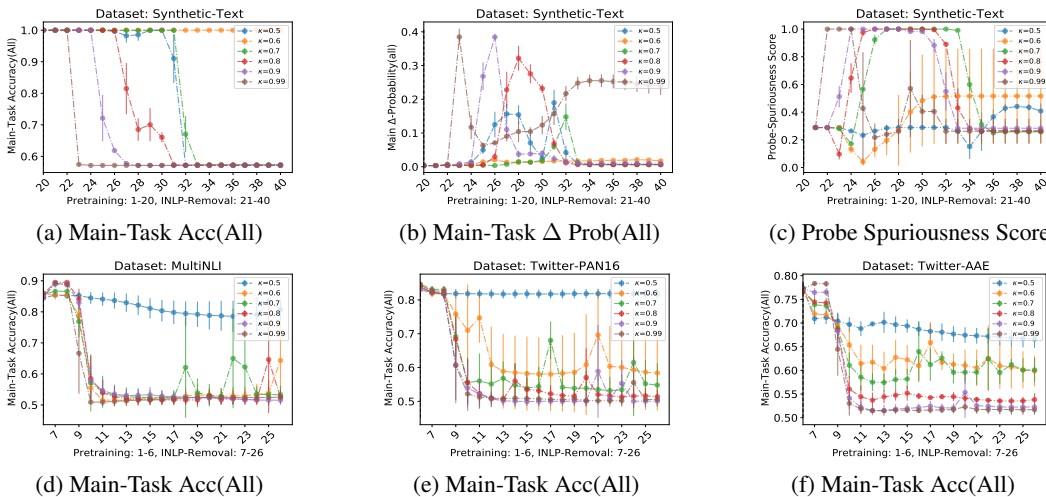

(a) Main-Task Acc(All)      (b) Main-Task $\Delta$ Prob(All)      (c) Probe Spuriousness Score

(d) Main-Task Acc(All)      (e) Main-Task Acc(All)      (f) Main-Task Acc(All)

Figure 3: **Null space removal failure.** Top row corresponds to the Synthetic-Text dataset and bottom row shows the failure on three real-world datasets. In each figure, the x-axis shows the INLP iteration and y-axis shows different evaluation metrics. Colored lines correspond to the different levels of predictive correlation ($\kappa$) in the datasets used by INLP. **(a), (d), (e), (f)** show that as INLP removal progresses, main-task classifier is getting corrupted which leads to drop in its accuracy (see §4.2).

we expect that INLP should not have any effect on the main task classifier. Note that we keep the main task classifier frozen in all the experiments described below. For the setting where the main task classifier is retrained after every projection step of INLP, refer §F.2 and Fig. 9.

**Eventually all task-relevant features are destroyed.** We start with the Synthetic-Text dataset by training a clean classifier on the main-task and inputting it to INLP for removing the spurious concept. To keep the conditions favorable for INLP, both the main task and concept-probing task can achieve 100% accuracy using their causally derived features respectively. In Fig. 3a, colored lines show datasets with different levels of predictive correlation $\kappa$ that are provided to INLP and iterations 21-40 show individual steps of null-space removal. Since, the given pre-trained classifier was *clean*, i.e., not using the concept features, null-space removal shouldn't have any effect on it. We observe that for all values of $\kappa$, the main-task classifier's accuracy eventually goes to 50% random guess accuracy implying that the main-task related attribute has been removed by INLP, as predicted by Theorem 3.2. Higher the value of correlation $\kappa$, faster the removal of main-task attribute happens. We obtain a similar pattern over real-world datasets. Fig. 3d,3e and 3f show a decrease in the main-task accuracy even when the input classifier for each dataset is ensured to be *clean*: except for $\kappa = 0.5$ (no correlation), all values of $\kappa$ yield a random-guess classifier after applying INLP on MultiNLI while they yield classifiers with less than 60% accuracy for Twitter-PAN16 and Twitter-AAE.

**Early stopping increases the reliance on spurious features.** To avoid full collapse of the main-task features, a stopping criterion in INLP is to stop when the main-task classifier's performance drops [32]. In Fig. 3b we measure spuriousness, sensitivity of the Synthetic-Text main task classifier w.r.t. to the spurious concept, using $\Delta$Prob (see §4.1 and E.8). At lower iterations of INLP, $\Delta$Prob is higher than that of the input classifier. For example, for $\kappa = 0.8$, when the main-task classifier's performance drops at iteration 27, the classifier has a high $\Delta$Prob $\approx 25\%$, higher than the input classifier (Fig. 3b). Hence it is possible that stopping prematurely will lead us to a classifier that is more reliant on the spurious concept than it was before, consistent with the statement 1(a) in Theorem 3.2. The reason is the mixing of the main task and concept-causal features in each iteration, as shown in Fig. 3c using the spuriousness score of the probing classifier (Def 3.1). At lower iterations, the spuriousness score of probing classifier increases to a very high value (close to max value 1), for all values of $\kappa$.

**Failure of causally-inspired probing.** Amnesic Probing [13] declares that a sensitive concept is being used by the model if, after removal of the concept from from the latent representation using INLP, there is a drop in the main-task performance. But Fig. 3a, 3d, 3e and 3f show that even when the input classifier for its corresponding main task is clean, i.e., does not use the sensitive

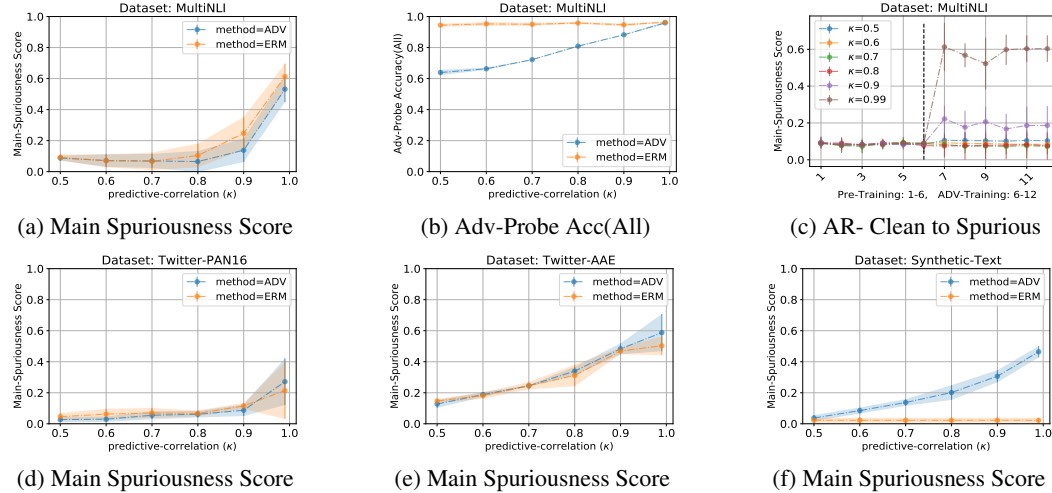

(a) Main Spuriousness Score     (b) Adv-Probe Acc(All)     (c) AR- Clean to Spurious

(d) Main Spuriousness Score     (e) Main Spuriousness Score     (f) Main Spuriousness Score

Figure 4: **Adversarial Removal Failure.** Top row explains failure of the AR method on MultiNLI. Bottom row shows the failure on Twitter-PAN16, Twitter-AAE, and Synthetic-Text datasets. In each figure, the x-axis shows different levels of predictive-correlation ($\kappa$) between the main task and concept labels in the dataset used by AR and the y-axis shows different evaluation metrics. Orange lines denote the ERM model and the blue lines denote the model trained using AR. **(b), (d), (e), (f)** show that AR is unable to completely remove the spurious concept from the main task classifier. **(c)** shows a stronger failure where the AR method introduces spuriousness into a clean input classifier.

concept, INLP leads to drop in performance of the main-classifier. Hence, removal-based methods like Amnesic probing will falsely conclude that the sensitive concept is being used.

### 4.3 Results: Adversarial removal

We now demonstrate failure of the adversarial-removal method (AR) in removing the spurious concept from the main classifier. We train a separate main-task classifier without any adversarial objective with standard cross-entropy loss objective (referred as *ERM*). Then we compare standard ERM training of the main classifier with the AR method over the same number of epochs (20). We follow the training procedure of [12] and conduct a hyper-parameter sweep on the adversarial training strength to select the value which is most effective in removing the concept. For details, refer to §E. **Cannot remove the spurious concept fully.** For MultiNLI, Fig. 4a shows the spuriousness score (Def 3.1) of ERM and AR classifiers as we vary the predictive correlation ($\kappa$) between the main-task label and sensitive concept label in the training dataset. While the spuriousness score for classifier trained using AR (blue curve) is lower than that of ERM for all values of $\kappa$, it is substantially away from zero. Thus, the AR method fails to completely remove the spurious concept completely from the latent representation. By inspecting the concept probing classifier accuracy for ERM and AR in Fig. 4b, we obtain a possible explanation. The probe accuracy after adversarial training doesn't decrease to 50% but stops at accuracy proportional to the predictive correlation $\kappa$. This is expected since even if the AR would have been successful in removing the concept-causal features, the main-task features would still be predictive of the concept label by $\kappa$ due to the spurious correlation between them. However, the converse is not true: an accuracy of $\kappa$ does not imply that the concept is fully removed. The results substantiate the first statement of Theorem 3.3: given two representations where one (*desired*) does not have concept features while the other (*undesired*) contains the concept features, the undesired one may be better for the main task accuracy even as both may have the same probing accuracy. Fig. 4d, 4e and 4f show the spuriousness score of AR in comparison to classifier trained with ERM on Twitter-PAN16, Twitter-AAE and Synthetic-Text datasets respectively. The failure of AR is worse here: there is no significant reduction in spuriousness score for AR in comparison to ERM. For the Synthetic-Text dataset, ERM has zero spuriousness score but AR has non-zero score. We expand more on this observation and include additional results on adversarial removal in §F.3.

**AR makes a clean classifier use the spurious concept.** In Fig. 4c we provide a clean main task classifier (see §F.3 for training details) as input to AR method. For all values of $\kappa$, the input classifier's spuriousness score is low (iteration 1-6). From iteration 7 onwards, the AR method corrupts the clean classifier as shown by increasing spuriousness scores. For more results, see Fig. 12b in §F.3.

**Comparison with previous work.** If post-removal the latent representation used by the main-task classifier is still predictive of the removed concept, [12] claimed it as a failure of the adversarial removal method. However, this claim may not be correct since a feature could be present in the latent space and yet not used by model [35]. Our proposed spuriousness score metric avoids this limitation.

**Ablations.** In Appendix, we report results on using BERT instead of RoBERTa as the input encoder (§F.2, F.3), the effect of using different modeling choices like loss-function, regularization, e.t.c. (§F.4), and the behavior of probing classifiers when concept is not present in latent space (§F.1).

## 5   Related work

**Concept removal methods.** When the removal of a concept can be simulated in input space (e.g., in tabular data or simpler concepts), removing a concept directly using data augmentation [23] or gradient regularization [38, 22] can work. However, concept removal is non-trivial when change in a concept cannot be propagated via change in input tokens. Combining the ideas of null space and adversarial removal [32, 24, 48, 12], methods like [33, 34] restrict the adversarial function to be a projection operation and derive a closed form solution. Other approaches use explanations of the classifier's prediction for concept removal [17]. Our work highlights the difficulty of building an estimator for the features causally derived from a concept, as a general limitation for concept removal.

**Limitations of a probing classifier for model interpretability.** We also contribute to the growing literature on the limitations of a probing classifier's accuracy in capturing whether the main classifier is using a concept [4]. It is known that probing classifiers capture not just the concept but any other features that may be correlated with it [39, 2, 23, 45]. As a result, many improvements have been proposed to better estimate whether a concept is being used, including the use of control labels or datasets [19, 35]. Parallelly, new causality-inspired probing methods [13] compare the main task accuracy on a representation without the concept constructed using the null space removal method. The hope is that such improvements can make probing more robust. Our results question this direction. To demonstrate the fundamental unreliability of probing classifier, we construct a setup that is most favorable for learning only the concept's features and still find that learned probing classifier includes non-zero weight for other features, limiting effectiveness of any interpretation method based on it.

## 6   Conclusion

Our theoretical and empirical evaluations show that it is difficult to create a probing-based explainability and removal method due to the fundamental limitation of learning a "clean" probing classifier. We recommend two tests for validating removal methods. First, we provide a sanity-check: any reasonable removal method should not modify a "clean" classifier that does not use any spurious features to produce a final classifier that uses those features. Second, we propose a spuriousness score that can be used to evaluate the dependence of any classifier on spurious features. As a future step, we encourage the community to develop more such sanity-check tests to evaluate proposed methods.

Alternatively, we point attention to other approaches that may provide better guarantees for concept removal. An example is extending data augmentation techniques like counterfactual data augmentation ([23, 9]) to non-trivial concepts. For a given training point that may include a spurious correlation, a new data point is generated that breaks the correlation but keeps the semantics of the rest of the text identical (hence the name, "counterfactual"). This can be done by human labeling or handcrafted rules for modifying text (e.g., Checklists [37]). Then the main classifier is regularized to have the same representation for such pairs of inputs ([3, 26]). By construction, with good quality pairs, such a method will not remove task-relevant features and will satisfy the sanity checks listed above. That said, a limitation is that the removal quality will depend on the diversity of the counterfactual examples generated and whether they capture all aspects of the spurious concept. Another direction is to take inspiration from the algorithmic fairness literature [18, 28] and focus on the predictions of the classifier rather than the representation. Compared to removal in latent space, enforcing certain fairness properties on model predictions is a more well-formed task, more interpretable, and definitely more relevant if the final goal is fair decision making.

**Limitations.**   A limitation of our theoretical work is assuming frozen or non-trainable latent representation which makes the analysis of task-classifier trained on top of them relatively easier. We address this limitation in our empirical work where we do not make such assumptions. Also, our work addresses failure modes of two popular methods, null space removal and adversarial removal. We conjecture that any other method based on probing classifiers will lead to similar failure modes.

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
