## A    Broader Impact and Ethical Consideration

Removal of spurious or sensitive concepts is an important problem to ensure that machine learning classifiers generalize better to new data and are fair towards all groups. We found multiple limitations with current removal methods and recommend caution against the use of these methods in practice.

## B    Probing and Main Classifier Failure Proofs

### B.1    Notation and Setup: Max-margin Classifier

We assume that encoder $h : \boldsymbol{X} \to \boldsymbol{Z}$, mapping the input to latent representation is frozen/non-trainable. Thus for every input $\boldsymbol{x}^i$ in the dataset $\mathcal{D}$, we have a corresponding latent representation $\boldsymbol{z}^i$ which is fixed. Also, the latent representation $\boldsymbol{Z}$ is disentangled i.e $\boldsymbol{z} = [\boldsymbol{z}_m, \boldsymbol{z}_p]$ where $\boldsymbol{z}_m$ are the main task features, i.e., causally derived from the main task label and $\boldsymbol{z}_p$ are the concept-causal features, causally derived from the concept label. Let $c_p(\boldsymbol{z}) = \boldsymbol{w}_p \cdot \boldsymbol{z}_p + \boldsymbol{w}_m \cdot \boldsymbol{z}_m$ be the linear probing classifier which we train using max-margin objective. The hyperplane $c_p(\boldsymbol{z}) = 0$ is the decision boundary of this linear classifier. The points which fall on one side of the decision boundary ($c_p(\boldsymbol{z}) > 0$) are assigned one label (say positive label 1) and the rest are assigned another label (say negative label -1). The *margin* $\mathcal{M}_{c_p}$ of this probing classifier ($c_p(\boldsymbol{z})$) is the distance of the closest latent representation ($\boldsymbol{z}$) from the decision boundary. The points which are closest to the decision boundary are called the *margin points*. The distance of a given latent representation $\boldsymbol{z}^i$ having class label $y^i$, where $y^i \in \{-1, 1\}$, from the decision boundary is given by

$$\mathcal{M}_{c_p}(\boldsymbol{z}^i) := \frac{m_{c_p}(\boldsymbol{z}^i)}{\|\boldsymbol{w}\|} = \frac{y_p^i \cdot c_p(\boldsymbol{z}^i)}{\|\boldsymbol{w}\|} = \frac{y_p^i \cdot (\boldsymbol{w} \cdot \boldsymbol{z}^i + b)}{\|\boldsymbol{w}\|} \tag{1}$$

where $\|\boldsymbol{w}\|$ is the L2 norm of parameters $\boldsymbol{w} = [\boldsymbol{w}_p, \boldsymbol{w}_m]$ of the probing classifier $c_p(\boldsymbol{z})$.

**Max-Margin (MM):**    Then the max-margin classifier is trained by optimizing the following objective:

$$\underset{\boldsymbol{w}, b}{\operatorname{argmax}} \left\{ \min_i \mathcal{M}_{c_p}(\boldsymbol{z}^i) \right\} \tag{2}$$

For ease of exposition we convert this objective into multiple equivalent forms. To do this we observe that scaling the parameters of $c_p(\boldsymbol{z})$ by a positive scalar $\gamma$ i.e $\boldsymbol{w} \to \gamma \boldsymbol{w}$ and $b \to \gamma b$ does not change the distance of the point ($\mathcal{M}_{c_p}(\boldsymbol{z}^i)$) from the decision boundary.

**MM-Denominator Version:**    We can use this freedom of scaling the parameters to set $m_{c_p}(\boldsymbol{z}^i) = 1$ for the closest point of any given probing classifier $c_p(\boldsymbol{z})$ , thus all the data points will satisfy the constraint,

$$m_{c_p}(\boldsymbol{z}^i) = y^i \cdot c_p(\boldsymbol{z}^i) \geq 1 \tag{3}$$

giving us the final max-margin objective:

$$\underset{\boldsymbol{w}}{\operatorname{argmax}} \left\{ \frac{1}{\|\boldsymbol{w}\|} \right\} \tag{4}$$

under the constraint $m_{c_p}(\boldsymbol{z}^i) \geq 1$ corresponding to all the points in the dataset.

**MM-Numerator Version:**    Alternatively, one can choose $\gamma$ such that $\|\boldsymbol{w}\| = c$ where $c \in \mathbb{R}$ is some constant value. The the modified objective becomes:

$$\underset{\boldsymbol{w}, b}{\operatorname{argmax}} \left\{ m_{c_p}(\boldsymbol{z}^i) \right\} \tag{5}$$

under constraint $\|\boldsymbol{w}\| = c$ which is usually set to 1.

We will use one of these formulations in our proofs based on the ease of exposition and give a clear indication when we do so. One can refer to Chapter 7, Section 7.1 of [5] for further details about max-margin classifiers and different formulations of the max-margin objective.

## B.2 Problem with learning a *clean* main-task classifier

In this section, we will restate the assumptions and results of Lemma 3.1 for the main-task classifier (instead of the probing classifier) and show that the same results will hold.

Assm 3.1 remains the same since it is made on the latent-representation being disentangled and frozen/non-trainable. Next, parallel to Assm 3.2, we show that even when main-task feature is 100% predictive of main-task and is linearly separable, the trained main-task classifier will also use the concept-causal features. Formally,

**Assumption B.1** (main-task feature Linear Separability). *The main-task features ($z_m$) of the latent representation ($z$) for every point are linearly separable/fully predictive for the main-task labels $y_m$, i.e $y_m^i \cdot (\hat{\epsilon}_m \cdot z_m^i + b_m) > 0$ for all datapoints $(x^i, y_m^i)$ for some $\hat{\epsilon}_m \in \mathbb{R}^{d_m}$ and $b_m \in \mathbb{R}$. For the case of zero-centered latent space, we have $b_m = 0$.*

Next similar to Assm 3.3, we define the spurious correlation between main-task and concept label: a function using only $z_p$ may also be able to classify correctly on some non-empty subset of points w.r.t. main-task label ($y_m$).

**Assumption B.2** (Main-Task Spurious Correlation). *For a subset of training points $\mathcal{S} \subset \mathcal{D}_m$, main-task label $y_m$ is linearly separable using $z_p$ i.e $y_m^i \cdot (\hat{\epsilon}_p \cdot z_p^i + b_p) > 0$ for some $\hat{\epsilon}_p \in \mathbb{R}^{d_p}$ and $b_p \in \mathbb{R}$. For the case of zero-centered latent space we have $b_p = 0$.*

Next we rephrase Lemma 3.1 which shows that for only a *few special* points if the concept-causal features $z_p$ are linearly-separable w.r.t. to main task classifier $y_m$ (Assm B.2), then the main-task classifier $c_m(z)$ will use those features.

**Lemma B.1** (Sufficient Condition for Main-task Classifier). *Let the latent representation be frozen and disentangled such that $z = [z_m, z_p]$ (Assm 3.1), where main-task-features $z_m$ be fully predictive (Assm B.1). Let $c_m^*(z) = w_m \cdot z_m$ be the desired/clean linear main-task classifier trained using max-margin objective (§B.1) which only uses $z_m$ for its prediction. Let $z_p$ be the spurious feature s.t. for the margin points of $c_m^*(z)$, $z_p$ be linearly-separable w.r.t. task label $y_m$ (Assm B.2). Then, assuming the latent space is centered around $\mathbf{0}$ (i.e. $b_m = 0$ and $b_p = 0$), the main-task classifier trained using max-margin objective will be of form $c_m(z) = w_m \cdot z_m + w_p \cdot z_p$ where $w_p \neq \mathbf{0}$.*

The proof of Lemma B.1 is identical to Lemma 3.1 and is provided in §B.3.

## B.3 Proof of Sufficient Condition: Lemma 3.1 and Lemma B.1

**Lemma 3.1.** *Let the latent representation be frozen and disentangled such that $z = [z_m, z_p]$ (Assm 3.1), and concept-causal features $z_p$ are fully predictive for the concept label $y_p$ (Assm 3.2). Let $c_p^*(z) = w_p \cdot z_p$ where $w_p \in \mathbb{R}^{d_p}$ be the desired* clean *linear classifier trained using the max-margin objective (§B.1) that only uses $z_p$ for its prediction. Let $z_m$ be the main task features, spuriously correlated s.t. $z_m$ are linearly-separable w.r.t. probing task label $y_p$ for the margin points of $c_p^*(z)$ (Assm 3.3). Then, assuming a zero-centered latent space ($b_p = 0$), a concept-probing classifier $c_p$ trained using the max-margin objective will use spurious features, i.e., $c_p(z) = w_p \cdot z_p + w_m \cdot z_m$ where $w_m \neq \mathbf{0}$ and $w_m \in \mathbb{R}^{d_m}$.*

In this section we prove that, given the assumption in Lemma 3.1 is satisfied, they are sufficient for a probing classifier $c_p(z)$ to use the spuriously correlated main-task feature $z_m$. See §B.1 for detailed setup and max-margin training objective. Also, we could use the same line of reasoning to prove a similar result for the main-task classifier i.e. when conditions in Lemma B.1 are satisfied, the main-task classifier will use the spuriously correlated concept-causal feature $z_p$. To keep the proof general for both the lemmas, we prove the result for a general classifier $c(z)$ trained to predict a task label $y$. Here the latent representation $z$ be of form $z = [z_{inv}, z_{sp}]$ where $z_{inv}$ are the features which are causally-derived from the task concept ("invariant" features) and $z_{sp}$ be the features spuriously correlated to the task label $y$. With respect to probing classifier $c_p(z)$ in Lemma 3.1 $z_{inv} := z_p$ and $z_{sp} := z_m$. Similarly, for the main-task classifier in Lemma B.1, $z_{inv} := z_m$ and $z_{sp} := z_p$. For ease of exposition, we define two categories of classifiers based on which features they use:

**Definition B.1** (Purely-Invariant Classifier). *A linear classifier of form $c(z) = w_{inv} \cdot z_{inv} + w_{sp} \cdot z_{sp} + b$ is called "purely-invariant" if it does not use the spurious features $z_{sp}$ i.e., $w_{sp} = \mathbf{0}$.*

**Definition B.2** (Spurious-Using Classifier). *A linear classifier of form $c(z) = w_{inv} \cdot z_{inv} + w_{sp} \cdot z_{sp} + b$ is called "spurious-using" if it uses the spurious features $z_{sp}$ i.e. $w_{sp} \neq \mathbf{0}$.*

*Proof of Lemma 3.1 and B.1.* Let $c_{inv}(\boldsymbol{z}) = \boldsymbol{w}_{inv} \cdot \boldsymbol{z}_{inv}$ be the *clean*/purely-invariant classifier trained using the max-margin objective using the *MM-Denominator* formulation given in Eq. 4 such that $\boldsymbol{w}_{inv} \neq \boldsymbol{0}$. The classifier $c_{inv}(\boldsymbol{z})$ is 100% predictive of the task labels $y$ (from Assm 3.2 for the probing task or Assm B.1 for the main-task). Here the bias term $b = 0$ since we assume the latent representation $\boldsymbol{z}$ is zero-centered. The norm of this classifier ($c_{inv}(\boldsymbol{z})$) is $\|\boldsymbol{w}_{inv}\|$ and the distance of each input latent representation ($\boldsymbol{z}^i$) with class label $y^i$ ($y^i \in \{-1, 1\}$) from the decision boundary ($c_{inv}(\boldsymbol{z}) = 0$) is given by Eq. 1 i.e.:

$$\mathcal{M}_{inv}(\boldsymbol{z}^i) = \frac{m_{inv}(\boldsymbol{z}^i)}{\|\boldsymbol{w}_{inv}\|} = \frac{y^i \cdot c_{inv}(\boldsymbol{z}^i)}{\|\boldsymbol{w}_{inv}\|} = \frac{y^i \cdot (\boldsymbol{w}_{inv} \cdot \boldsymbol{z}_{inv}^i)}{\|\boldsymbol{w}_{inv}\|} \tag{6}$$

Since we have used the *MM-Denominator* version of max-margin to train $c_{inv}(\boldsymbol{z})$, from Eq. 3 we have $m_{inv}(\boldsymbol{z}^i) = 1$ for the *margin-points* and greater than 1 for rest of the points. Next we will construct a new classifier parameterized by $\alpha \in [0, 1]$ by perturbing the clean/purely-invariant classifier $c_{inv}(\boldsymbol{z})$ such that:

$$c_{\alpha}(\boldsymbol{z}) = \alpha\big(\boldsymbol{w}_{inv} \cdot \boldsymbol{z}_{inv}\big) + \|\boldsymbol{w}_{inv}\|\sqrt{1 - \alpha^2}\big(\hat{\boldsymbol{\epsilon}}_{sp} \cdot \boldsymbol{z}_{sp}\big) \tag{7}$$

where $\hat{\boldsymbol{\epsilon}}_{sp} \in \mathbb{R}^{d_{sp}}$ is a unit vector in spurious subspace of features, $d_{sp}$ is the dimension of the spurious feature subspace ($\boldsymbol{z}_{sp}$). We observe that the norm of this perturbed classifier $c_{\alpha}(\boldsymbol{z})$ is also $\|\boldsymbol{w}_{inv}\|$, which is same as the clean/purely-invariant classifier $c_{inv}(\boldsymbol{z})$. Thus from Eq. 1, the distance of any input $\boldsymbol{z}^i$ with class label $y^i$ from the decision boundary of this perturbed classifier $c_{\alpha}(\boldsymbol{z})$ is given by:

$$\mathcal{M}_{\alpha}(\boldsymbol{z}^i) = \frac{m_{\alpha}(\boldsymbol{z}_i)}{\|\boldsymbol{w}_{inv}\|} = \frac{y^i \cdot c_{\alpha}(\boldsymbol{z}^i)}{\|\boldsymbol{w}_{inv}\|} \tag{8}$$

The perturbed classifier will be *spurious-using* i.e use the spurious feature $\boldsymbol{z}_{sp}$ when $\alpha \in [0, 1)$ since $\boldsymbol{w}_{sp} = (\|\boldsymbol{w}_{inv}\|\sqrt{1 - \alpha^2}) \neq 0$ for these setting of $\alpha$. Thus to show that there exist a *spurious-using* classifier which has a margin greater than the margin of the *purely-invariant* classifier, we need to prove that there exist an $\alpha \in [0, 1)$ such that $c_{\alpha}(\boldsymbol{z})$ has bigger margin than $c_{inv}(\boldsymbol{z})$ i.e. $\min_{\boldsymbol{z}} \mathcal{M}_{\alpha}(\boldsymbol{z}) > \min_{\boldsymbol{z}} \mathcal{M}_{inv}(\boldsymbol{z})$. Since norm of parameters of both the classifier is same, substituting the expression of $\mathcal{M}_{\alpha}$ and $\mathcal{M}_{inv}$ from Eq. 6 and 8, we need to show $m_{\alpha}(\boldsymbol{z}^i) > 1$ for all $\boldsymbol{z}^i$. We have:

$$m_{\alpha}(\boldsymbol{z}^i) = y^i \cdot \left( \alpha\big(\boldsymbol{w}_{inv} \cdot \boldsymbol{z}_{inv}^i\big) + \|\boldsymbol{w}_{inv}\|\sqrt{1 - \alpha^2}\big(\hat{\boldsymbol{\epsilon}}_{sp} \cdot \boldsymbol{z}_{sp}^i\big) \right) \tag{9}$$

$$= \alpha \cdot m_{inv}(\boldsymbol{z}^i) + y^i\|\boldsymbol{w}_{inv}\|\sqrt{1 - \alpha^2}\big(\hat{\boldsymbol{\epsilon}}_{sp} \cdot \boldsymbol{z}_{sp}^i\big) \tag{10}$$

Let $\mathcal{S}_y^m$ denote the set of *margin-points* of purely-invariant classifier $c_{inv}(\boldsymbol{z})$ with class label $y$ having $m_{inv}(\boldsymbol{z}) = 1$ and $\mathcal{S}_y^r$ contain rest of points (non-margin points) having $m_{inv}(\boldsymbol{z}) > 1$ with the class label $y$. Here "m" stands for margin-point in superscript of $\mathcal{S}$ and "r" stands for rest of point with label $y$. In rest of the proof, first we will show that for margin-points $\boldsymbol{z}^m \in (\mathcal{S}_{y=1}^m \cup \mathcal{S}_{y=-1}^m)$, we need the assumption that spurious feature ($\boldsymbol{z}_{sp}^m$) be linearly separable with respect to class label $y$ (Assm 3.3 for probing task or B.2 for main-task) for having $m_{\alpha}(\boldsymbol{z}^i) > 1$. But for all non-margin points $\boldsymbol{z}^r \in (\mathcal{S}_{y=1}^r \cup \mathcal{S}_{y=-1}^r)$, we can always choose $\alpha \in [0, 1)$ such that $m_{\alpha}(\boldsymbol{z}^i) > 1$. Below we handle margin and non-margin of points separately.

**Case 1 : Margin Points** $(\mathcal{S}_{y=1}^m \cup \mathcal{S}_{y=-1}^m)$: For the margin-points in latent space, $\boldsymbol{z}^m \in \mathcal{S}_y^m$ we have $m_{inv}(\boldsymbol{z}^m) = 1$ and we need to show that there exists $\alpha \in [0, 1)$ such that $m_{\alpha}(\boldsymbol{z}^m) > 1$ for all $\boldsymbol{z}^m \in \mathcal{S}_y^m$. From Eq. 10 we have:

$$m_{\alpha}(\boldsymbol{z}^m) = \alpha \cdot 1 + y\|\boldsymbol{w}_{inv}\|\sqrt{1 - \alpha^2}\big(\hat{\boldsymbol{\epsilon}}_{sp} \cdot \boldsymbol{z}_{sp}^m\big) > 1 \tag{11}$$

$$\big(\|\boldsymbol{w}_{inv}\|\sqrt{1 - \alpha^2}\big)y\big(\hat{\boldsymbol{\epsilon}}_{sp} \cdot \boldsymbol{z}_{sp}^m\big) > 1 - \alpha \tag{12}$$

From Assm 3.3 for probing task or B.2 for the main-task, we know that spurious-feature $\boldsymbol{z}_{sp}^m$ of margin-points $\boldsymbol{z}^m$ are linearly-separable w.r.t to task label $y$. Since $\hat{\boldsymbol{\epsilon}}_{sp} \in \mathbb{R}^{d_{sp}}$ used in the perturbed classifier $c_{\alpha}(\boldsymbol{z})$ is arbitrary, let's set it to be an unit vector such that $y\big(\hat{\boldsymbol{\epsilon}}_{sp} \cdot \boldsymbol{z}_{sp}^m\big) > 0$ for all $\boldsymbol{z}^m \in \mathcal{S}_y^m$ (guaranteed by Assm 3.3 or B.2). Also since $\alpha \in [0, 1)$ and $\|\boldsymbol{w}_{inv}\| > 0$, we have $\big(\|\boldsymbol{w}_{inv}\|\sqrt{1 - \alpha^2}\big) > 0$. Hence the left hand side of Eq. 12 is $> 0$ for such $\hat{\boldsymbol{\epsilon}}_{sp}$. If Assm 3.3 or

B.2 (corresponding to the task) wouldn't have been satisfied then the above equation might have been inconsistent since right hand side is always $> 0$; since we need to find a solution to Eq. 12 when $\alpha \in [0, 1)$ thus $(1 - \alpha) > 0$, but left hand side wouldn't have been always greater than 0. This shows the motivation why we need Assm 3.3 or B.2 for proving this lemma. Continuing, let $\beta := (y(\hat{\boldsymbol{\epsilon}}_{sp} \cdot \boldsymbol{z}_{sp}^m))$, then squaring both sides and cancelling $(1 - \alpha)$ since we need to find a solution to Eq. 12 when $\alpha \in [0, 1) \implies (1 - \alpha) > 0$, we get:

$$\|\boldsymbol{w}_{inv}\|^2 \cancel{(1 - \alpha)}(1 + \alpha)\left( y(\hat{\boldsymbol{\epsilon}}_{sp} \cdot \boldsymbol{z}_{sp}^m)\right)^2 > \cancel{(1 - \alpha)}(1 - \alpha) \tag{13}$$

$$\|\boldsymbol{w}_{inv}\|^2(1 + \alpha)\beta^2 > (1 - \alpha) \tag{14}$$

$$\|\boldsymbol{w}_{inv}\|^2\beta^2 + \alpha\|\boldsymbol{w}_{inv}\|^2\beta^2 > 1 - \alpha \tag{15}$$

$$\alpha\left( 1 + \|\boldsymbol{w}_{inv}\|^2\beta^2 \right) > \left( 1 - \|\boldsymbol{w}_{inv}\|^2\beta^2 \right) \tag{16}$$

After substituting back the value of $\beta$ and rearranging we get:

$$\alpha > \frac{1 - \|\boldsymbol{w}_{inv}\|^2 \cdot \left( y(\hat{\boldsymbol{\epsilon}}_{sp} \cdot \boldsymbol{z}_{sp}^m)\right)^2}{1 + \|\boldsymbol{w}_{inv}\|^2 \cdot \left( y(\hat{\boldsymbol{\epsilon}}_{sp} \cdot \boldsymbol{z}_{sp}^m)\right)^2} := \alpha_y^{lb_1}(\boldsymbol{z}^m) \tag{17}$$

Lets define $\alpha_y^{lb_1} := \max_{\boldsymbol{z}^m \in \mathcal{S}_y^m}(\alpha_y^{lb_1}(\boldsymbol{z}^m))$. Since $\|\boldsymbol{w}_{inv}\|^2 \cdot \left( y(\hat{\boldsymbol{\epsilon}}_{sp} \cdot \boldsymbol{z}_{sp}^m)\right)^2 > 0$, the right hand side of above equation $\alpha_y^{lb_1}(\boldsymbol{z}^m) < 1$ for all $\boldsymbol{z}^m \in \mathcal{S}_y^m \implies \alpha_y^{lb_1} < 1$, which sets a new lower bound on allowed value of $\alpha$ for which Eq. 12 is satisfied. Thus when $\alpha \in (\alpha_y^{lb_1}, 1)$, $m_p(\boldsymbol{z}^m) > 1$ for all $\boldsymbol{z}^m \in \mathcal{S}_y^m$. That is, the perturbed probing classifier $c_\alpha(\boldsymbol{z})$ has larger margin than purely-invariant/clean classifier $c_{inv}(\boldsymbol{z})$ for the margin points $\boldsymbol{z}^m \in \mathcal{S}_y^m$.

**Case 2: Non-Margin Points** $(\mathcal{S}_{y=1}^r \cup \mathcal{S}_{y=-1}^r)$**:** For the non-margin points $\boldsymbol{z}^r \in \mathcal{S}_y^r$ in the latent space we have $m_{inv}(\boldsymbol{z}^r) > 1$. Let $\gamma := \min_{\boldsymbol{z}^r \in \mathcal{S}_y^r}\left( m_{inv}(\boldsymbol{z}^r)\right)$ thus we also have $\gamma > 1$. Let $\alpha \neq 0$ and we choose $\alpha$ such that:

$$\frac{1}{\alpha} < \gamma \tag{18}$$

$$\alpha > \frac{1}{\gamma} \tag{19}$$

Substituting the value of $\gamma$ we get:

$$\alpha > \frac{1}{\min_{\boldsymbol{z}^r \in \mathcal{S}_y^r}\left( m_{inv}(\boldsymbol{z}^r)\right)} = \alpha_y^{lb_2} \tag{20}$$

Since $\gamma > 1$, thus right hand side in above equation $\alpha_y^{lb_2} < 1$, which sets a new lower bound on allowed values of $\alpha$. Since $m_{inv}(\boldsymbol{z}^r) \geq \gamma > \frac{1}{\alpha}$ for all $\boldsymbol{z}^r \in \mathcal{S}_y^r$ for $\alpha \in (\alpha_y^{lb_2}, 1)$ (Eq. 20), we can write $m_{inv}(\boldsymbol{z}^r) = \frac{1}{\alpha} + \eta(\boldsymbol{z}^r)$ where $\eta(\boldsymbol{z}^r) := \left( m_{inv}(\boldsymbol{z}^r) - \frac{1}{\alpha}\right) > 0$ for all $\boldsymbol{z}^r \in \mathcal{S}_y^r$. Now we need to show that there exist an $\alpha \in (\alpha_y^{lb_2}, 1)$ such that $m_\alpha(\boldsymbol{z}^r) > 1$ for all $\boldsymbol{z}^r \in \mathcal{S}_y^r$. Thus from Eq. 10 we need:

$$m_\alpha(\boldsymbol{z}^r) = \alpha \cdot m_{inv}(\boldsymbol{z}^r) + \|\boldsymbol{w}_{inv}\|\sqrt{1 - \alpha^2}\left( y(\hat{\boldsymbol{\epsilon}}_{sp} \cdot \boldsymbol{z}_{sp}^r)\right) > 1 \tag{21}$$

$$\alpha \cdot (\frac{1}{\alpha} + \eta(\boldsymbol{z}^r)) + \|\boldsymbol{w}_{inv}\|\sqrt{1 - \alpha^2}\left( y(\hat{\boldsymbol{\epsilon}}_{sp} \cdot \boldsymbol{z}_{sp}^r)\right) > 1 \tag{22}$$

$$\|\boldsymbol{w}_{inv}\|\sqrt{1 - \alpha^2}\left( y(\hat{\boldsymbol{\epsilon}}_{sp} \cdot \boldsymbol{z}_{sp}^r)\right) > -\left( \alpha \cdot \eta(\boldsymbol{z}^r)\right) \tag{23}$$

Since $\alpha \in (\alpha_{lb}^2, 1)$, we have $(\alpha \cdot \eta(\boldsymbol{z}^r)) > 0$ and $\|\boldsymbol{w}_{inv}\|\sqrt{1 - \alpha^2} > 0$. Let's define $\delta(\boldsymbol{z}^r) := y(\hat{\boldsymbol{\epsilon}}_{sp} \cdot \boldsymbol{z}_{sp}^r)$. Thus for the latent-points $\boldsymbol{z}^r \in \mathcal{S}_y^r$ which have $\delta(\boldsymbol{z}^r) \geq 0$, Eq. 23 is always satisfied

since left side of inequality is greater than or equal to zero and right side is always less than zero. For the points for which $\delta(\boldsymbol{z}^r) < 0$ we have:

$$\|\boldsymbol{w}_{inv}\|\sqrt{1-\alpha^2} \cdot (-1) \cdot |\delta(\boldsymbol{z}^r)| > -(\alpha \cdot \eta(\boldsymbol{z}^r)) \tag{24}$$

$$\|\boldsymbol{w}_{inv}\|\sqrt{1-\alpha^2}|\delta(\boldsymbol{z}^r)| < (\alpha \cdot \eta(\boldsymbol{z}^r)) \tag{25}$$

$$\|\boldsymbol{w}_{inv}\|^2(1-\alpha^2)\delta(\boldsymbol{z}^r)^2 < (\alpha \cdot \eta(\boldsymbol{z}^r))^2 \tag{26}$$

$$\|\boldsymbol{w}_{inv}\|^2\delta(\boldsymbol{z}^r)^2 < \alpha^2 \cdot \left( \eta(\boldsymbol{z}^r)^2 + \|\boldsymbol{w}_{inv}\|^2\delta(\boldsymbol{z}^r)^2 \right) \tag{27}$$

$$\alpha > \sqrt{\frac{\|\boldsymbol{w}_{inv}\|^2\delta(\boldsymbol{z}^r)^2}{\eta(\boldsymbol{z}^r)^2 + \|\boldsymbol{w}_{inv}\|^2\delta(\boldsymbol{z}^r)^2}} = \alpha_y^{lb_3}(\boldsymbol{z}^r) \tag{28}$$

Now different $\boldsymbol{z}^r$ will have different $\eta(\boldsymbol{z}^r)$ which will give different lower bound of $\alpha$. Since the $m_\alpha(\boldsymbol{z}^r) > 1$ has to be satisfied for every point in $\boldsymbol{z}^r \in \mathcal{S}_y^r$ we will choose the maximum value of $\alpha_y^{lb_3}(\boldsymbol{z}^r)$ which will give tightest lower bound on value of $\alpha$. Lets define $\alpha_y^{lb_3} := \max_{\boldsymbol{z}^r \in \mathcal{S}_y^r}(\boldsymbol{z}^r)$, then for $m_\alpha(\boldsymbol{z}^r) > 1$, we need $\alpha > \alpha_y^{lb_3}$. Also, since for all $\boldsymbol{z}^r \in \mathcal{S}_y^r$, $\eta(\boldsymbol{z}^r) > 0$ we have $\alpha_y^{lb_3}(\boldsymbol{z}^r) < 1 \implies \alpha_y^{lb_3} < 1$.

Finally, combining all the lower bound of $\alpha$ from Eq. 17, Eq. 20 and Eq. 28 let the overall lower bound of $\alpha$ be $\alpha_{lb}$ given by:

$$\alpha_{lb} = \max\{\alpha_{y=1}^{lb_1}, \alpha_{y=-1}^{lb_1}, \alpha_{y=1}^{lb_2}, \alpha_{y=-1}^{lb_2}, \alpha_{y=1}^{lb_3}, \alpha_{y=-1}^{lb_3},\} \tag{29}$$

This provides a way to construct a spurious-using classifier: given any *purely-invariant* classifier, we can always choose $\alpha \in (\alpha_{lb}, 1)$ and construct a perturbed *spurious-using* classifier from Eq. 7 which has a bigger margin than *purely-invariant*. Thus, given all the assumptions, there always exists a *spurious-using* classifier which has greater margin than the *purely-invariant* classifier.

□

### B.4 Proof of necessary condition

In this section we will show that Assm 3.3 is also a necessary condition for the probing classifier to use the spuriously correlated main task features ($\boldsymbol{z}_m$) when the dimension of concept-causal feature $d_p = 1$. That is, the probing classifier will use the spurious features if and only if spurious features satisfy Assm 3.3 for the margin points of the clean/purely-invariant (Def B.1) probing classifier when the concept-causal feature is 1-dimensional. Also, same line of reasoning will hold for the main-task classifier where we will show that main-task classifier will use the spurious feature ($\boldsymbol{z}_p$) iff spurious feature satisfies Assm B.2 for the margin point of clean main-task classifier. Formally:

**Lemma B.2** (Necessary Condition for concept-Probing Classifier). *Let the latent representation be frozen and disentangled (Assm 3.1) such that $\boldsymbol{z} = [\boldsymbol{z}_m, z_p]$ where $z_p$ is the concept-causal feature which is 1-dimensional scalar and fully predictive (Assm 3.2) and $\boldsymbol{z}_m \in \mathbb{R}^{d_m}$. Let $c_p^*(\boldsymbol{z}) = w_p \cdot z_p$ be the desired clean/purely-invariant probing classifier trained using max-margin objective which only uses $z_p$ for prediction. Then the probing classifier trained using max-margin objective will be spurious-using i.e. $c_p(\boldsymbol{z}) = w_p \cdot z_p + \boldsymbol{w}_m \cdot \boldsymbol{z}_m$ where $\boldsymbol{w}_m \neq 0$ iff the spurious feature $\boldsymbol{z}_m$ is linearly separable w.r.t to probing task label $y_p$ for the margin point of $c_p^*(\boldsymbol{z})$ (Assm 3.3).*

**Lemma B.3** (Necessary Condition for Main-task Classifier). *Let the latent representation be frozen and disentangled (Def 3.1) such that $\boldsymbol{z} = [z_m, \boldsymbol{z}_p]$ where $z_m$ is the main-task feature which is 1-dimensional scalar and fully predictive (Assm B.1) and $\boldsymbol{z}_p \in \mathbb{R}^{d_p}$. Let $c_m^*(\boldsymbol{z}) = w_m \cdot z_m$ be the desired clean/purely-invariant main-task classifier trained using max-margin objective which only uses $z_m$ for prediction. Then the main-task classifier trained using max-margin objective will be spurious-using i.e. $c_m(\boldsymbol{z}) = w_m \cdot z_m + \boldsymbol{w}_p \cdot \boldsymbol{z}_p$ where $\boldsymbol{w}_p \neq 0$ iff the spurious feature $\boldsymbol{z}_p$ is linearly separable w.r.t to main task label $y_m$ for the margin point of $c_m^*(\boldsymbol{z})$ (Assm B.2) .*

Since proof of both Lemma B.2 and B.3 follows same line of reasoning, hence for brevity, following §B.3, we will prove the lemma for a general classifier $c(\boldsymbol{z})$ trained using max-margin objective to predict the task-label $y$. Let the latent representation be of form $\boldsymbol{z} = [z_{inv}, \boldsymbol{z}_{sp}]$ where $z_{inv} \in \mathbb{R}$ is the feature causally derived from the task concept and $\boldsymbol{z}_{sp} \in \mathbb{R}_{sp}^d$ is the feature spuriously correlated

to task label $y$. With respect to probing classifier $c_p(\boldsymbol{z})$ in Lemma B.2 $z_{inv} := z_p$ and $\boldsymbol{z}_{sp} := \boldsymbol{z}_m$. Similarly, for the main-task classifier in Lemma B.3, $z_{inv} := z_m$ and $\boldsymbol{z}_{sp} := \boldsymbol{z}_p$.

*Proof of Lemma B.2 and B.3.* Our goal is to show that Assm 3.3 for probing classifier or Assm B.2 for the main-task classifier is necessary for obtaining a *spurious-using* classifier for the case when $z_{inv}$ is one-dimensional. We show this by assuming that optimal classifier is *spurious-using* even when Assm 3.3 or B.2 breaks and then show that this will lead to contradiction.

*Contradiction Assumption:* Formally, let's assume that Assm 3.3 or B.2 is not satisfied for probing or main task respectively, and the optimal classifier for the given classification task is *spurious-using* $c_*(\boldsymbol{z})$, where:

$$c_*(\boldsymbol{z}) = w_{inv}^* \cdot z_{inv} + \|\boldsymbol{w}_{sp}^*\|(\hat{\boldsymbol{w}}_{sp}^* \cdot \boldsymbol{z}_{sp}) \tag{30}$$

where $\|\boldsymbol{w}_{sp}^*\| \neq 0$ and $\hat{\boldsymbol{w}}_{sp}^* \in \mathbb{R}^{d_{sp}}$ is a unit vector in spurious-feature subspace with dimension $d_{sp}$.

Let $c_{inv}(\boldsymbol{z}) = w_{inv} \cdot z_{inv}$ be the optimal *purely-invariant* classifier. Let both $c_*(\boldsymbol{z})$ and $c_{inv}(\boldsymbol{z})$ be trained using the max-margin objective using *MM-Denominator* formulation in Eq. 4. Thus from the constraints of this formulation (Eq. 3), for all $\boldsymbol{z}$ we have:

$$m_*(\boldsymbol{z}) = y \cdot c_*(\boldsymbol{z}) = y \cdot (w_{inv}^* \cdot z_{inv} + \|\boldsymbol{w}_{sp}^*\|(\hat{\boldsymbol{w}}_{sp}^* \cdot \boldsymbol{z}_{sp})) \geq 1 \ , \& \tag{31}$$

$$m_{inv}(\boldsymbol{z}) = y \cdot c_{inv}(\boldsymbol{z}) = y \cdot (w_{inv} \cdot z_{inv}) \geq 1 \tag{32}$$

From Assm 3.2 or B.1, the invariant feature $z_{inv}$ is 100% predictive and linearly separable w.r.t task label $y$. Then without loss of generality let's assume that:

$$z_{inv} > 0, \quad \text{when} \quad y = +1 \tag{33}$$

$$z_{inv} < 0, \quad \text{when} \quad y = -1 \tag{34}$$

From Eq. 33 and 34 we have $y \cdot z_{inv} > 0$ thus from Eq. 32 we get:

$$w_{inv} \geq 0 \tag{35}$$

Also, from our *contradiction-assumption* the max-margin trained classifier is *spurious-using*, thus the norm of parameters of $c_*(\boldsymbol{z})$ is less or equal to $c_{inv}(\boldsymbol{z})$ (Eq. 4). Thus we have:

$$\sqrt{(w_{inv}^*)^2 + (\|\boldsymbol{w}_{sp}^*\|)^2} \leq |w_{inv}| \tag{36}$$

$$\implies |w_{inv}^*| < |w_{inv}| \qquad (\|\boldsymbol{w}_{sp}^*\| \neq 0) \tag{37}$$

$$\implies |w_{inv}^*| < w_{inv} \qquad (w_{inv} \geq 0, Eq.\ 35) \tag{38}$$

$$\implies w_{inv}^* < w_{inv} \tag{39}$$

Form our *contradiction-assumption*, Assm 3.3 for concept-probing task or Assm B.2 for the main-task breaks by one of the following two ways:

1. Opposite Side Failure: This occurs when the spurious part of margin points (of $c_{inv}(\boldsymbol{z})$) on opposite sides of decision-boundary of the optimal task classifier ($c_*(\boldsymbol{z}) = 0$) are not linearly-separable with respect to task label $y$. Formally, there exist two datapoints, $P^{m+} := [z_{inv}^{m+}, \boldsymbol{z}_{sp}^{m+}]$ and $P^{m-} := [z_{inv}^{m-}, \boldsymbol{z}_{sp}^{m-}]$ such that they are margin points of *purely-invariant* classifier $c_{inv}(\boldsymbol{z})$ where $P^{m+}$ has class label $y = +1$ and $P^{m-}$ has class label $y = -1$ and $\forall \hat{\boldsymbol{\epsilon}}_{sp} \in \mathbb{R}^{d_{sp}}$, the spurious feature $\boldsymbol{z}_{sp}$ of both the points lies on same side of $\hat{\boldsymbol{\epsilon}}_{sp}$ i.e:

$$\left((\hat{\boldsymbol{\epsilon}}_{sp} \cdot \boldsymbol{z}_{sp}^{m+}) \cdot (\hat{\boldsymbol{\epsilon}}_{sp} \cdot \boldsymbol{z}_{sp}^{m-})\right) \geq 0 \tag{40}$$

2. Same Side Failure: This occurs when the spurious part of margin points (of $c_{inv}(\boldsymbol{z})$) on same side of decision-boundary ($c_*(\boldsymbol{z}) = 0$) are linearly-separable. Formally, there exist two datapoints, $P_y^{m1} := [z_{inv}^{m1}, \boldsymbol{z}_{sp}^{m1}]$ and $P_y^{m2} := [z_{inv}^{m2}, \boldsymbol{z}_{sp}^{m2}]$ such that they are margin points of *purely-invariant* classifier $c_{inv}(\boldsymbol{z})$ and both points have same class label $y$ and $\forall \hat{\boldsymbol{\epsilon}}_{sp} \in \mathbb{R}^{d_{sp}}$, w.l.o.g we have:

$$\left((\hat{\boldsymbol{\epsilon}}_{sp} \cdot \boldsymbol{z}_{sp}^{m1}) \cdot (\hat{\boldsymbol{\epsilon}}_{sp} \cdot \boldsymbol{z}_{sp}^{m2})\right) \leq 0. \tag{41}$$

We will use the following two lemma to proceed with our proof:

**Lemma B.4.** *If Assm 3.3 or B.2 breaks by* opposite-side failure *mode, it leads to contradiction.*

**Lemma B.5.** *If Assm 3.3 or B.2 breaks by* same-side failure *mode, it leads to contradiction.*

This implies that our *contradiction-assumption* which said that the max-margin trained optimal classifier is *spurious-using* even when Assm 3.3 or B.2 breaks, is wrong. Thus Assm 3.3 for concept-probing task or Assm B.2 for main-task is necessary for the optimal max-margin classifier to be *spurious-using*. This completes our proof. □

*Proof of Lemma B.4.* We have two points, $P^{m+} := [z_{inv}^{m+}, \boldsymbol{z}_{sp}^{m+}]$ and $P^{m-} := [z_{inv}^{m-}, \boldsymbol{z}_{sp}^{m-}]$, which break the Assm 3.3 or B.2. From Eq. 33, $z_{inv} > 0$ for all the points with label $y = 1$, thus we have $z_{inv}^{m+} > 0$ and using Eq. 39 ($w_{inv}^* < w_{inv}$) we get:

$$w_{inv}^* < w_{inv} \tag{42}$$

$$w_{inv}^* \cdot z_{inv}^{m+} < w_{inv} \cdot z_{inv}^{m+} \tag{43}$$

$$w_{inv}^* \cdot z_{inv}^{m+} < 1 \tag{44}$$

where the right hand side $w_{inv} \cdot z_{inv}^{m+} = 1$ since $P^{m+}$ is the margin-point of $c_{inv}(\boldsymbol{z})$ (Eq. 32). Similarly from Eq. 34, $z_{inv} < 0$ for all the points with label $y = -1$, thus we have $z_{inv}^{m-} < 0$ and using Eq. 39 ($w_{inv}^* < w_{inv}$) we get:

$$w_{inv}^* < w_{inv} \tag{45}$$

$$(-1) \cdot w_{inv}^* \cdot z_{inv}^{m-} < (-1) \cdot w_{inv} \cdot z_{inv}^{m-} \tag{46}$$

$$(-1) \cdot w_{inv}^* \cdot z_{inv}^{m-} < 1 \tag{47}$$

where the right hand side $(-1) \cdot (w_{inv}^p \cdot z_{inv}^{m-}) = 1$ since $P^{m-}$ is the margin-point of $c_{inv}(\boldsymbol{z})$ (Eq. 32).

Next from Eq. 31 we have $m_*(\boldsymbol{z}) \geq 1$ for all $\boldsymbol{z}$ hence it is also true for $P^{m+}$ with $y = 1$ and $P^{m-}$ with $y = -1$. Then:

$$m_*(P^{m+}) = y \cdot c_*(P^{m+}) = 1 \cdot \left\{ w_{inv}^* z_{inv}^{m+} + \|\boldsymbol{w}_{sp}^*\| (\hat{\boldsymbol{w}}_{sp}^* \cdot \boldsymbol{z}_{sp}^{m+}) \right\} \geq 1 \tag{48}$$

$$\implies w_{inv}^* z_{inv}^{m+} + \|\boldsymbol{w}_{sp}^*\| \cdot \beta^{m+} \geq 1 \tag{49}$$

$$\implies w_{inv}^* z_{inv}^{m+} \geq 1 - \|\boldsymbol{w}_{sp}^*\| \cdot \beta^{m+} \tag{50}$$

where $\beta^{m+} = (\hat{\boldsymbol{w}}_{sp}^* \cdot \boldsymbol{z}_{sp}^{m+})$. Also we have:

$$m_*(P^{m-}) = y \cdot c_*(P^{m-}) = -1 \cdot \left\{ w_{inv}^* z_{inv}^{m-} + \|\boldsymbol{w}_{sp}^*\| (\hat{\boldsymbol{w}}_{sp}^* \cdot \boldsymbol{z}_{sp}^{m-}) \right\} \geq 1 \tag{51}$$

$$\implies -w_{inv}^* z_{inv}^{m-} - \|\boldsymbol{w}_{sp}^*\| \cdot \beta^{m-} \geq 1 \tag{52}$$

$$\implies -w_{inv}^* z_{inv}^{m-} \geq 1 + \|\boldsymbol{w}_{sp}^*\| \cdot \beta^{m-} \tag{53}$$

where $\beta^{m-} = (\hat{\boldsymbol{w}}_{sp}^* \cdot \boldsymbol{z}_{sp}^{m-})$. From Eq. 40 we have $((\hat{\boldsymbol{\epsilon}}_{sp} \cdot \boldsymbol{z}_{sp}^{m+}) \cdot (\hat{\boldsymbol{\epsilon}}_{sp} \cdot \boldsymbol{z}_{sp}^{m-})) \geq 0$ for all $\hat{\boldsymbol{\epsilon}}_{sp} \in \mathbb{R}^{d_{sp}}$ which states the *opposite-side failure* of Assm 3.3 or B.2. Thus:

$$\beta^{m+} \cdot \beta^{m-} \geq 0 \tag{54}$$

Now we will show that Eq. 44, 47, 50 and 53 cannot be satisfied simultaneously for any allowed value of $\beta^{m+}$ and $\beta^{m-}$ (given by Eq. 54) which are:

1. $\beta^{m+} > 0$ and $\beta^{m-} > 0$: From Eq. 53 we have $-w_{inv}^* z_{inv}^{m-} > 1$ since $\|\boldsymbol{w}_{sp}^*\| \neq 0$ and $\beta^{m-} > 0$. But from Eq. 47 we have $-w_{inv}^* z_{inv}^{m-} < 1$ which is a contradiction.

2. $\beta^{m+} < 0$ and $\beta^{m-} < 0$: From Eq. 50 we have $w_{inv}^* z_{inv}^{m+} > 1$ since $\|\boldsymbol{w}_{sp}^*\| \neq 0$ and $\beta^{m+} < 0$. But from Eq. 44 we have $w_{inv}^* z_{inv}^{m+} < 1$ which is a contradiction.

3. $\beta^{m+} = 0$ and $\beta^{m-} \in \mathbb{R}$: From Eq. 50 we have $w_{inv}^* z_{inv}^{m+} \geq 1$ but from Eq. 44 we have $w_{inv}^* z_{inv}^{m+} < 1$ which is a contradiction.

4. $\beta^{m+} \in \mathbb{R}$ and $\beta^{m-} = 0$: From Eq. 53 we have $-w^*_{inv}z^{m-}_{inv} \geq 1$ but from Eq. 47 we have $-w^*_{inv}z^{m-}_{inv} < 1$ which is a contradiction.

Thus we have a contradiction for all the possible values $\beta^{m+}$ and $\beta^{m-}$ could take, completing the proof of this lemma.

$\square$

*Proof of Lemma B.5.* We have two margin-points, $P^{m1}_y := [z^{m1}_{inv}, \boldsymbol{z}^{m1}_{sp}]$ and $P^{m2}_y := [z^{m2}_{inv}, \boldsymbol{z}^{m2}_{sp}]$, which break Assm 3.3 or B.2. From Eq. 33 and Eq. 34 we have $y \cdot z^{m1}_{inv} > 0$ and $y \cdot z^{m2}_{inv} > 0$. Using Eq. 39 ($w^*_{inv} < w_{inv}$) we get:

$$w^*_{inv} < w_{inv} \tag{55}$$

$$w^*_{inv} \cdot (y \cdot z^{mj}_{inv}) < w_{inv} \cdot (y \cdot z^{mj}_{inv}) \tag{56}$$

$$y \cdot (w^*_{inv} \cdot z^{mj}_{inv}) < 1 \tag{57}$$

where $j \in \{1, 2\}$ and right hand side $w_{inv} \cdot (y \cdot z^{mj}_{inv}) = 1$ since $P^{mj}_y$ is the margin point of purely-invariant classifier $c_{inv}(\boldsymbol{z})$ (Eq. 32).

From Eq. 31 we have $m_*(\boldsymbol{z}) \geq 1$ for all $\boldsymbol{z}$ thus also true for $P^{m1}_y$ and $P^{m2}_y$. Then:

$$m_*(P^{m1}_y) = y \cdot c_*(P^{m1}_y) = y \cdot \left\{ w^*_{inv}z^{m1}_{inv} + \|\boldsymbol{w}^*_{sp}\|(\hat{\boldsymbol{w}}^*_{sp} \cdot \boldsymbol{z}^{m1}_{sp}) \right\} \geq 1 \tag{58}$$

$$\implies y \cdot (w^*_{inv}z^{m1}_{inv}) + y \cdot (\|\boldsymbol{w}^*_{sp}\| \cdot \beta^{m1}) \geq 1 \tag{59}$$

$$\implies y \cdot (w^*_{inv}z^{m1}_{inv}) \geq 1 - y \cdot (\|\boldsymbol{w}^*_{sp}\| \cdot \beta^{m1}) \tag{60}$$

where $\beta^{m1} = (\hat{\boldsymbol{w}}^*_{sp} \cdot \boldsymbol{z}^{m1}_{sp})$. Also we have:

$$m_*(P^{m2}_y) = y \cdot c_*(P^{m2}_y) = y \cdot \left\{ w^*_{inv}z^{m2}_{inv} + \|\boldsymbol{w}^*_{sp}\|(\hat{\boldsymbol{w}}^*_{sp} \cdot \boldsymbol{z}^{m2}_{sp}) \right\} \geq 1 \tag{61}$$

$$\implies y \cdot (w^*_{inv}z^{m2}_{inv}) + y \cdot (\|\boldsymbol{w}^*_{sp}\| \cdot \beta^{m2}) \geq 1 \tag{62}$$

$$\implies y \cdot (w^*_{inv}z^{m2}_{inv}) \geq 1 - y \cdot (\|\boldsymbol{w}^*_{sp}\| \cdot \beta^{m2}) \tag{63}$$

where $\beta^{m2} = (\hat{\boldsymbol{w}}^*_{sp} \cdot \boldsymbol{z}^{m2}_{sp})$. Now from Eq. 41 we have $((\hat{\boldsymbol{\epsilon}}_{sp} \cdot \boldsymbol{z}^{m1}_{sp}) \cdot (\hat{\boldsymbol{\epsilon}}_{sp} \cdot \boldsymbol{z}^{m2}_{sp})) \leq 0$ for all unit vectors $\hat{\boldsymbol{\epsilon}}_{sp} \in \mathbb{R}^{d_{sp}}$ which states the *same-side* failure mode of Assm 3.3 or B.2. Thus we have:

$$\beta^{m1} \cdot \beta^{m2} \leq 0 \tag{64}$$

Now we will show that for all allowed values of $\beta^{m1}$ and $\beta^{m2}$, Eq. 57, 60 and 63 will lead to a contradiction. Following are the cases for different allowed values of $\beta^{m1}$ and $\beta^{m2}$:

1. $\beta^{m1} = 0$ and $\beta^{m2} \in \mathbb{R}$: Substituting $\beta^{m1} = 0$ in Eq. 60 we get $y \cdot (w^*_{inv}z^{m1}_{inv}) \geq 1$, but from Eq. 57 we have $y \cdot (w^*_{inv}z^{m1}_{inv}) < 1$. Thus we have a contradiction.

2. $\beta^{m1} \in \mathbb{R}$ and $\beta^{m2} = 0$: Substituting $\beta^{m2} = 0$ in Eq. 63 we get $y \cdot (w^*_{inv}z^{m2}_{inv}) \geq 1$, but from Eq. 57 we have $y \cdot (w^*_{inv}z^{m2}_{inv}) < 1$. Thus we have a contradiction.

3. The only case which is left now is when both $\beta^{m1}$ and $\beta^{m2}$ is non-zero but of opposite sign. Without loss of generality, let $\beta^{m1} > 0$, $\beta^{m2} < 0$ and $y = (+1)$: Substituting $\beta^{m2} < 0$ and $y = (+1)$ in Eq. 63 we get $y \cdot (w^*_{inv}z^{m2}_{inv}) \geq 1$, but from Eq. 57 we have $y \cdot (w^*_{inv}z^{m2}_{inv}) < 1$. Thus we have a contradiction.

4. $\beta^{m1} > 0$, $\beta^{m2} < 0$ and $y = (-1)$: Substituting $\beta^{m1} > 0$ and $y = (-1)$ in Eq. 60 we get $y \cdot (w^*_{inv}z^{m1}_{inv}) \geq 1$, but from Eq. 57 we have $y \cdot (w^*_{inv}z^{m1}_{inv}) < 1$. Thus we have a contradiction.

Thus we have a contradiction for all the possible values $\beta^{m1}$, $\beta^{m2}$ and $y$ could take, completing the proof of this lemma.

$\square$

# C Null-Space Removal Failure: Setup and Proof of Theorem 3.2

## C.1 Null-Space Setup

As described in §3, the given main-task classifier have an encoder $h : \boldsymbol{X} \to \boldsymbol{Z}$ mapping the input $\boldsymbol{X}$ to latent representation $\boldsymbol{Z}$. Post that, the main-task classifier $c_m : \boldsymbol{Z} \to Y_m$ is used to predict the main-task label $y_m^i$ from latent representation $\boldsymbol{z}^i$ for every input $\boldsymbol{x}^i$. Given this (pre) trained main-task classifier the goal of a post-hoc removal method is to remove any undesired/sensitive/spurious concept from the latent representation $\boldsymbol{Z}$ without retraining the encoder $h$ or main-task classifier $c_m(\boldsymbol{z})$.

The null space method [32, 13], henceforth referred to as *INLP*, is one such post-hoc removal method that removes a concept from latent space by projecting the latent space to a subspace that is not discriminative of that attribute. First, it estimates the subspace in the latent space discriminative of the concept we want to remove by training a probing classifier $c_p(\boldsymbol{z}) \to y_p$, where $y_p$ is the concept label. [32] used a linear probing classifier $(c_p(\boldsymbol{z}))$ to ensure that the any linear classifier cannot recover the removed concept from modified latent representation $\boldsymbol{z}'$ and hence the main task classifier $(c_m(\boldsymbol{z}'))$, which is also a linear layer, become invariant to removed attribute. Let linear probing classifier $c_p(\boldsymbol{z})$ be parametrized by a matrix $W$, and null-space of matrix $W$ is defined as space $N(W) = \{\boldsymbol{z}|W\boldsymbol{z} = \boldsymbol{0}\}$. Give the basis vectors for the $N(W)$ we can construct the projection matrix $P_{N(W)}$ such that $W(P_{N(W)}\boldsymbol{z}) = \boldsymbol{0}$ for all $\boldsymbol{z}$. This projection matrix is defined as the guarding operator $g := P_{N(\mathcal{W})}$ (estimated by $c_p(\boldsymbol{z})$), when applied on the $\boldsymbol{z}$ will remove the features which are discriminative of undesired concept from $\boldsymbol{z}$. For the setting when $Y_p$ is binary we have:

$$P_{N(W)} = I - \hat{w}\hat{w}^T \tag{65}$$

where $I$ is the identity matrix and $\hat{w}$ is the unit vector in the direction of parameters of classifier $c_p(\boldsymbol{z})$ ([32]). Also, the authors recommend running this removal step for multiple iterations to ensure that the unwanted concept is removed completely. Thus after $S$ steps of removal, the final guarding function is:

$$g := \prod_{i=1}^{S} P_{N(W)}^i \tag{66}$$

where $P_{N(W)}^i$ is the projection matrix at $i^{th}$ removal step. Amnesic Probing ([13]) builds upon this idea for testing whether concept is being used by a given pre-trained classifier or not. The core idea is to remove the concept we want to test from the latent representation. If the prediction of the given classifier is influenced by this removal then the concept was being used by the given classifier otherwise not.

## C.2 Null-Space Removal Failure : Proof of Theorem 3.2

**Theorem 3.2.** *Let $\boldsymbol{c}_m(\boldsymbol{z})$ be a pre-trained main-task classifier where the latent representation $\boldsymbol{z} = [\boldsymbol{z}_m, \boldsymbol{z}_p]$ satisfies Assm 3.1 and 3.2. Let $\boldsymbol{c}_p(\boldsymbol{z})$ be the probing classifier used by INLP to remove the unwanted features $\boldsymbol{z}_p$ from the latent representation. Under Assm 3.3, Lemma 3.1 is satisfied for the probing classifier $\boldsymbol{c}_p(\boldsymbol{z})$ such that $c_p(\boldsymbol{z}) = \boldsymbol{w}_p \cdot \boldsymbol{z}_p + \boldsymbol{w}_m \cdot \boldsymbol{z}_m$ and $\boldsymbol{w}_m \neq \boldsymbol{0}$. Then,*

1. ***Damage in the first step of INLP.*** *The first step of linear-INLP will corrupt the main-task features and this corruption is non-invertible with subsequent projection steps of INLP.*

   (a) ***Mixing:*** *If $\boldsymbol{w}_p \neq \boldsymbol{0}$, the main task $\boldsymbol{z}_m$ and concept-causal features $\boldsymbol{z}_p$ will get mixed such that $\boldsymbol{z}^{i(1)} = [g(\boldsymbol{z}_m^i, \boldsymbol{z}_p^i), f(\boldsymbol{z}_p^i, \boldsymbol{z}_m^i)] \neq [g(\boldsymbol{z}_m^i), f(\boldsymbol{z}_p^i)]$ for some function "$f$" and "$g$". Thus, the latent representation is no longer disentangled and removal of concept-causal features will also lead to removal of main task features.*

   (b) ***Removal:*** *If $\boldsymbol{w}_p = \boldsymbol{0}$, then the first projection step of INLP will do opposite of what is intended, i.e., damage the main task features $\boldsymbol{z}_m$ (in case $\boldsymbol{z}_m \in \mathbb{R}$, it will completely remove $\boldsymbol{z}_m$) but have no effect on the concept-causal features $\boldsymbol{z}_p$.*

2. ***Removal in the long term:*** *The L2-norm of the latent representation $\boldsymbol{z}$ decreases with every projection step as long as the parameters of probing classifier ($\boldsymbol{w}^k$) at a step "$k$" does not lie completely in the space spanned by parameters of previous probing classifiers, i.e., span($\boldsymbol{w}^1, \ldots, \boldsymbol{w}^{k-1}$), $\boldsymbol{z}^{i(k-1)}$, $\boldsymbol{z}^{i(0)}$ and $\boldsymbol{z}^{i(0)}$ in direction of $\boldsymbol{w}^k$ is not trivially zero. Thus, after sufficiently many steps, INLP can destroy all information in the representation s.t. $\boldsymbol{z}^{i(\infty)} = [\boldsymbol{0}, \boldsymbol{0}]$.*

The proof of Theorem 3.2 proceeds in following steps:

1. First using Lemma 3.1, we show that even under very favourable conditions probing classifier will not be clean i.e will also use other features in addition to concept-causal feature for it's prediction. Then, for the more likely case when probing classifier uses both main-task and concept-causal feature, we show that after first step of null-space projection (INLP), both the main-task features and concept-causal features get *mixed*.

2. Next, for the extreme case when probing classifier uses only main-task feature, the first step of INLP will do opposite of what is intended. It will damage the main-task feature but will have no effect on the concept-causal feature which we wanted to remove from latent space representation.

3. In addition, we also show that the *damage* or *mixing* of latent space after first step of INLP projection cannot be corrected in subsequent step since the projection operation is non-invertible.

4. Next, we show that the projection operation is lossy, i.e removes the norm of latent representation under some conditions. Hence after sufficient steps, INLP could destroy all the information in latent representation.

*Proof of Theorem 3.2.* **First Claim (1a).** Let $c_p(z) = w_p z_p + w_m z_m$ be the linear probing classifier trained to predict the concept label $y_p$ from the latent representation $z$. Since all the assumptions of Lemma 3.1 are satisfied for the probing classifier $c_p(z)$, it is *spurious using*, i.e., $w_m \neq 0$ and for the claim 1(a) we have $w_p \neq 0$. Since the concept label $y_p$ is binary, the projection matrix for the first step of INLP removal is defined as $P^1_{N(W)} = I - \hat{w}\hat{w}^T$ where $\hat{w}^T = [\hat{w}_m, \hat{w}_p]$, $\hat{w}_m$ and $\hat{w}_p$ are the unit norm parameters of $c_p(z)$ i.e $w_m$ and $w_p$ respectively. On applying this projection on the latent space representation $z^i$ we get new projected representation $z^{i(1)}$ s.t.:

$$\begin{bmatrix} z_m^{i(1)} \\ z_p^{i(1)} \end{bmatrix} = \left( I - \begin{bmatrix} \hat{w}_m \\ \hat{w}_p \end{bmatrix} \begin{bmatrix} \hat{w}_m & \hat{w}_p \end{bmatrix} \right) \begin{bmatrix} z_m^i \\ z_p^i \end{bmatrix} \tag{67}$$

$$= \begin{bmatrix} z_m^i \\ z_p^i \end{bmatrix} - \hat{c}_p(z^i) \cdot \begin{bmatrix} \hat{w}_m \\ \hat{w}_p \end{bmatrix} \qquad \text{define } \hat{c}_p(z^i) := \hat{w}_m \cdot z_m^i + \hat{w}_p \cdot z_p^i \tag{68}$$

$$= \begin{bmatrix} z_m^i - \hat{c}_p(z^i)\hat{w}_m \\ z_p^i - \hat{c}_p(z^i)\hat{w}_p \end{bmatrix} \tag{69}$$

$$= \begin{bmatrix} g(z_m^i, z_p^i) \\ f(z_m^i, z_p^i) \end{bmatrix} \tag{70}$$

Next, we will show that the main task features and probing features get mixed after projection. To do so, we first show that $g(z_m^i, z_p^i) \neq g(z_m^i)$ for some function $g$ i.e projected main task features $z_m^{i(1)} = g(z_m^i, z_p^i)$ are not independent of probing features $z_p^i$. From Eq. 69, we have:

$$z_m^{i(1)} = g(z_m^i, z_p^i) \tag{71}$$

$$= z_m^i - (\hat{w}_m \cdot z_m^i + \hat{w}_p \cdot z_p^i)\hat{w}_m \tag{72}$$

$$= (I - \hat{w}_m \hat{w}_m^T)z_m^i - (\hat{w}_p \cdot z_p^i)\hat{w}_m \tag{73}$$

In this case we are given $w_p \neq 0$ and $w_m \neq 0$. Since $z_p^i$ can take any value (subject to Assm 3.2), $\hat{w}_p \cdot z_p^i$ is not trivially zero for all $z_p^i \implies (\hat{w}_p \cdot z_p^i)\hat{w}_m \neq 0$. Thus $g(z_m^i, z_p^i)$ is not independent of $z_p^i$.

Next, we will show that $f(z_m^i, z_p^i) \neq f(z_p^i)$ for some function $f$ i.e projected probing feature $z_p^{i(1)} = f(z_m^i, z_p^i)$ is not independent of the main task feature $z_m^i$. From Eq. 69, we have:

$$z_p^{i(1)} = f(z_m^i, z_p^i) \tag{74}$$

$$= z_p^i - (\hat{w}_m \cdot z_m^i + \hat{w}_p \cdot z_p^i)\hat{w}_p \tag{75}$$

$$= (I - \hat{w}_p \hat{w}_p^T)z_p^i - (\hat{w}_m \cdot z_m^i)\hat{w}_p \tag{76}$$

Again, in this case we are given $\boldsymbol{w}_p \neq \boldsymbol{0}$ and $\boldsymbol{w}_m \neq \boldsymbol{0}$. Since $z_m^i$ can take any value (subject to Assm 3.3), $\hat{\boldsymbol{w}}_m \cdot z_m^i$ is not trivially zero for all $z_m^i \implies (\hat{\boldsymbol{w}}_m \cdot z_m^i)\hat{\boldsymbol{w}}_p \neq \boldsymbol{0}$. Thus $f(z_m^i, z_p^i)$ is not independent of $z_m^i$. Hence both concept-feature $\boldsymbol{z}_p$ and the main-task feature $\boldsymbol{z}_m$ got mixed after the first step of projection.

Next, we will show that this mixing of the main task and concept-causal feature cannot be corrected in subsequent steps of null-space projection. Formally, the following Lemma C.1 proves that the above projection matrix $(P_{N(W)}^1)$ which resulted in mixing of features is non-invertible. The subsequent steps of INLP applies projection transformation which can be combined into one single matrix $P_{N(W)}^{>1} = \prod_{j>1} P_{N(W)}^j$. In order for mixing to be reversed, we need $P_{N(W)}^{>1} \times P_{N(W)}^1 = I$, thus we need $P_{N(W)}^{>1} = (P_{N(W)}^1)^{-1}$ which is not possible from Lemma C.1. Hence the mixing of the main-task feature and the concept-causal feature which happened after the first step of projection cannot be corrected in the subsequent steps of INLP thus completing the first claim of our proof.

**Lemma C.1.** *The projection matrix $P_{N(W)}^j$ at any projection step of INLP is non invertible.*

*Proof of Lemma C.1.* The projection matrix for binary target case is defined as $P := P_{N(W)}^j = I - A$ where $A = \hat{\boldsymbol{w}}\hat{\boldsymbol{w}}^T$ be a $n \times n$ matrix and $\boldsymbol{w}$ is the parameter vector of the probing classifier $c_p(\boldsymbol{z})$ trained at $j^{th}$-step of INLP. We can see that $A$ is a symmetric matrix. Every symmetric matrix is diagonalizable (Equation W.9 in [11]), hence we can write $A = Q\Lambda Q^T$, where $Q$ is a some orthonormal matrix such that $QQ^T = I$ and $\Lambda = diag(\lambda_1, \ldots, \lambda_n)$ be a $n \times n$ diagonal matrix where the diagonal entries $(\lambda_1 \ldots \lambda_n)$ are the eigen-values of $A$. Since $QQ^T = I$ we can write $P = I - A = QQ^T - Q\Lambda Q^T = Q(I - \Lambda)Q^T$. Next, for the projection matrix $P$ to be invertible $P^{-1}$ should exist. We have:

$$P^{-1} = \left(Q(I - \Lambda)Q^T\right)^{-1} \tag{77}$$

$$= (Q^T)^{-1}(I - \Lambda)^{-1}Q^{-1} \tag{78}$$

$$= Q(I - \Lambda)^{-1}Q^T \tag{79}$$

So projection matrix is only invertible when $(I - \Lambda)$ is invertible. We will show next that $(I - \Lambda)$ is not invertible thus completing our proof. We have $I - \Lambda = diag(1 - \lambda_1, \ldots, 1 - \lambda_n)$, hence:

$$(I - \Lambda)^{-1} = diag(\frac{1}{1 - \lambda_1}, \ldots, \frac{1}{1 - \lambda_2}) \tag{80}$$

Now, if one of the eigenvalues of $A$ is 1, then the diagonal matrix $(I - \Lambda)$ is not invertible. If one of the eigenvalues of $A$ is 1, then there exist an eigenvector $\boldsymbol{x}$ such that $A\boldsymbol{x} = \hat{\boldsymbol{w}}\hat{\boldsymbol{w}}^T \times \boldsymbol{x} = 1 \times \boldsymbol{x}$. The vector $\boldsymbol{x} = \hat{\boldsymbol{w}}$ is the eigenvector of $A$ with eigenvalue 1: $A\hat{\boldsymbol{w}} = \hat{\boldsymbol{w}}\hat{\boldsymbol{w}}^T \times \hat{\boldsymbol{w}} = 1 \times \hat{\boldsymbol{w}}$ since $\hat{\boldsymbol{w}}^T \times \hat{\boldsymbol{w}} = 1$ as it is a unit vector. Hence the projection matrix is not invertible.

$\square$

**First Claim (1b).** For a probing classifier of form $c_p^{(1)}(\boldsymbol{z}) = \boldsymbol{w}_p \cdot \boldsymbol{z}_p + \boldsymbol{w}_m \cdot \boldsymbol{z}_m$ for the first step of INLP projection —denoted by superscript (1)— trained to predict concept label $y_p$ and Assm 3.1,3.2 and 3.3 of Lemma 3.1 is satisfied then we have $\boldsymbol{w}_m \neq \boldsymbol{0}$ i.e main task feature $\boldsymbol{z}_m$ will be used by probing classifier along with the concept feature $\boldsymbol{z}_p$. For this second case, we are given that $\boldsymbol{w}_p = \boldsymbol{0}$ i.e probing classifier will not use concept feature at all. This is only possible when the main-task feature is fully predictive of the concept label i.e Assm 3.3 is satisfied for all the points in the dataset, otherwise optimal probing classifier will use the concept-causal feature to achieve better margin and accuracy. Moreover, even if we assume Assm 3.3 is satisfied for all the points in the dataset, to have $\boldsymbol{w}_p = \boldsymbol{0}$, the margin achieved by probing classifier using only main task feature $(\boldsymbol{z}_m)$ should be bigger than any other classifier i.e one using both the main-task feature and probing feature or using probing feature alone. Thus, it is very unlikely that the optimal probing classifier will have $\boldsymbol{w}_p = \boldsymbol{0}$.

Having said this, even in the case when we have $\boldsymbol{w}_p = \boldsymbol{0}$, we show that the first projection step of INLP will do something unintended, i.e., damage the main-task features while having no effect on concept-causal features which we intended to remove. First, we will show that main-task features

will get damaged. From Eq. 73 we have:

$$z_m^{i(1)} = (I - \hat{w}_m \hat{w}_m^T) z_m^i - (\hat{w}_p \cdot z_p^i) \hat{w}_m \tag{81}$$

$$= z_m^i - (\hat{w}_m \cdot z_m^i) \hat{w}_m - \mathbf{0} \qquad \text{(since } w_p = \mathbf{0}) \tag{82}$$

Since $w_m \neq \mathbf{0}$ and $z_m^i$ can take any value (subject to Assm 3.3), $\hat{w}_m \cdot z_m^i$ is not trivially zero for all the $z_m^i \implies (\hat{w}_m \cdot z_m^i) \hat{w}_m \neq \mathbf{0}$. Thus, projected main-task feature $z_m^{i(1)} \neq z_m^i$. In case $z_m \in \mathbb{R}$, we have $\hat{w}_m = \hat{z}_m^i$, thus $(\hat{w}_m \cdot z_m^i) \hat{w}_m = z_m^i$. Consequently, $z_m^{i(1)} = z_m^i - z_m^i = \mathbf{0}$. Thus, first projection step of INLP leads to complete removal of main-task feature $z_m$ when $z_m \in \mathbb{R}$. Also, from Lemma C.1, this projection step is non-invertible and hence the main-task feature cannot be recovered back in the subsequent projection step.

Next, we will show the first projection step has no effect on the concept-causal features which we wanted to remove in the first place. From Eq. 76, we have:

$$z_p^{i(1)} = (I - \hat{w}_p \hat{w}_p^T) z_p^i - (\hat{w}_m \cdot z_m^i) \hat{w}_p \tag{83}$$

$$= z_p^i - \mathbf{0} - \mathbf{0} \qquad \text{(since } w_p = \mathbf{0}) \tag{84}$$

Thus the first step of projection has no effect on the concept-causal feature we wanted to remove. In the next step of projection, if we again have $w_p = \mathbf{0}$, then this same case will repeat. Otherwise if Assm 3.3 still holds for main-task feature for the margin points of optimal probing classifier $c_p^{*(2)}(z)$ for this second step of projection, then we will have both $w_m \neq \mathbf{0}$ and $w_p \neq \mathbf{0}$ and the first case of this theorem will apply.

**Second Claim.** Now for proving the second statement, we will make use of the following lemma. The proof of the lemma is given below the proof of this theorem.

**Lemma C.2.** *After every projection step of INLP, the norm of every latent representation $z^i$ decreases, i.e., $\|z^{i(k)}\| < \|z^{i(k-1)}\|$ for step $k$ and $k-1$, if (1) $z^{i(k-1)} \neq \mathbf{0}$, (2) $z_{\hat{w}^k}^{i(0)} \neq \mathbf{0}$ and (3) the parameters of probing classifier in step "k" i.e $\hat{w}^k$ don't lie in the space spanned by parameters of previous probing classifier, $\text{span}(\hat{w}^1, \ldots, \hat{w}^{k-1})$.*

Next, we will show that starting from the first step, at every $k^{\text{th}}$-step of projection either we will have $z^{i(k)} = \mathbf{0}$ or the norm will decrease after projection. Once we reach a step when $z^{i(k)} = \mathbf{0}$, then after every subsequent projection we will have $z^{i(k+1)} = \mathbf{0} \implies \|z^{i(k+1)}\| = 0$ since:

$$z^{i(k+1)} = P_{N(w^k)} z^{i(k)} = P_{N(w^k)} \mathbf{0} = \mathbf{0} \tag{85}$$

where $P_{N(w^k)}$ is the projection matrix at step "k". Since $\|\cdot\| \geq 0$ and $\|z^{i(k)}\|$ is decreasing with every stey, thus with large number of $z^{i(\infty)} \to \mathbf{0}$.

We will start with the first step of projection. In the second statement of this Theorem 3.2, we are given that $z^{i(0)}$ is not trivially zero in direction of $w^0$ i.e $z_{w^0}^{i(0)} \neq \mathbf{0}$ (satisfying Assm(2) of above Lemma C.2). We are also given that $z^{i(0)} \neq \mathbf{0}$ (satisfying Assm(1) of above lemma) and since this is the first step of projection Assm(3) of above Lemma C.2) is also satisfied. Thus from Lemma C.2, we have $\|z^{i(1)}\| < \|z^{i(0)}\|$. Now, either $z^{i(1)} = \mathbf{0}$, which will imply that $\|z^{i(1)}\| = 0$ and will remain 0 for all subsequent step (from Eq. 85). Otherwise if $z^{i(1)} \neq \mathbf{0}$, it satisfies the Assm(1) of Lemma C.2, for next step of projection. Since Assm (2) and (3) are already satisfied (from the assumption in the second claim of Theorem 3.2), again we will have $\|z^{i(2)}\| < \|z^{i(1)}\|$ and the same idea will repeat eventually making $z^{i(k)} = \mathbf{0}$ at some step-k, thus completing our proof.

$\square$

*Proof of Lemma C.2.* After $(k-1)$-steps of INLP let the latent space representation $z^i$ be denoted as $z^{i(k-1)}$. Let $\hat{w}^k$ be the parameters of classifier $c_p(z^{k-1})$ trained to predict the concept label $y_p$ which we want to remove at step $k$. Then prior to the projection step in the $k^{th}$ iteration of the INLP, we can write $z^{i(k-1)}$ as:

$$z_B^{i(k-1)} = z_{\hat{w}^k}^{i(k-1)} \hat{w}^k + z_{N(\hat{w}^k)}^{i(k-1)} N(\hat{w}^k) \tag{86}$$

where $B = \{\hat{\boldsymbol{w}}^k, N(\hat{\boldsymbol{w}}^k)\}$ is the basis set and $N(\hat{\boldsymbol{w}}^k)$ is the null-space of $\hat{\boldsymbol{w}}^k$. The parameter $\hat{\boldsymbol{w}}^k$ in this new basis is:

$$\hat{\boldsymbol{w}}_B^k = I_{\hat{\boldsymbol{w}}^k}\hat{\boldsymbol{w}}^k + \mathbf{0}N(\hat{\boldsymbol{w}}^k) \tag{87}$$

where $I_{\hat{\boldsymbol{w}}^k}$ is identity matrix with dimension $d_{\hat{\boldsymbol{w}}^k} \times d_{\hat{\boldsymbol{w}}^k}$. Now, in the new basis when we project the $\boldsymbol{z}^{k-1}$ to the null space of $\hat{\boldsymbol{w}}^{i(k)}$ we have:

$$\boldsymbol{z}^{i(k)} = P_{N(\hat{\boldsymbol{w}}^k)}\boldsymbol{z}^{i(k-1)} \tag{88}$$

$$\boldsymbol{z}_B^{i(k)} = \left(I - \hat{\boldsymbol{w}}_B^k(\hat{\boldsymbol{w}}_B^k)^T)\right)\boldsymbol{z}_B^{i(k-1)} \tag{89}$$

$$= \left(I - \begin{bmatrix} I_{\hat{\boldsymbol{w}}^k} \\ \mathbf{0} \end{bmatrix}\begin{bmatrix} I_{\hat{\boldsymbol{w}}^k} & \mathbf{0} \end{bmatrix}\right)\begin{bmatrix} \boldsymbol{z}_{\hat{\boldsymbol{w}}^k}^{i(k-1)} \\ \boldsymbol{z}_{N(\hat{\boldsymbol{w}}^k)}^{i(k-1)} \end{bmatrix} \tag{90}$$

$$= \begin{bmatrix} \boldsymbol{z}_{\hat{\boldsymbol{w}}^k}^{i(k-1)} \\ \boldsymbol{z}_{N(\hat{\boldsymbol{w}}^k)}^{i(k-1)} \end{bmatrix} - \begin{bmatrix} \boldsymbol{z}_{\hat{\boldsymbol{w}}^k}^{i(k-1)} \\ \mathbf{0} \end{bmatrix} \tag{91}$$

$$= \begin{bmatrix} \mathbf{0} \\ \boldsymbol{z}_{N(\hat{\boldsymbol{w}}^k)}^{i(k-1)} \end{bmatrix} \tag{92}$$

Thus the norm of $\|\boldsymbol{z}^{i(k)}\| = \sqrt{\|\boldsymbol{z}_{N(\hat{\boldsymbol{w}}^k)}^{i(k-1)}\| + 0}$ is less than $\|\boldsymbol{z}^{k-1}\| = \sqrt{\|\boldsymbol{z}_{\hat{\boldsymbol{w}}^k}^{i(k-1)}\|^2 + \|\boldsymbol{z}_{N(\hat{\boldsymbol{w}}^k)}^{i(k-1)}\|^2}$ if $\boldsymbol{z}_{\hat{\boldsymbol{w}}^k}^{i(k-1)} \neq 0$. Next we will show that $\boldsymbol{z}_{\hat{\boldsymbol{w}}^k}^{i(k-1)}$ cannot be zero. From assumption (2) in C.2 $\boldsymbol{z}_{\boldsymbol{w}^k}^{i(0)} \neq \mathbf{0}$ i.e $\boldsymbol{z}_{\boldsymbol{w}^k}^{i(0)}$ is not trivially zero in the given latent representation $\boldsymbol{z}^{i(0)}$ before any projection from INLP, thus $\boldsymbol{z}_{\hat{\boldsymbol{w}}^k}^{i(k-1)}$ is not trivially zero from beginning. Also, from Eq. 92, we observe that at any step "$k$" INLP removes the part of the representation from $\boldsymbol{z}^{i(k-1)}$ which is in the direction of $\hat{\boldsymbol{w}}^k$ i.e. $\boldsymbol{z}_{\hat{\boldsymbol{w}}^k}^{i(k-1)}$. Consequently, a sequence of removal steps with parameters $\hat{\boldsymbol{w}}^1, \ldots, \hat{\boldsymbol{w}}^{k-1}$ will remove the part of $\boldsymbol{z}$ which lies in the span($\hat{\boldsymbol{w}}^1, \ldots, \hat{\boldsymbol{w}}^{k-1}$). Thus $\boldsymbol{z}_{\hat{\boldsymbol{w}}^k}^{i(k-1)} = \mathbf{0}$ if $\hat{\boldsymbol{w}}^k$ lies in the span of parameters of previous classifier i.e span($\hat{\boldsymbol{w}}^1, \ldots, \hat{\boldsymbol{w}}^{k-1}$) which violates the assumption (3) in Lemma C.2. Thus $\boldsymbol{z}_{\hat{\boldsymbol{w}}^k}^{i(k-1)}$ is neither trivially zero from the beginning nor it could have been removed in the previous steps of projection as long as the assumption in Lemma C.2 is satisfied, which completes our proof of the lemma.

$\square$

**Remark.** *The following lemma from [32] tells us some of the sufficient conditions when the parameters of the probing classifier at the current iteration will not be same as the previous step.*

**Lemma C.3** (Lemma A.1 from [32]). *If the concept-probing classifier is being trained using SGD (stochastic gradient descent) and the loss function is convex, then parameters of the probing classifier at step $k$, $\boldsymbol{w}^k$, are orthogonal to parameters at step $k-1$, $\boldsymbol{w}^{k-1}$.*

*We conjecture that Lemma C.3 will be true for any loss since after $k-1$ steps of projection, the component of $\boldsymbol{z}$ in the direction of span($\boldsymbol{w}^1, \ldots, \boldsymbol{w}^{k-1}$) will be removed. Hence the concept-probing classifier at step $k$ should be orthogonal to span($\boldsymbol{w}^1, \ldots, \boldsymbol{w}^{k-1}$) in order to have non-random guess accuracy on probing task.*

# D   Adversarial Removal: Setup and Proof

## D.1   Adversarial Setup

As described in §3.3, let $h : \boldsymbol{X} \to \boldsymbol{Z}$ be an encoder mapping the input $\boldsymbol{x}$ to latent representation $\boldsymbol{z}$. The main task classifier $c_m : \boldsymbol{Z} \to Y_m$ is applied on top of $\boldsymbol{z}$ to predict the main task label $y_m$ for every input $\boldsymbol{x}$. As described in §3.3, the goal of an adversarial removal method, henceforth referred as AR, is to remove any undesired/sensitive/spurious concept from the latent representation $\boldsymbol{z}$. Once the concept is removed from the latent representation, any (main-task) classifier using the latent representation $\boldsymbol{Z}$ will not be able to use it [15, 48, 12]. These methods jointly train the main-task classifier $c_m(\boldsymbol{z})$ and the probing classifier $c_p : \boldsymbol{Z} \to Y_p$. The probing classifier is adversarially

trained to predict the concept label $y_p^i$ from latent representation $z^i$. Hence, AR methods optimize the following two objectives simultaneously:

$$arg \min_{c_p} L(c_p(h(\boldsymbol{x})), y_p) \tag{93}$$

$$arg \min_{h,c_m} \left\{ L(c_m(h(\boldsymbol{x}), y_m) - L(c_p(h(\boldsymbol{x}), y_p) \right\} \tag{94}$$

Here $L(\cdot)$ is a loss function which estimates the error given the ground truth $y_m/y_p$ and corresponding prediction $c_m(\boldsymbol{z})/c_p(\boldsymbol{z})$. The above adversarial objective between the encoder and probing classifier is a min-max game. The encoder wants to learn a latent representation $z$ s.t. it maximize the loss of probing classifier but at the same time probing classifier tries to minimize it's loss. The desired solution and simultaneously a valid equilibrium point of the above min-max objective is an encoder $h$ such that it removes all the features from latent space that are useful for prediction of $y_p$ while keeping intact other features causally derived from the main task prediction. In practice, the optimization to above objective is performed using a gradient-reversal (GRL) layer ([15]). It introduces an additional layer $g_\lambda$ between the latent representation $h(\boldsymbol{z})$ and the adversarial classifier $c_p(\boldsymbol{z})$. The $g_\lambda$ layer acts as an identity layer (i.e., has no effect) during the forward pass but scales the gradient by $(-\lambda)$ when back-propagating it during the backward pass. Thus resulting combined objective is:

$$arg \min_{h,c_m,c_p} \left\{ L(c_m(h(\boldsymbol{z})), y_m) + L(c_p(g_\lambda(h(\boldsymbol{z}))), y_p) \right\} \tag{95}$$

**Setup for theoretical result:** As stated in Theorem 3.3, for our theoretical result showing the failure mode of adversarial removal, we assume that the encoder is divided into two sub-parts. The first encoder i.e $h_1 : \boldsymbol{X} \to \boldsymbol{Z}$ is frozen (non-trainable) and maps the input $\boldsymbol{x}^i$ to intermediate latent representation $\boldsymbol{z}^i$ which is frozen and disentangled (Assm 3.1). The second encoder $h_2 : \boldsymbol{Z} \to \boldsymbol{\zeta}$ is a linear transformation mapping the intermediate latent representation $\boldsymbol{z}^i$ to final latent representation $\boldsymbol{\zeta}^i$ and is trainable. On top of this final latent representation $\boldsymbol{\zeta}^i$, we train the main task classifier $c_m(\boldsymbol{\zeta}^i)$ and probing classifier $c_p(\boldsymbol{\zeta}^i)$. Thus the training objective from Eq. 93 and 94 becomes:

$$arg \min_{c_p} L(c_p(h_2(\boldsymbol{z})), y_p) \tag{96}$$

$$arg \min_{h_2,c_m} \left\{ L(c_m(h_2(\boldsymbol{z}), y_m) - L(c_p(h_2(\boldsymbol{z}), y_p) \right\} \tag{97}$$

### D.2 Adversarial Proof

For a detailed discussion of adversarial training objective and specific setup for our theoretical result refer §D.1.

We formally state the new assumption made in the second statement of Theorem 3.3. This assumption imposes constraints on strength of correlation between main task label and concept-causal feature i.e it requires the concept-causal feature to be more predictive of main task label than for probing task.

**Assumption D.1** (Strength of Correlation). *Let $\hat{\boldsymbol{w}}_p \in \mathbb{R}^{d_p}$ be the unit vector s.t. $\boldsymbol{z}_p$ is linearly separable for concept-label $y_p$ (see Assm 3.2) and let $h_2^*(\boldsymbol{z})$ be the desired encoder which is successful in removing the concept-causal features $\boldsymbol{z}_p$ from $\boldsymbol{\zeta}$. Then, concept-causal features $\boldsymbol{z}_p^M$ of any margin point $\boldsymbol{z}^M$ of $c_m(h_2^*(\cdot))$ is more predictive of the main task than concept-causal features $\boldsymbol{z}_p^P$ of any margin-point $\boldsymbol{z}^P$ of $c_p(h_2^*(\cdot))$ for probing task by a factor of $|\beta| \in \mathbb{R}$ where $|\beta|$ is the norm of parameter of probing classifier $c_p(h_2^*(\cdot))$ i.e $y_m(\hat{\boldsymbol{w}}_p \cdot \boldsymbol{z}_p^M) > |\beta| y_p(\hat{\boldsymbol{w}}_p \cdot \boldsymbol{z}_p^P)$.*

**Theorem 3.3.** *Let the latent representation $\boldsymbol{z}$ satisfy Assm 3.1 and be frozen, $h_2(\boldsymbol{z})$ be a linear transformation over $\boldsymbol{Z}$ s.t. $h_2 : \boldsymbol{Z} \to \boldsymbol{\zeta}$, the main-task classifier be $c_m(\boldsymbol{\zeta}) = \boldsymbol{w}_{c_m} \cdot \boldsymbol{\zeta}$, and the adversarial probing classifier be $c_p(\boldsymbol{\zeta}) = \boldsymbol{w}_{c_p} \cdot \boldsymbol{\zeta}$. Let all the assumptions of Lemma B.1 be satisfied for main-classifier $c_m(\cdot)$ when using $\boldsymbol{z}$ directly as input and Assm 3.2 be satisfied on $\boldsymbol{z}$ w.r.t. the adversarial task. Let $h_2^*(\boldsymbol{z})$ be the desired encoder which is successful in removing $\boldsymbol{z}_p$ from $\boldsymbol{\zeta}$. Then there exists an undesired/incorrect encoder $h_2^\alpha(\boldsymbol{z})$ s.t. $h_2^\alpha(\boldsymbol{z})$ is dependent on $\boldsymbol{z}_p$ and the main-task classifier $c_m(h_2^\alpha(\boldsymbol{z}))$ has bigger margin than $c_m(h_2^*(\boldsymbol{z}))$ and has,*

1. *$Accuracy(c_p(h_2^\alpha(\boldsymbol{z})), y_p) = Accuracy(c_p(h_2^*(\boldsymbol{z})), y_p)$; when adversarial probing classifier $c_p(\cdot)$ is trained using any learning objective like max-margin or cross-entropy loss. Thus, the undesired encoder $h_2^\alpha(\boldsymbol{z})$ is indistinguishable from desired encoder $h_2^*(\boldsymbol{z})$ in terms of adversarial task prediction accuracy but better for main-task in terms of max-margin objective.*

2. $L_{h_2}\big(c_m(h_2^\alpha(\boldsymbol{z})), c_p(h_2^\alpha(\boldsymbol{z}))\big) < L_{h_2}\big(c_m(h_2^*(\boldsymbol{z})), c_p(h_2^*(\boldsymbol{z}))\big)$; *when Assm 3.4 is satisfied and concept-causal features* $\boldsymbol{z}_p^M$ *of any margin point* $\boldsymbol{z}^M$ *of* $c_m(h_2^*(\boldsymbol{z}))$ *are more predictive of the main task label than* $\boldsymbol{z}_p^P$ *of any margin point* $\boldsymbol{z}^P$ *of* $c_p(h_2^*(\boldsymbol{z}))$ *is predictive for the probing label (Assm D.1). Thus, undesired encoder* $h_2^\alpha(\boldsymbol{z})$ *is preferable over desired encoder* $h_2^*(\boldsymbol{z})$ *for both main and combined adversarial objective. Here* $L_{h_2} = L(c_m(\cdot)) - L(c_p(\cdot))$ *is the combined adversarial loss w.r.t. to* $h_2$ *and* $L(c(\cdot))$ *is the max-margin loss for a classifier "c" (see §D.1).*

*Proof of Theorem 3.3.* Let the main classifier be of the form $c_m(\boldsymbol{\zeta}) = \boldsymbol{w}_{c_m} \cdot \boldsymbol{\zeta}$ where $\boldsymbol{w}_{c_m}$ and $\boldsymbol{\zeta}$ are $d_\zeta$ dimensional vectors. Since both parameters $\boldsymbol{w}_{c_m}$ and $\boldsymbol{\zeta}$ are learnable, for ease of exposition we constrain $\boldsymbol{w}_{c_m}$ to be $[1, 0, \ldots, 0]$. This constraint on $\boldsymbol{w}_{c_m}$ is w.l.o.g. since $\boldsymbol{w}_{c_m}$ makes the prediction for main-task by projecting $\boldsymbol{\zeta}$ into one specific direction to get a scalar $(\boldsymbol{w}_{c_m} \cdot \boldsymbol{\zeta})$. We constrain that direction to be the first dimension of $\boldsymbol{\zeta}$. Since the encoder $h_2(\boldsymbol{z}) : \boldsymbol{Z} \to \boldsymbol{\zeta}$ is trainable it could learn to encode the scalar $(\boldsymbol{w}_{c_m} \cdot \boldsymbol{\zeta})$ in the first dimension of $\boldsymbol{\zeta}$. Thus, effectively a single dimension of the representation $\boldsymbol{\zeta}$ encodes the main-task information. As a result, the main classifier is effectively of the form $c_m(\boldsymbol{\zeta}) = \boldsymbol{w}_{c_m}^{(0)} \times \boldsymbol{\zeta}^{(0)} = 1 \times \boldsymbol{\zeta}^{(0)}$ where $\boldsymbol{w}_{c_m}^{(0)}$ and $\boldsymbol{\zeta}^{(0)}$ are the first elements of $\boldsymbol{w}_{c_m}$ and $\boldsymbol{\zeta}$ respectively and $\boldsymbol{w}_{c_m}^{(0)} = 1$. We can now write the goal of the adversarial method as removing the information of $\boldsymbol{z}_p$ from $\boldsymbol{\zeta}^{(0)}$ because the other dimensions are not used by the main classifier. Also, the adversarial probing classifier can be written effectively as $c_p(\boldsymbol{\zeta}) = \beta \times \boldsymbol{\zeta}^{(0)}$ where $\beta \in \mathbb{R}$ is a trainable parameter. Since both the main and adversarial classifier are using only $\boldsymbol{\zeta}^{(0)}$, the second encoder, with a slight abuse of notation, can be simplified as $\zeta := \boldsymbol{\zeta}^{(0)} := h_2(\boldsymbol{z}) = \boldsymbol{w}_m \cdot \boldsymbol{z}_m + \boldsymbol{w}_p \cdot \boldsymbol{z}_p$, where $\zeta \in \mathbb{R}$ and $\boldsymbol{w}_m$ and $\boldsymbol{w}_p$ are the weights that determine the first dimension of $\boldsymbol{\zeta}$. Also, let the desired (correct) second encoder which is successful in removing the concept-causal feature $\boldsymbol{z}_p$ from $\zeta$ be $\zeta^* := \boldsymbol{\zeta}^{*(0)} := h_2^*(\boldsymbol{z}) = \boldsymbol{w}_m^* \cdot \boldsymbol{z}_m$. Thus using Eq. 96 and 97, our overall objective for adversarial removal method becomes:

$$arg \min_\beta L(c_p(h_2(\boldsymbol{z})), y_p) \tag{98}$$

$$arg \min_{h_2} \left\{ L(c_m(h_2(\boldsymbol{z}), y_m) - L(c_p(h_2(\boldsymbol{z}), y_p) \right\} \tag{99}$$

**1. First claim.** The ideal main classifier with desired encoder can be written as, $c_m(\zeta^*) = 1 \times h_2^*(\boldsymbol{z}) = \boldsymbol{w}_m^* \cdot \boldsymbol{z}_m$. Therefore, it can be trained using the *MM-Denominator* formulation of the max-margin objective and would satisfy the constraint in Eq. 3:

$$m(c_m(\zeta^{*i})) = m(h_2^*(\boldsymbol{z}^i)) = y_m^i \cdot h_2^*(\boldsymbol{z}^i) \geq 1 \tag{100}$$

for all the points $\boldsymbol{x}^i$ with latent representation $\boldsymbol{z}^i$ and $m(\cdot)$ is the numerator of the distance of point from the decision boundary of classifier (Eq. 1).

However, the main task classifier which does not use the desired encoder is of the form, $c_m(\zeta) = 1 \times h_2^\alpha(\boldsymbol{z}) = \boldsymbol{w}_m \cdot \boldsymbol{z}_m + \boldsymbol{w}_p \cdot \boldsymbol{z}_p$. Since this main task classifier is trained using max-margin objective by *MM-Denominator* formulation, it would satisfy the constraint in Eq. 3:

$$m(c_m(\zeta^i)) = m(h_2^\alpha(\boldsymbol{z}^i)) = y_m^i \cdot h_2^\alpha(\boldsymbol{z}^i) \geq 1 \tag{101}$$

Since in our case, main task classifier is the same as the encoder i.e $c_m(\zeta) = 1 \times h_2^\alpha(\boldsymbol{z}) = \boldsymbol{w}_m \cdot \boldsymbol{z}_m + \boldsymbol{w}_p \cdot \boldsymbol{z}_p$, and the latent representation $\boldsymbol{z}$ satisfies the Assm 3.1, B.1 and B.2, from Lemma B.1 the main-task classifier is *spurious-using* i.e $\boldsymbol{z}_p \neq \boldsymbol{0}$. Hence there exists an undesired/incorrect encoder $h_2^\alpha(\boldsymbol{z})$ such that the main classifier $c_m(\zeta) = h_2^\alpha(\boldsymbol{z})$ has bigger margin than $c_m(\zeta^*) = h_2^*(\boldsymbol{z})$.

Next, we show that the accuracy of the adversarial classifier remains the same irrespective of whether the desired ($h_2^*(\boldsymbol{z})$) or undesired encoder $h_2^\alpha(\boldsymbol{z})$ is used. The accuracy of the adversarial classifier $c_p(\zeta) = \beta \times \zeta$, using the desired/correct encoder $\zeta = h_2^*(\boldsymbol{z})$ is given by:

$$Accuracy(c_p(\zeta^*), y_p) = \frac{\sum_{i=1}^N \mathbf{1}\Big(sign\big(\beta \cdot h_2^*(\boldsymbol{z}^i)\big) == y_p^i\Big)}{N} \tag{102}$$

where $\mathbf{1}(\cdot)$ is an indicator function which takes the value 1 if the argument is true otherwise 0, and $sign(\gamma) = +1$ if $\gamma \geq 0$ and $-1$ otherwise. Combining Eq. 100 and Eq. 101, since $y_m^i \in \{-1, 1\}$,

we see that whenever $h_2^\alpha(\boldsymbol{z}^i) > 1$ we also have $h_2^*(\boldsymbol{z}^i) > 1$ and similarly whenever $h_2^\alpha(\boldsymbol{z}^i) < -1$, we have $h_2^*(\boldsymbol{z}^i) < -1$. Thus,

$$h_2^\alpha(\boldsymbol{z}) \cdot h_2^*(\boldsymbol{z}) > 0 \tag{103}$$

From Eq. 103, $h_2^*(\boldsymbol{z}^i)$ and $h_2^\alpha(\boldsymbol{z}^i)$ has the same sign for every input $\boldsymbol{z}^i \implies sign(\beta \cdot h_2^*(\boldsymbol{z}^i)) = sign(\beta \cdot h_2^\alpha(\boldsymbol{z}^i))$. Thus we can replace $h_2^*(\boldsymbol{z}^i)$ with $h_2^\alpha(\boldsymbol{z}^i)$ in the above equation and we have:

$$Accuracy(c_p(\zeta^*), y_p) = \frac{\sum_{i=1}^{N} \mathbf{1}\Big(sign\big(\beta \cdot h_2^\alpha(\boldsymbol{z}^i)\big) == y_p^i\Big)}{N}$$

$$Accuracy(c_p(h_2^*(\boldsymbol{z^i})), y_p) = Accuracy(c_p(h_2^\alpha(\boldsymbol{z^i})), y_p)$$

thus completing the first part our proof.

**2. Second claim.** Since we are training both the main task and the probing classifier with a max-margin objective (see MM-Numerator version at Eq. 5), we can effectively write the adversarial objective (from 98 and 99) as:

$$arg\max_{\beta}(P(\beta)) := arg\max_{\beta}\left\{\min_{\boldsymbol{z}^i} m_{c_p}(h_2(\boldsymbol{z}^i))\right\} \tag{104}$$

$$arg\max_{h_2}(E(h_2)) := arg\max_{h_2}\left\{\min_{\boldsymbol{z}^i} m_{c_m}(h_2(\boldsymbol{z}^i)) - \min_{\boldsymbol{z}^i} m_{c_p}(h_2(\boldsymbol{z}^i))\right\} \tag{105}$$

where $m_{c_m}(h_2(\boldsymbol{z}))$ and $m_{c_p}(h_2(\boldsymbol{z}))$ are the numerator of margin of a point (Eq. 1). Next, our goal is to show that the desired encoder $h_2^*$ is not an equilibrium point of the above adversarial objective. To do so, we will create an undesired/incorrect encoder $h_2^\alpha(\boldsymbol{z})$ by perturbing $h_2^*$ by small amount and showing that the combined encoder objective $E(h_2^\alpha) > E(h_2^*)$ (Eq. 105) irrespective of choice of $\beta$ chosen by the probing objective $P(\beta)$ (Eq. 104).

**Construction of the undesired/incorrect encoder.** We have $h_2^*(\boldsymbol{z}) = \|\boldsymbol{w}_m\|(\hat{\boldsymbol{w}}_m^* \cdot \boldsymbol{z}_m)$ where $\hat{\boldsymbol{w}}_m^* \in \mathbb{R}^{d_m}$ is a unit vector. We will perturb this desired encoder by parameterizing with $\alpha \in [0, 1)$ s.t.:

$$h_2^\alpha(\boldsymbol{z}) = \alpha\|\boldsymbol{w}_m^*\|(\hat{\boldsymbol{w}}_m^* \cdot \boldsymbol{z}_m) + \sqrt{1-\alpha^2}\|\boldsymbol{w}_m^*\|(\hat{\boldsymbol{\epsilon}}_p \cdot \boldsymbol{z}_p) \tag{106}$$

where $\hat{\boldsymbol{\epsilon}}_p \in \mathbb{R}^{d_p}$ is a unit vector. The clean main-task classifier is defined as $c_m^*(h_2^*(\boldsymbol{z})) = h_2^*(\boldsymbol{z})$. The main-task classifier $c_m$ when using the incorrect encoder takes form $c_m(h_2^\alpha(\boldsymbol{z})) = h_2^\alpha(\boldsymbol{z})$. As stated in the theorem statement, all the assumptions of Lemma B.1 are satisfied. Since Assm B.2 (one of the assumptions of Lemma B.1) are satisfied, there exists a unit-vector in $\mathbb{R}^d_p$ such that concept-causal features of margin points of the main task classifier using encoder $h_2^*$ are linearly separable w.r.t main-task label. Let $\hat{\boldsymbol{\epsilon}}_p$ in our constructed undesired encoder $h_2^\alpha$ (Eq. 106) be set to that unit vector such that:

$$y_m^M \cdot (\hat{\boldsymbol{\epsilon}}_p \cdot \boldsymbol{z}_p^M) > 0 \tag{107}$$

where $\boldsymbol{z}_p^M$ is the concept-causal feature of margin point $\boldsymbol{z}^M$ of the main-task classifier when using encoder $h_2^*$. Now since all the assumption of Lemma B.1 is satisfied, the margin of main-task classifier when using undesired encoder $h_2^\alpha(\boldsymbol{z})$ is bigger than when desired encoder $h_2^*$ is used for some $\alpha \in (\alpha_{lb}^1, 1)$. Consequently, we have:

$$m_{c_m}(h_2^\alpha(\boldsymbol{z}^M)) > m_{c_m}(h_2^*(\boldsymbol{z}^M)) \tag{108}$$

where $\boldsymbol{z}^M$ is the margin point of $c_m(h_2^*)$. Since Assm 3.2 is satisfied, we have a fully predictive concept-causal feature $\boldsymbol{z}_p$ for prediction of adversarial label $y_p$ such that for some unit vector $\hat{\boldsymbol{w}}_p \in \mathbb{R}^{d_p}$ we have:

$$y_p^i(\hat{\boldsymbol{w}}_p \cdot \boldsymbol{z}_p^i) > 0 \quad \forall(\boldsymbol{z}^i, y_p^i) \tag{109}$$

Next, since Assm 3.4 is also satisfied for this second part of theorem, we have $y_p^i = y_m^i$ for every margin point of the desired/correct main-task classifier using the desired/correct encoder $h_2^*(\boldsymbol{z})$. Thus we can assign $\hat{\boldsymbol{\epsilon}}_p := \hat{\boldsymbol{w}}_p$ which satisfies the inequality in Eq. 107. Hence, our *incorrect* encoder $h_2^\alpha(\boldsymbol{z})$ take the following form:

$$h_2^\alpha(\boldsymbol{z}) = \alpha\|\boldsymbol{w}_m^*\|(\hat{\boldsymbol{w}}_m^* \cdot \boldsymbol{z}_m) + \sqrt{1-\alpha^2}\|\boldsymbol{w}_m^*\|(\hat{\boldsymbol{w}}_p \cdot \boldsymbol{z}_p) \tag{110}$$

Note that when $\alpha = 1$, we recover back the correct encoder $h_2^*$. Thus to perturb the $h_2^*$, we set $\alpha$ close to but less than 1.

**Showing $h_2^*$ is not the equilibrium point.** From Eq. 105, we want to show that for some $\alpha \in [0, 1)$ s.t. $\alpha \to 1$ ($\alpha$ close to but less than 1), the undesired encoder $h_2^\alpha$ has bigger combined objective than desired encoder $h_2^*$. Since the combined adversarial objective for encoder $h_2(z)$ ($E(h_2(z))$ in Eq. 105) is evaluated on the margin points of main-task and probing task classifier. We use the following lemma to show that for small perturbation of the optimal encoder ($\alpha \to 1$), the margin point of main-task classifier and probing classifier when using perturbed encoder $h_2^\alpha$ remains same or is a subset of margin points when using desired encoder $h_2^*$. The proof of the lemma below is given after the proof of the current theorem.

**Lemma D.1.** *There exist an $\alpha_{lb}^2 \in [0, 1)$ s.t. when $\alpha > \alpha_{lb}^2$ we have: (i) margin points of probing classifier when using perturbed encoder $h_2^\alpha$ is same or is a subset of margin points when using desired encoder $h_2^*$. (ii) margin points of main-task classifier when using perturbed encoder $h_2^\alpha$ is same or is a subset of margin points when using desired encoder $h_2^*$.*

Let $z^{M*}$ be one of the margin point of main-task classifier $c_m$ and $z^{P*}$ be one of the margin point of probing classifier $c_p$ when using the correct encoder $h_2^*$. Let $z^{M\alpha}$ be one of the margin point of main-task classifier $c_m$ and $z^{P\alpha}$ be one of the margin point of probing classifier $c_p$ when using the perturbed encoder $h_2^\alpha$. Thus we want to show that for all $(z^{M*}, z^{P*}, z^{M\alpha}, z^{P\alpha})$ tuple, there exists some $\alpha$ close to but less than 1 s.t. we have:

$$E(h_2^\alpha) > E(h_2^*) \tag{111}$$

$$m_{c_m}(h_2^\alpha(z^{M\alpha})) - m_{c_p}(h_2^\alpha(z^{P\alpha})) > m_{c_m}(h_2^*(z^{M*})) - m_{c_p}(h_2^*(z^{P*})) \tag{112}$$

$$m_{c_m}(h_2^\alpha(z^{M\alpha})) - m_{c_m}(h_2^*(z^{M*})) > m_{c_p}(h_2^\alpha(z^{P\alpha})) - m_{c_p}(h_2^*(z^{P*})) \tag{113}$$

**For $\beta < 0$.** From Lemma D.1, both $z^{P\alpha}$ and $z^{P*}$ are the margin point of probing classifier when using the desired encoder $h_2^*$. Thus we have:

$$m_{c_p}(h_2^*(z)^{P\alpha}) = m_{c_p}(h_2^*(z^{P*})) \tag{114}$$

$$y_p^{P\alpha}\beta(\boldsymbol{w}_m^* \cdot \boldsymbol{z}_m^{P\alpha}) = y_p^{P*}\beta(\boldsymbol{w}_m^* \cdot \boldsymbol{z}_m^{P*}) \tag{115}$$

Also, since $\alpha \in [0, 1)$, from above equation we have:

$$y_p^{P\alpha}\beta\alpha(\boldsymbol{w}_m^* \cdot \boldsymbol{z}_m^{P\alpha}) < y_p^{P*}\beta(\boldsymbol{w}_m^* \cdot \boldsymbol{z}_m^{P*}) \tag{116}$$

From Eq. 109 we have $y_p^{P\alpha}(\hat{\boldsymbol{w}}_p \cdot \boldsymbol{z}_p^{P\alpha}) > 0$. Since $\beta < 0$ and $\alpha \in [0, 1)$, we have $\sqrt{1-\alpha^2}\beta\|\boldsymbol{w}_m^*\|y_p^{P\alpha}(\hat{\boldsymbol{w}}_p \cdot \boldsymbol{z}_p^{P\alpha}) < 0$. Adding this to LHS of the above equation we get:

$$y_p^{P\alpha}\beta\alpha(\boldsymbol{w}_m^* \cdot \boldsymbol{z}_m^{P\alpha}) + \sqrt{1-\alpha^2}\beta\|\boldsymbol{w}_m^*\|y_p^{P\alpha}(\hat{\boldsymbol{w}}_p \cdot \boldsymbol{z}_p^{P\alpha}) < y_p^{P*}\beta(\boldsymbol{w}_m^* \cdot \boldsymbol{z}_m^{P*}) \tag{117}$$

$$y_p^{P\alpha}\beta\Big\{\alpha(\boldsymbol{w}_m^* \cdot \boldsymbol{z}_m^{P\alpha}) + \sqrt{1-\alpha^2}\|\boldsymbol{w}_m^*\|(\hat{\boldsymbol{w}}_p \cdot \boldsymbol{z}_p^{P\alpha})\Big\} < y_p^{P*}\beta(\boldsymbol{w}_m^* \cdot \boldsymbol{z}_m^{P*}) \tag{118}$$

$$y_p^{P\alpha}\beta h_2^\alpha(z^{P\alpha}) < y_p^{P*}\beta(\boldsymbol{w}_m^* \cdot \boldsymbol{z}_m^{P*}) \tag{119}$$

$$m_{c_p}(h_2^\alpha(z^{P\alpha})) < m_{c_p}(h_2^*(z^{P*})) \tag{120}$$

From Eq. 120 the RHS of Eq. 113 is less than zero. Also, from Eq. 108, for $\alpha \in (\alpha_{lb}^1, 1)$ we have $m_{c_m}(h_2^\alpha(z^{M\alpha})) - m_{c_m}(h_2^*(z^{M*})) > 0$ where value of $\alpha_{lb}^1$ is given by Lemma B.1. Thus the LHS of Eq. 113 is greater than 0. Thus the inequality in 113 is always satisfied when $\beta < 0$ and $\alpha \in (max\{\alpha_{lb}^1, \alpha_{lb}^2\}, 1)$. The constraint $\alpha > \alpha_{lb}^1$ is enforced by Lemma B.1 when constructing the perturbed encoder and $\alpha > \alpha_{lb}^2$ is enforced by Lemma D.1 which ensures $z^{P\alpha}$ is also a margin point of probing classifier when using desired encoder $h_2^*$. Hence, we have shown that when $\beta < 0$, $h_2^*$ is not the equilibrium point since there exist a perturbed undesired encoder $h_2^\alpha$ such that the combined encoder objective is greater in Eq. 105 and consequently the optimizer will try to move away from/change $h_2^*$.

**For $\beta > 0$.** Next we have to show that there exist $\alpha \in [0, 1)$ s.t. when $\alpha \to 1$ we have Eq. 113 satisfied. Thus we solve for allowed values of $\alpha$:

$$\Big\{m_{c_m}(h_2^\alpha(z^{M\alpha})) - m_{c_m}(h_2^*(z^{M*}))\Big\} > \Big\{m_{c_p}(h_2^\alpha(z^{P\alpha})) - m_{c_p}(h_2^*(z^{P*}))\Big\} \tag{121}$$

$$\Big\{y_m h_2^\alpha(z^{M\alpha}) - y_m h_2^*(z^{M*})\Big\} > \Big\{y_p(\beta \cdot h_2^\alpha(z^{P\alpha})) - y_p(\beta \cdot h_2^*(z^{P*}))\Big\} \tag{122}$$

From the second statement from Lemma D.1, $\boldsymbol{z}^{M_\alpha}$ and $\boldsymbol{z}^{M_*}$ both are margin point of main-task classifier using the desired encoder $h_2^*$. Thus we have $m_{c_m}(h_2^*(\boldsymbol{z}^{M_\alpha})) = m_{c_m}(h_2^*(\boldsymbol{z}^{M_*})) \implies y_m^{M_\alpha} h_2^*(\boldsymbol{z}^{M_\alpha}) = y_m^{M_*} h_2^*(\boldsymbol{z}^{M_*}) \implies y_m^{M_\alpha}(\boldsymbol{w}_m^* \cdot \boldsymbol{z}_m^{M_\alpha}) = y_m^{M_*}(\boldsymbol{w}_m^* \cdot \boldsymbol{z}_m^{M_*})$. Substituting this observation in LHS of Eq. 121 we get $m_{c_m}(h_2^\alpha(\boldsymbol{z}^{M_\alpha})) - m_{c_m}(h_2^*(\boldsymbol{z}^{M_*})) =$

$$= y_m^{M_\alpha}\Big\{\alpha(\boldsymbol{w}_m^* \cdot \boldsymbol{z}_m^{M_\alpha}) + \sqrt{1-\alpha^2}\|\boldsymbol{w}_m^*\|(\hat{\boldsymbol{w}}_p \cdot \boldsymbol{z}_p^{M_\alpha})\Big\} - y_m^{M_*}\Big\{(\boldsymbol{w}_m^* \cdot \boldsymbol{z}_m^{M_*})\Big\} \tag{123}$$

$$= y_m^{M_\alpha}\Big\{\alpha(\boldsymbol{w}_m^* \cdot \boldsymbol{z}_m^{M_\alpha}) + \sqrt{1-\alpha^2}\|\boldsymbol{w}_m^*\|(\hat{\boldsymbol{w}}_p \cdot \boldsymbol{z}_p^{M_\alpha})\Big\} - y_m^{M_\alpha}\Big\{(\boldsymbol{w}_m^* \cdot \boldsymbol{z}_m^{M_\alpha})\Big\} \tag{124}$$

$$= (\alpha-1)y_m^{M_\alpha}\Big\{\|\boldsymbol{w}_m^*\|(\hat{\boldsymbol{w}}_m^* \cdot \boldsymbol{z}_m^{M_\alpha})\Big\} + \sqrt{1-\alpha^2}y_m^{M_\alpha}\Big\{\|\boldsymbol{w}_m^*\|(\hat{\boldsymbol{w}}_p \cdot \boldsymbol{z}_p^{M_\alpha})\Big\} \tag{125}$$

$$= (\alpha-1)y_m^M\Big\{\|\boldsymbol{w}_m^*\|(\hat{\boldsymbol{w}}_m^* \cdot \boldsymbol{z}_m^M)\Big\} + \sqrt{1-\alpha^2}y_m^M\Big\{\|\boldsymbol{w}_m^*\|(\hat{\boldsymbol{w}}_p \cdot \boldsymbol{z}_p^M)\Big\} \tag{126}$$

where for ease of exposition we have defined $M := M_\alpha$. Now again for RHS of Eq. 121, from Lemma D.1, $\boldsymbol{z}^{P_\alpha}$ and $\boldsymbol{z}^{P_*}$ both are margin point of probing classifier using the desired encoder $h_2^*$. Thus we have $m_{c_p}(h_2^*(\boldsymbol{z}^{P_\alpha})) = m_{c_p}(h_2^*(\boldsymbol{z}^{P_*})) \implies y_p^{P_\alpha}\beta h_2^*(\boldsymbol{z}^{P_\alpha}) = y_p^{P_*}\beta h_2^*(\boldsymbol{z}^{P_*}) \implies y_p^{P_\alpha}(\boldsymbol{w}_m^* \cdot \boldsymbol{z}_m^{P_\alpha}) = y_p^{P_*}(\boldsymbol{w}_m^* \cdot \boldsymbol{z}_m^{P_*})$. Substituting this observation in RHS of Eq. 121 we get $m_{c_p}(h_2^\alpha(\boldsymbol{z}^{P_\alpha})) - m_{c_p}(h_2^*(\boldsymbol{z}^{P_*})) =$

$$= y_p^{P_\alpha}\Big\{\alpha\beta(\boldsymbol{w}_m^* \cdot \boldsymbol{z}_m^{P_\alpha}) + \sqrt{1-\alpha^2}\|\boldsymbol{w}_m^*\|\beta(\hat{\boldsymbol{w}}_p \cdot \boldsymbol{z}_p^{P_\alpha})\Big\} - y_p^{P_*}\Big\{\beta(\boldsymbol{w}_m^* \cdot \boldsymbol{z}_m^{P_*})\Big\} \tag{127}$$

$$= y_p^{P_\alpha}\Big\{\alpha\beta(\boldsymbol{w}_m^* \cdot \boldsymbol{z}_m^{P_\alpha}) + \sqrt{1-\alpha^2}\|\boldsymbol{w}_m^*\|\beta(\hat{\boldsymbol{w}}_p \cdot \boldsymbol{z}_p^{P_\alpha})\Big\} - y_p^{P_\alpha}\Big\{\beta(\boldsymbol{w}_m^* \cdot \boldsymbol{z}_m^{P_\alpha})\Big\} \tag{128}$$

$$= (\alpha-1)y_p^{P_\alpha}\Big\{\|\boldsymbol{w}_m^*\|\beta(\hat{\boldsymbol{w}}_m^* \cdot \boldsymbol{z}_m^{P_\alpha})\Big\} + \sqrt{1-\alpha^2}y_p^{P_\alpha}\Big\{\|\boldsymbol{w}_m^*\|\beta(\hat{\boldsymbol{w}}_p \cdot \boldsymbol{z}_p^{P_\alpha})\Big\} \tag{129}$$

$$= (\alpha-1)y_p^P\Big\{\|\boldsymbol{w}_m^*\|\beta(\hat{\boldsymbol{w}}_m^* \cdot \boldsymbol{z}_m^P)\Big\} + \sqrt{1-\alpha^2}y_p^P\Big\{\|\boldsymbol{w}_m^*\|\beta(\hat{\boldsymbol{w}}_p \cdot \boldsymbol{z}_p^P)\Big\} \tag{130}$$

where for ease of exposition we have defined $P := P_\alpha$. Substituting RHS (Eq. 130) and LHS (Eq. 126) back in Eq. 121 and rearranging we get:

$$\sqrt{1-\alpha^2}\Big\{y_m^M(\hat{\boldsymbol{w}}_p \cdot \boldsymbol{z}_p^M) - y_p^P\beta(\hat{\boldsymbol{w}}_p \cdot \boldsymbol{z}_p^P)\Big\} > (1-\alpha)\Big\{y_m^M(\hat{\boldsymbol{w}}_m^* \cdot \boldsymbol{z}_m^M) - y_p^P\beta(\hat{\boldsymbol{w}}_m^* \cdot \boldsymbol{z}_m^P)\Big\} \tag{131}$$

Now, since Assm B.1 is satisfied, the main task feature $\boldsymbol{z}_m^M$ is linearly separable w.r.t main-task label $y_m^M$. Thus we have $y_m^M(\hat{\boldsymbol{w}}_m^* \cdot \boldsymbol{z}_m^M) > 0$.

**Case 1: Main-task feature is not fully predictive of probing label** ($\exists \boldsymbol{z}$ s.t. $y_p(\hat{\boldsymbol{w}}_m^* \cdot \boldsymbol{z}_m) < 0$). Since main-task feature is not fully predictive of the probing label $y_p$, there will be some points which will be misclassified (will be on the opposite side of decision boundary) when probing classifier uses desired encoder $c_p(h_2^*(\boldsymbol{z})) = \beta h_2^*(\boldsymbol{z}) = \boldsymbol{w}_m^* \cdot \boldsymbol{z}_m$. Thus margin for those points will be negative and one of them will be the margin point $\boldsymbol{z}^P$ of the probing classifier. That is, $m_{c_p}(h_2^*(\boldsymbol{z}^P)) = y_p^P\beta(\hat{\boldsymbol{w}}_m^* \cdot \boldsymbol{z}_m^P) < 0$. Then the term $\big(y_m^M(\hat{\boldsymbol{w}}_m^* \cdot \boldsymbol{z}_m^M) - y_p^P\beta(\hat{\boldsymbol{w}}_m^* \cdot \boldsymbol{z}_m^P)\big) > 0$ in the above Eq. 131. Hence, rewriting the above equation we have:

$$\frac{\Big\{y_m^M(\hat{\boldsymbol{w}}_p \cdot \boldsymbol{z}_p^M) - y_p^P\beta(\hat{\boldsymbol{w}}_p \cdot \boldsymbol{z}_p^P)\Big\}}{\Big\{y_m^M(\hat{\boldsymbol{w}}_m^* \cdot \boldsymbol{z}_m^M) - y_p^P\beta(\hat{\boldsymbol{w}}_m^* \cdot \boldsymbol{z}_m^P)\Big\}} > \frac{1-\alpha}{\sqrt{1-\alpha^2}} \tag{132}$$

Next, from Lemma D.1 both $\boldsymbol{z}^M := \boldsymbol{z}^{M_\alpha}$ and $\boldsymbol{z}^P := \boldsymbol{z}^{P_\alpha}$ are also the margin point of main-task and probing classifier respectively when the classifiers use the desired encoder $h_2^\alpha$. Then, since Assm D.1 is satisfied the numerator in LHS of above equation $\big(y_m^M(\hat{\boldsymbol{w}}_p \cdot \boldsymbol{z}_p^M) - y_p^P\beta(\hat{\boldsymbol{w}}_p \cdot \boldsymbol{z}_p^P)\big) > 0$. Thus,

the whole LHS in the above equation is greater than zero. Denoting the LHS by $\gamma(\boldsymbol{z}^M, \boldsymbol{z}^P)$ gives us:

$$\gamma(\boldsymbol{z}^M, \boldsymbol{z}^P) > \frac{1 - \alpha}{\sqrt{1 - \alpha^2}} \tag{133}$$

$$\gamma^2(\boldsymbol{z}^M, \boldsymbol{z}^P) > \frac{\cancel{(1 - \alpha)}(1 - \alpha)}{\cancel{(1 - \alpha)}(1 + \alpha)} \tag{134}$$

$$\gamma^2(\boldsymbol{z}^M, \boldsymbol{z}^P) + \alpha\gamma^2(\boldsymbol{z}^M, \boldsymbol{z}^P) > 1 - \alpha \tag{135}$$

$$\left(1 + \gamma^2(\boldsymbol{z}^M, \boldsymbol{z}^P)\right)\alpha > 1 - \gamma^2(\boldsymbol{z}^M, \boldsymbol{z}^P) \tag{136}$$

$$\alpha > \frac{1 - \gamma^2(\boldsymbol{z}^M, \boldsymbol{z}^P)}{1 + \gamma^2(\boldsymbol{z}^M, \boldsymbol{z}^P)} = \alpha_{lb}^3(\boldsymbol{z}^M, \boldsymbol{z}^P)) \tag{137}$$

Since $\gamma^2(\boldsymbol{z}^M, \boldsymbol{z}^P) > 0$, $\alpha_{lb}^3(\boldsymbol{z}^M, \boldsymbol{z}^P)) < 1$. Let $\alpha_{lb}^3 = \max_{(\boldsymbol{z}^M, \boldsymbol{z}^P)}(\alpha_{lb}^3(\boldsymbol{z}^M, \boldsymbol{z}^P)))$ which is $< 1$ gives us the tight lower-bound on $\alpha$ such that Eq. 113 is satisfied for any pair of margin point $\boldsymbol{z}^M$ and $\boldsymbol{z}^P$.

**Case 2: Main-task is fully predictive of probing label.**  $(\forall \boldsymbol{z}, y_p(\hat{\boldsymbol{w}}_m^* \cdot \boldsymbol{z}_m) > 0)$. Since Assm B.1 (from Lemma B.1) is satisfied, we have that main-task features are fully predictive of main-task label i.e $y_m(\hat{\boldsymbol{w}}_m^* \cdot \boldsymbol{z}_m) > 0$ for all $\boldsymbol{z}$. Thus for this case $y_m(\hat{\boldsymbol{w}}_m^* \cdot \boldsymbol{z}_m) > 0$ and $y_p(\hat{\boldsymbol{w}}_m^* \cdot \boldsymbol{z}_m) > 0 \implies y_m = y_p$ for all $\boldsymbol{z}$. Also, for this case, there will be no misclassified points for the probing classifier when using the desired encoder $h_2^*$. Thus the margin point for both the main and the probing classifier is same i.e $\boldsymbol{z}^M = \boldsymbol{z}^P$. Since Assm D.1 is satisfied, $y_m = y_p$ for all $\boldsymbol{z}$, $y_p(\hat{\boldsymbol{w}}_p \cdot \boldsymbol{z}) > 0$ for all $\boldsymbol{z}$ from Assm 3.2 and $\boldsymbol{z}^P = \boldsymbol{z}^M$ we have:

$$y_m^M(\hat{\boldsymbol{w}}_p \cdot \boldsymbol{z}_p^M) > y_p^P \beta(\hat{\boldsymbol{w}}_p \cdot \boldsymbol{z}_p^P) \qquad (Assm \ D.1) \tag{138}$$

$$1 \cdot (\cancel{y_p^P(\hat{\boldsymbol{w}}_p \cdot \boldsymbol{z}_p^P)}) > \beta(\cancel{y_p^P(\hat{\boldsymbol{w}}_p \cdot \boldsymbol{z}_p^P)}) \tag{139}$$

$$\beta < 1 \tag{140}$$

Thus, in this case the RHS in Eq. 131, could be simplified to : $y_m^M(\hat{\boldsymbol{w}}_m^* \cdot \boldsymbol{z}_m^M) - y_p^P \beta(\hat{\boldsymbol{w}}_m^* \cdot \boldsymbol{z}_m^P) = y_m^M(\hat{\boldsymbol{w}}_m^* \cdot \boldsymbol{z}_m^M) - \beta y_m^M(\hat{\boldsymbol{w}}_m^* \cdot \boldsymbol{z}_m^M) = (1 - \beta)y_m^M(\hat{\boldsymbol{w}}_m^* \cdot \boldsymbol{z}_m^M) > 0$ since $0 < \beta < 1$ from above Eq. 140 and $y_m(\hat{\boldsymbol{w}}_m^* \cdot \boldsymbol{z}_m^M) > 0$ from Assm B.1. Thus we can rewrite Eq. 131 as:

$$\frac{\left\{y_m^M(\hat{\boldsymbol{w}}_p \cdot \boldsymbol{z}_p^M) - y_p^P \beta(\hat{\boldsymbol{w}}_p \cdot \boldsymbol{z}_p^P)\right\}}{\left\{y_m^M(\hat{\boldsymbol{w}}_m^* \cdot \boldsymbol{z}_m^M) - y_p^P \beta(\hat{\boldsymbol{w}}_m^* \cdot \boldsymbol{z}_m^P)\right\}} > \frac{1 - \alpha}{\sqrt{1 - \alpha^2}} \tag{141}$$

Again, from Lemma D.1 both $\boldsymbol{z}^M := \boldsymbol{z}^{M\alpha}$ and $\boldsymbol{z}^P := \boldsymbol{z}^{P\alpha}$ are also the margin point of main-task and probing classifier respectively when the classifiers use the desired encoder $h_2^\alpha$. Thus from Assm D.1, we have numerator of LHS in above equation greater than 0, thus we can follow the same steps from Eq. 133 to 137 to get the $\alpha_{lb}^3$ for this case.

So far, we have three lower bounds on $\alpha$ needed for this proof, so lets define $\alpha_{lb} = max\{\alpha_{lb}^1, \alpha_{lb}^2, \alpha_{lb}^3\}$, where $\alpha_{lb}^1$ is enforced by Lemma B.1 on undesired encoder $h_2^\alpha$ construction, $\alpha_{lb}^2$ is enforced by Lemma D.1 and $\alpha_{lb}^3$ is enforced by Eq. 113. Thus, when $\alpha \in (\alpha_{lb}, 1]$ we have a bigger combined objective (Eq. 105) for $h_2^\alpha$ than $h_2^*$. Thus, we can always perturb the desired encoder $h_2^\alpha$ by choosing $\alpha \in (\alpha_{lb}, 1]$ close to but less than 1 to create $h_2^\alpha$ which will have better combined encoder objective. Hence any optimizer will prefer to change the desired encoder $h_2^*$ and it is not an equilibrium solution to the overall adversarial objective.

$\square$

*Proof of Lemma D.1.*  First, we will prove the statement for the probing classifier. Let $\boldsymbol{z}^M$ be one of the margin points of the probing classifier when using the desired encoder $h_2^*$ and let $\boldsymbol{z}^R$ be any other (non-margin) points. Then we have to show that the margin-point of the probing classifier when using perturbed encoder $h_2^\alpha$ cannot be $\boldsymbol{z}^R$. This will imply that the margin points for probing classifier when using $h_2^\alpha$ has to be the same or a subset of margin points when using $h_2^*$. Since norm of parameters of both $c_p(h_2^\alpha(\boldsymbol{z})) = \beta h_2^\alpha(\boldsymbol{z})$ and $c_p(h_2^*(\boldsymbol{z})) = \beta h_2^*(\boldsymbol{z})$ is the same and margin-point of a classifier

is the point which have minimum margin, we have to show that $m_{c_p(h_2^\alpha)}(z^R) > m_{c_p(h_2^\alpha)}(z^M)$ for some $\alpha \in [0, 1)$. We have:

$$m_{c_p(h_2^\alpha)}(z) = \alpha y_p \Big\{ \beta(w_m^* \cdot z_m) \Big\} + \sqrt{1 - \alpha^2} \|w_m^*\| y_p \Big\{ \beta(\hat{w}_p \cdot z_p) \Big\} \tag{142}$$

$$= \alpha m_{c_p(h_2^*)}(z) + \sqrt{1 - \alpha^2} \|w_m^*\| y_p \Big\{ \beta(\hat{w}_p \cdot z_p) \Big\} \tag{143}$$

Thus we have to find an $\alpha \in [0, 1)$ s.t.:

$$\alpha m_{c_p(h_2^*)}(z^R) + \sqrt{1 - \alpha^2} \|w_m^*\| y_p^R \Big\{ \beta(\hat{w}_p \cdot z_p^R) \Big\} >$$
$$\alpha m_{c_p(h_2^*)}(z^M) + \sqrt{1 - \alpha^2} \|w_m^*\| y_p^M \Big\{ \beta(\hat{w}_p \cdot z_p^M) \Big\}$$

Rearranging we get:

$$\alpha \Big\{ m_{c_p(h_2^*)}(z^R) - m_{c_p(h_2^*)}(z^M) \Big\} > \sqrt{1 - \alpha^2} \|w_m^*\| \beta \Big\{ y_p^M(\hat{w}_p \cdot z_p^M) - y_p^R(\hat{w}_p \cdot z_p^R) \Big\} \tag{144}$$

Since $z^M$ is the margin point of the probing classifier when using $h_2^*$, we have $m_{c_p(h_2^*)}(z^R) > m_{c_p(h_2^*)}(z^M)$. Now, if $\beta \Big\{ y_p^M(\hat{w}_p \cdot z_p^M) - y_p^R(\hat{w}_p \cdot z_p^R) \Big\} \leq 0$, then above equation is trivially satisfied for all values of $\alpha \in (0, 1)$, since RHS of above equation is greater than 0 and LHS is less than 0. For the case when $\beta \Big\{ y_p^M(\hat{w}_p \cdot z_p^M) - y_p^R(\hat{w}_p \cdot z_p^R) \Big\} > 0$ we need:

$$\frac{\alpha}{\sqrt{1 - \alpha^2}} > \frac{\|w_m^*\| \beta \Big\{ y_p^M(\hat{w}_p \cdot z_p^M) - y_p^R(\hat{w}_p \cdot z_p^R) \Big\}}{\Big\{ m_{c_p(h_2^*)}(z^R) - m_{c_p(h_2^*)}(z^M) \Big\}} := \gamma(z^M, z^P) > 0 \tag{145}$$

$$\frac{\alpha^2}{1 - \alpha^2} > \gamma^2(z^M, z^P) \tag{146}$$

$$\alpha^2(1 + \gamma^2(z^M, z^P)) > \gamma^2(z^M, z^P) \tag{147}$$

$$\alpha > \sqrt{\frac{\gamma^2(z^M, z^P)}{1 + \gamma^2(z^M, z^P)}} := \alpha_{lb}^p(z^M, z^P) \tag{148}$$

Since we have $\gamma > 0 \implies \alpha_{lb}^p(z^M, z^P) < 1$. Lets define $\alpha_{lb}^p := \max_{(z^M, z^P)}(\alpha_{lb}^p(z^M, z^P)) < 1$, which gives the tightest lower bound on $\alpha$ s.t. when $\alpha \in (\alpha_{lb}^p, 1)$, the margin point of the probing classifier when using the perturbed encoder is same or is a subset of margin point when using desired encoder $h_2^*$. This completes the first part of the proof.

Next, we prove the second part of this lemma for the main-task classifier. Let $z^M$ be one of the margin points of the main-task classifier when using the desired encoder $h_2^*$ and let $z^R$ be any other (non-margin) point. Then we have to show that the margin-point of the main-task classifier when using perturbed encoder $h_2^\alpha$ cannot be $z^R$. Since norm of parameter of both $c_m(h_2^\alpha(z)) = h_2^\alpha(z)$ and $c_m(h_2^*(z)) = h_2^*(z)$ is same and margin-point of a classifier is the point which have minimum margin, we have to show that $m_{c_m(h_2^\alpha)}(z^R) > m_{c_m(h_2^\alpha)}(z^M)$ for some $\alpha \in [0, 1)$. We have:

$$m_{c_m(h_2^\alpha)}(z) = \alpha y_m \Big\{ (w_m^* \cdot z_m) \Big\} + \sqrt{1 - \alpha^2} \|w_m^*\| y_m \Big\{ (\hat{w}_p \cdot z_p) \Big\} \tag{149}$$

$$= \alpha m_{c_m(h_2^*)}(z) + \sqrt{1 - \alpha^2} \|w_m^*\| y_m \Big\{ (\hat{w}_p \cdot z_p) \Big\} \tag{150}$$

Thus we have find an $\alpha$ s.t.:

$$\alpha m_{c_m(h_2^*)}(z^R) + \sqrt{1 - \alpha^2} \|w_m^*\| y_m^R \Big\{ (\hat{w}_p \cdot z_p^R) \Big\} >$$
$$\alpha m_{c_m(h_2^*)}(z^M) + \sqrt{1 - \alpha^2} \|w_m^*\| y_m^M \Big\{ (\hat{w}_p \cdot z_p^M) \Big\}$$

Rearranging we get:

$$\alpha \Big\{ m_{c_m(h_2^*)}(z^R) - m_{c_m(h_2^*)}(z^M) \Big\} > \sqrt{1 - \alpha^2} \|w_m^*\| \Big\{ y_m^M(\hat{w}_p \cdot z_p^M) - y_m^R(\hat{w}_p \cdot z_p^R) \Big\} \tag{151}$$

Since $\boldsymbol{z}^M$ is the margin point of the probing classifier when using $h_2^*$, we have $m_{c_m(h_2^*)}(\boldsymbol{z}^R) > m_{c_m(h_2^*)}(\boldsymbol{z}^M)$. We notice that, apart from $y_p$ being set to $y_m$ and $\beta$ being set to 1, the above equation is identical to Eq. 144. Since our argument (from Eq. 144 to 148) to derive the allowed value of $\alpha$ doesn't depend on $y_p$ and $\beta$, we could follow the same argument to get a lower bound $\alpha_{lb}^m$ s.t. the main-task classifier has the same or subset of the margin points when using the perturbed encoder as it has when using the desired encoder.

Let us define $\alpha_{lb}^2 = \max\{\alpha_{lb}^p, \alpha_{lb}^m\}$. Thus when $\alpha \in (\alpha_{lb}^2, 1)$, both the statements of this lemma are satisfied thus completing our proof.

$\square$

# E   Experimental Setup

## E.1   Dataset Description

As described in §4, we demonstrate the failure of Null-Space Removal (§4.2) and Adversarial Removal (§4.3) in removing the undesired concept from the latent representation on three real-world datasets: MultiNLI [46], Twitter-PAN16 [31] and Twitter-AAE [6]; and a synthetic dataset, Synthetic-Text. The detailed generation and evaluation strategies for each dataset are given below.

**MultiNLI Dataset.**   In the MultiNLI dataset, given two sentences—premise and hypothesis—the main task is to predict whether the hypothesis *entails*, *contradicts* or is *neutral* to the premise. As described in §4, we simplify it to a binary task of predicting whether a hypothesis *contradicts* the premise. The binary main-task label, $y_m = 1$ when a given hypothesis *contradicts* the premise otherwise it is -1. That is, we relabel the MNLI dataset by assigning label $y_m = 1$ to examples with contradiction labels and $y_m = -1$ to the example with neutral or entailment label. It has been reported that the *contradiction* label is spuriously correlated with the negation words like *nobody, no, never* and *nothing*[16]. Thus, we created a 'negation' concept denoting the presence of these words in the hypothesis of a given (hypothesis, premise) pair. The concept-label $y_p = 1$ when the *negation* concept is present in the hypothesis otherwise it is $-1$.

The standard MultiNLI dataset [1] has approximately 90% of data points in the training set, 5% as publicly available development set and the rest of $5\%$ in a separate held-out validation set accessible through online competition leader-board not accessible to the public. Thus, we create our own train and test split by subsampling $10k$ examples from the initial training set, converting it into binary contradiction vs. non-contradiction labels, labeling the negation-concept label, and splitting them into 80-20 train and test split. For pre-training a clean classifier that does not use the spurious-concept, we create a special training set following the method described in §E.2. For evaluating the robustness of both null-space and adversarial removal methods, we create multiple datasets with different *predictive-correlation* as described in §E.3 .

**Twitter-PAN16 Dataset.**   In Twitter-PAN16 dataset [31], following [12], given a tweet, the main task is to predict whether it contains a mention of another user or not. The dataset contains manually annotated binary Gender information (i.e Male or Female) of 436 Twitter users with at least 1k tweets each. The Gender annotation was done by assessing the name and photograph of the LinkedIn profile of each user [12]. The unclear cases were discarded in this process. We consider "Gender" as a sensitive concept that should not be used for main-task prediction. The dataset contains 160k tweets for training and 10k tweets for the test. We merged the full dataset, subsampled 10k examples, and created an 80-20 train and test split. For pre-training a clean classifier, we create a special training set following the method described in §E.2. To generate datasets with different predictive correlation, we follow the method from E.3. The dataset is acquired and processed using the code[2] made available by the [12]. According to Twitter's policy, one has to download tweets from a personal account using Twitter Academic Research access and cannot be released to the public or used for commercial purposes. We also adhere to this policy and don't release any data to the public or use it elsewhere.

---

[1]MultiNLI dataset and its license could be found online at: `https://cims.nyu.edu/~sbowman/multinli/`

[2]The code for Twitter-PAN16 and Twitter-AAE dataset acquisition is available at: `https://github.com/yanaiela/demog-text-removal`

**Twitter-AAE Dataset.** In Twitter-AAE dataset [6], again following [12], the main task is to predict a binary sentiment (Positive or Negative) from a given tweet. The dataset contains 59.2 million tweets by 2.8 million users. Each tweet is associated with "race" information of the user which is labeled based on both words in the tweet and the geo-location of the user. We consider "race" as the sensitive concept which should not be used for the main task of sentiment prediction. We use the AAE (African America English) and SAE (Standard American English) as a proxy for non-Hispanic blacks and non-Hispanic whites automatically labeled using code made available by [12]. Again, we subsampled 10k examples with 80-20 split from the dataset and followed the method described in §E.2 and E.3 to generate a clean dataset for pre-training a clean classifier and datasets with different predictive correlation respectively. The dataset is made publicly available online[3] only for research-purpose.

**Synthetic Dataset.** To accurately evaluate the whether a classifier is using the spurious concept or not, we introduce a Synthetic-Text dataset where it is possible to change the text input based on the change in concept (thus implementing Def 2.1). The main-task is to predict whether a sentence contains a numbered word (e.g. *one, fifteen* etc) or not, and the spurious concept is the length of the sentence which is correlated with the main task label. To create a sentence with numbered words, we randomly sample 10 words from the following set and combine them to form the sentence.

$$
\begin{aligned}
\text{Numbered Words} = \ &\text{one, two, three, four, five, six, seven, eight,} \\
&\text{nine, ten, eleven, twelve, thirteen, fourteen,} \\
&\text{fifteen, sixteen, seventeen, eighteen, twenty,} \\
&\text{thirty, forty, fifty, sixty, seventy, eighty,} \\
&\text{ninety, hundred, thousand}
\end{aligned}
$$

Otherwise, a sentence is created by adding 10 non-numbered words randomly sampled from the following set.

$$
\begin{aligned}
\text{Non-Numbered Words} = \ &\text{nice, device, try, picture, signature, trailer,} \\
&\text{harry, potter, malfoy, john, switch, taste,} \\
&\text{glove, balloon, dog, horse, switch, watch,} \\
&\text{sun, cloud, river, town, cow, shadow,} \\
&\text{pencil, eraser}
\end{aligned}
$$

Next, we introduce the spurious concept (length) by increasing the length of the sentences which contain numbered words. We do so by adding a special word "pad" 10 times. In our experiments, we use 1k examples created using the above method and create an 80-20 split for the train and test set. Again, we follow the method described in §E.2 and E.3 to generate a clean dataset for training a clean classifier and to generate datasets with different predictive correlations respectively. To simulate a real-world setting, we also introduce noise in the main-task and the probing label. To introduce noise (denoted by $n = x$) in the labels, we randomly flip $100x\%$ of the main-task and probing label in the dataset. Wherever applicable, we will explicitly mention the amount of noise we add in the labels.

### E.2 Creating a "clean" dataset with no spurious correlation with main-label

Unless otherwise specified, to construct a new dataset with no spurious correlation between the main-task and the concept label, we subsample only those examples from the the given dataset which have a fixed value of the spurious-concept label ($y_p$). Thus, if we train main-task classifier using this dataset, it cannot use the spurious-concept since they are not discriminative of the main task label [35].

In MultiNLI dataset, we select only those examples which have no *negation* words in the sentence for creating a clean dataset. Similarly, for Twitter-PAN16 dataset, we only select those examples which have gender label $y_p = -1$ (Female) in the processed dataset. And for Twitter-AAE dataset, we only select those examples which have *non-Hispanic whites* race label.

---

[3]TwitterAAE dataset could be found online at: `http://slanglab.cs.umass.edu/TwitterAAE/`

### E.3 Creating datasets with spurious correlated main and concept label

In our experimental setup, both the main-task label ($y_m$) and concept label ($y_p$) are binary ($-1$ or $1$). This creates $2 \times 2$ subgroups for each combination of ($y_m, y_p$). In MultiNLI dataset, the contradiction label ($y_m = 1$) is correlated with the presence of negation words $y_p = 1$, this implies that the not-contradiction label $y_m = -1$ is also correlated with *absence* of negation words in the sentence $y_p = -1$. Thus, the input example with ($y_m = 1, y_p = 1$) and ($y_m = -1, y_p = -1$) form the majority group, henceforth referred as $S_{maj}$ while groups ($y_m = 1, y_p = -1$) and ($y_m = -1, y_p = 1$) forms the minority group $S_{min}$. To evaluate the robustness of the removal methods, we create multiple datasets with different *predictive correlation* ($\kappa$) between the two labels $y_m$ and $y_p$ where $\kappa = P(y_m \cdot y_p) > 0$ as defined in §4. In other words, to create a dataset with a particular predictive correlation $\kappa$, we vary the size of $S_{maj}$ and $S_{min}$. More precisely, the predictive correlation can be equivalently defined in terms of the size of the these groups as:

$$\kappa = \frac{|S_{maj}|}{|S_{maj}| + |S_{min}|} \tag{152}$$

Similarly for Twitter-PAN16, Twitter-AAE, and Synthetic-Text datasets, we create datasets with different levels of spurious correlation between $y_m$ and $y_p$ by creating the $S_{maj}$ and $S_{min}$ to have the desired predictive correlation ($\kappa$).

### E.4 Encoder for real datasets

For all the experiments on real datasets in §4 we used RoBERTa as default encoder $h$. In §F, we report the results when using BERT instead of RoBERTa as input encoder.

**RoBERTa**   We use the Hugging Face[47] *transformers* implementation of RoBERTa[25] *roberta-base* model, starting with pretrained weights for encoding the text-input to latent representation. We use a default tokenizer and model configuration in our experiment.

**BERT**   We use the Hugging Face[47] *transformers* implementation of BERT[10] *bert-base-uncased* model, starting with pretrained weights for encoding the text-input to latent representation. We use a default tokenizer and model configuration in our experiment.

For both BERT and RoBERTa, the parameters of the encoder were fine-tuned as a part of training the main-task classifier for null-space removal and then frozen. For adversarial removal, the encoder, main-task classifier and the adversarial probing classifier are trained jointly. For both BERT and RoBERTa, we use the pooled output ($[CLS]$ token for BERT) from the the model, as the latent representation and is given to main-task and probing classifier. Main-task and probing classifier are a linear transformation layer followed by a softmax layer for prediction. We use a batch size of 32 samples for all training procedures that use BERT or RoBERTa for encoding the input.

### E.5 Encoder for synthetic Dataset

**nBOW: neural Bag of Word.**   For Synthetic-Text dataset, we use sum of pretrained-GloVe embedding[30] of the words in the sentence to encode the sentence into latent representation. We used Gensim [36] library for acquiring the 100-dimensional GloVe embedding (*glove-wiki-gigaword-100*). Throughout all our experiments, the word embedding was not trained. Post encoding, the latent representation were further passed through hidden layers consisting of a linear transformation layer followed by ReLU non-linearity. We will specify how many such hidden layers were used when discussing specific experiments in §F. The hidden layer dimensions were fixed to 50 dimensional space. We use a batch size of 32 samples for all training procedures that use nBOW for encoding the input.

### E.6 Null-Space Removal Experiment Setup

For null-space removal (INLP) experiment on both real and synthetic dataset the following procedure is followed:

1. Pretraining Phase: A *clean* pretrained main-task classifier is trained using the *clean* dataset obtained by method described in §E.2. This is to ensure that the main-task classifier does not

use the spurious feature, so that the INLP method doesn't have any effect on the main-task classifier. The main-task classifier is a linear-transformation on the latent-representation provided by encoder followed by softmax layer for prediction. Both the encoder and main-task classifier is fine-tuned during this process.

2. Removal Phase: Both the encoder and main-task classifier is frozen (made non-trainable). Next, a probing classifier is trained from the latent representation of the encoder (refer §E.4 and E.5 for more details about encoder). The probing classifier is also a linear transformation layer followed by softmax layer for prediction. For experiments on real-world datasets using BERT or RoBERTa as encoder, we train the the probing classifier for 1 epoch (one full pass though the probing dataset) before each projection step. For experiment on the Synthetic-Text dataset, we train the probing classifier for 10 epochs before each projection step. Note that, we also experiment with the setting when the main task classifier is also trained after every step of INLP projection (see §F.2 and Fig. 9 for results). The main task classifier is a linear transformation layer followed by a softmax layer for prediction trained using cross-entropy objective to predict the main task label. The main task classifier is trained for 1 epoch for the real-world datasets and 10 epochs for the Synthetic-Text dataset for the setting when we train the main task classifier in INLP removal phase. The encoder is frozen for both the setting (with or without main task classifier training) though-out the INLP removal phase.

The main-task classifier and encoder in the pretraining phase and the probing classifier in the removal phase is trained using cross-entropy loss for both real and synthetic datasets. For the real dataset, a fixed learning rate of $1 \times 10^{-5}$ is used when RoBERTa is used as encoder and $5 \times 10^{-5}$ when using BERT as encoder. For synthetic experiments, a fixed learning rate of $5 \times 10^{-3}$ is used when training both the nBOW encoder and main-task classifier in the pretraining stage and probing classifier in removal stage.

## E.7  Adversarial Removal Experiment Setup

For adversarial removal (AR) experiment, for both real and synthetic datasets, first the input text is encoded to latent representation using the encoder (§E.4 and E.5). Then for the main-task classifier, a linear transformation layer followed by a softmax layer is applied for the main-task prediction. The same latent representation output from the encoder is given to the probing classifier which is a separate linear transformation layer followed by a softmax layer. All components of the model, encoder, main-task classifier, and probing classifier are trained using the following modified objective from Eq. 95:

$$arg \min_{h, c_m, c_p} \left\{ L(c_m(h(\boldsymbol{z})), y_m) + \lambda L(c_p(g_{-1}(h(\boldsymbol{z}))), y_p) \right\}$$

where $h$ is the encoder, $c_m$ is the main task classifier, $c_p$ is the probing classifier, $g_{-1}$ is the gradient reversal layer with fixed reversal strength of $-1$. The first term in the objective is for training the main task classifier and the second term is the adversarial objective for training the probing classifier using gradient reversal method [15, 12] . The hyperparameter $\lambda$ controls the strength of the adversarial objective. In our experiment we very $\lambda \in \{0.00001, 0.0001, 0.001, 0.01, 0.1, 0.5, 1.0, 2.0\}$. When describing the experimental results in §F.3 we choose the $\lambda$ which performs the best for all datasets with different predictive correlation $\kappa$ in removing the undesired concept from the latent representation.

## E.8  Metrics Description

Analogous to spuriousness score (Def 3.1) for main-task classifier we define the score for probing classifier below.

**Definition E.1** (Probe Spuriousness Score). *Given a dataset, $\mathcal{D}_{m,p} = S_{min} \cup S_{maj}$ with binary task label and binary concept, let $Acc^f(S_{min})$ be the minority group accuracy of a given probing classifier ($f$) and $Acc^*(S_{min})$ be the minority group accuracy of a* clean *probing classifier that does not use the main-task feature. Then spuriousness score of f is: $\psi(f) = |1 - Acc^f(S_{min})/Acc^*(S_{min})|$.*

For simplicity, in all our experiments we assume that both the main and the correlated attribute labels are binary. We measure the degree of spuriousness using the following two metrics:

1. Spuriousness Score: As defined in §3.4, this metric help us quantify, how much a classifier is using the spurious feature (see Def 3.1 and E.1).

2. $\Delta$ Probability: In Synthetic-Text dataset as described in E.1, we have the ability to change the input corresponding to the change in concept label (thus implementing Def 2.1) . Thus we could measure if the main-classifier is using the spurious-concept by changing the concept in the input and measuring the corresponding change in the main-task classifier's prediction probability. The Higher the change in prediction probability higher the main-task classifier is dependent on spurious-concept.

### E.9 Compute and Resources

We used an internal cluster of Nvidia P40 and P100 GPUs for all our experiment. Each experiment setting was run on three random seed and mean results with variance are reported in all the experiment.

## F Additional Results

### F.1 Probing classifier Quality

Fig. 5 shows different failure modes of the probing classifier. In Fig. 5a and 5b, a *clean* main-task classifier which doesn't use the concept feature is trained on Synthetic-Text and MultiNLI dataset respectively using the method described in §E.2. Thus the latent representation doesn't have the concept feature. Then, to test the presence of concept-causal feature in the latent representation we train a probing classifier to predict concept-label. The first row show the accuracy of the probing classifier for testing the presence of concept in latent space. When $\kappa = 0.5$ i.e no correlation between the main-task and the concept label, the probing accuracy is approximately 50% which correctly shows the absence of the concept-causal feature in the latent representation. The accuracy increases as the correlation $\kappa$ between the main and concept-causal feature increases in dataset. This shows that even when concept-causal feature is not present in the latent representation, probing classifier will still claim presence of concept-causal feature if any correlated feature (main-task feature in this case) is present in the latent space. In Fig. 5c, the latent space contains the concept-causal feature as shown by accuracy of approximately 94.5% when $\kappa = 0.5$. But as $\kappa$ increases the probing classifier's accuracy increases in the presence of correlated main-task feature which falsely increases the confidence of presence of the concept-causal feature. The second row shows the spuriousness-score of concept-probing classifier is increasing as the correlation between the main-task and concept-causal feature increases which implies that the probing classifier is using relatively large *amount* of correlated main-task feature for concept-label prediction in all settings.

For all the experiments in this section (§F.1) with Synthetic-Text dataset, we didn't introduce any noise in the probing label (i.e. n=0.0) and have 1 hidden layer when training the encoder (see §E.5 for details). For the experiment on MultiNLI dataset, we use RoBERTa as the default encoder and rest of setup is same as described in §E.4.

### F.2 Extended Null-Space Removal Results

Fig. 6 and 7, shows the failure mode of null-space removal (INLP) in the real dataset when using RoBERTa and BERT as encoders respectively. Different columns of the figure are for three different real datasets — MultiNLI, Twitter-PAN16, and Twitter-AAE respectively. The x-axis from steps 8-26 is different INLP removal steps. The y-axis shows different metrics to evaluate the main task and probing classifier. Different colored lines show the spurious correlation ($\kappa$) in the probing dataset used by INLP for the removal of spurious-concept. The pretrained classifier is clean, i.e., does not use the spurious concept-causal feature; hence INLP shouldn't have any effect on main-classifier when removing concept-causal feature from the latent space. The first row shows that as the INLP iteration progresses, the norm of latent representation, which is being *cleaned* of concept-causal feature, decreases. This indicates that some features are being removed. However, the results are against our expectation from the second statement of Theorem 3.2, which states that the norm of the classifier will tend to zero as the INLP removal progresses. The possible reason is that from Theorem 3.2 the norm of latent representation will go zero when the latent representation only contains the spurious concept-causal feature and the other features correlated to it. But, the encoder representation could have other features which are not correlated with concept-label and hence not

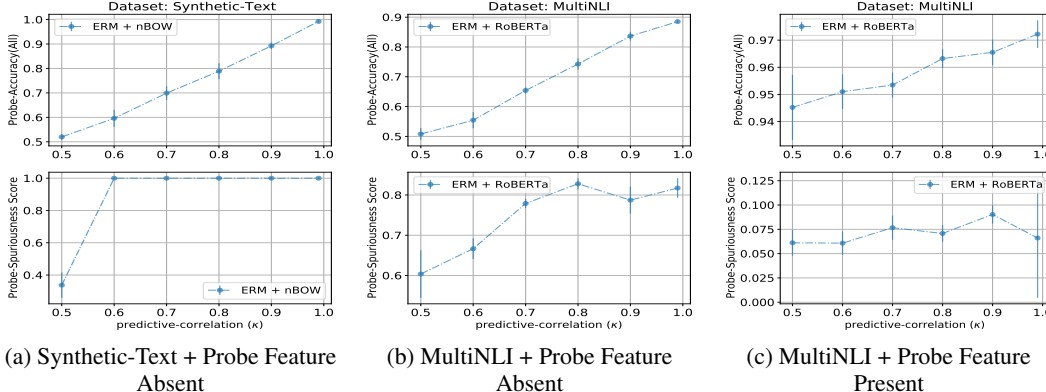

(a) Synthetic-Text + Probe Feature Absent

(b) MultiNLI + Probe Feature Absent

(c) MultiNLI + Probe Feature Present

Figure 5: **Failure Modes of Probing classifier:** The first row in Fig. 5a and 5b shows that even when the latent representation doesn't contain the probing concept-causal feature, the probing classifier is still has >50% accuracy when other correlated feature is present. The accuracy increases as the correlation $\kappa$ between the probing concept-causal feature and other correlated features increases. The first row Fig. 5c shows that presence of correlated features could increase the probing classifier's accuracy thus increasing the confidence in the presence of concept-causal feature in latent representation. The second row of all the figures shows that the probing classifier is getting more spurious as the $\kappa$ increases thus implying that the probing classifier is using some other correlated feature than concept-causal feature. For more discussion see §F.1.

removed. Since, the pretrained classifier given for INLP was *clean* (using method described in §E.2), we do not expect the INLP to have any effect on the main-task classifier.

The second row in Fig. 6 and 7 shows that the main classifier accuracy drops to random guess i.e 50% except for the case when probing dataset have $\kappa = 0.5$ i.e no correlation between the main and concept label. Thus INLP method corrupted a clean classifier and made it useless. The reason behind this could be observed from the fourth and fifth rows. The fourth row shows the accuracy of the probing classifier before the projection step. We can see that at step 8 on the x-axis $\kappa = 0.5$, the probing classifier correctly has an accuracy of 50% showing that the concept-causal feature is not present in the latent representation. But for other values of $\kappa$, the probing classifier accuracy is proportional to the value of $\kappa$ implying that the probing classifier is using the main-task feature for its prediction. Hence at the time of removal, it removes the main-task feature which leads drop in the main-task accuracy. This can also be verified from the last row of Fig. 6 and 7, which shows that the spuriousness score of probing classifier is high; thus it is using the main-task feature for its prediction. We observe similar results for Synthetic-Text dataset when using INLP in Fig. 8. For all the INLP experiment on Synthetic-Text dataset, there were no hidden layers after the nBOW encoder (see §E.5).

So far, we have kept the main-task classifier frozen when performing INLP removal. Note that, we also experiment with the setting when the main task classifier is trained after every projection step of INLP (see §E.6 for experimental setup and Fig. 9 for a result description). We observe a similar drop in the main-task accuracy with prolonged removal using INLP and early stopping leads to an even higher reliance on the spurious concept-causal feature than it had at the beginning of INLP. The rest of the experimental configurations were kept the same as the other INLP experiments described above.

### F.3 Extended Adversarial Removal Results

**Adversarial removal failure in real-world datasets.** Fig. 10 shows the failure mode of adversarial removal AR on real-world datasets. In the x-axis we vary the predictive correlation $\kappa$ between the main and the concept-label in different datasets and measure the performance of AR on different metrics on the y-axis. The second row shows the spuriousness score of the main-task classifier after AR as we vary $\kappa$ on the x-axis. When using RoBERTa as the encoder, the orange curve in second row shows the spuriousness score of the main-task classifier when trained using the ERM loss. The spuriousness score describes how much unwanted concept-causal feature the main-task classifier is using. The blue curve shows that the AR method reduces the spuriousness of main-task though

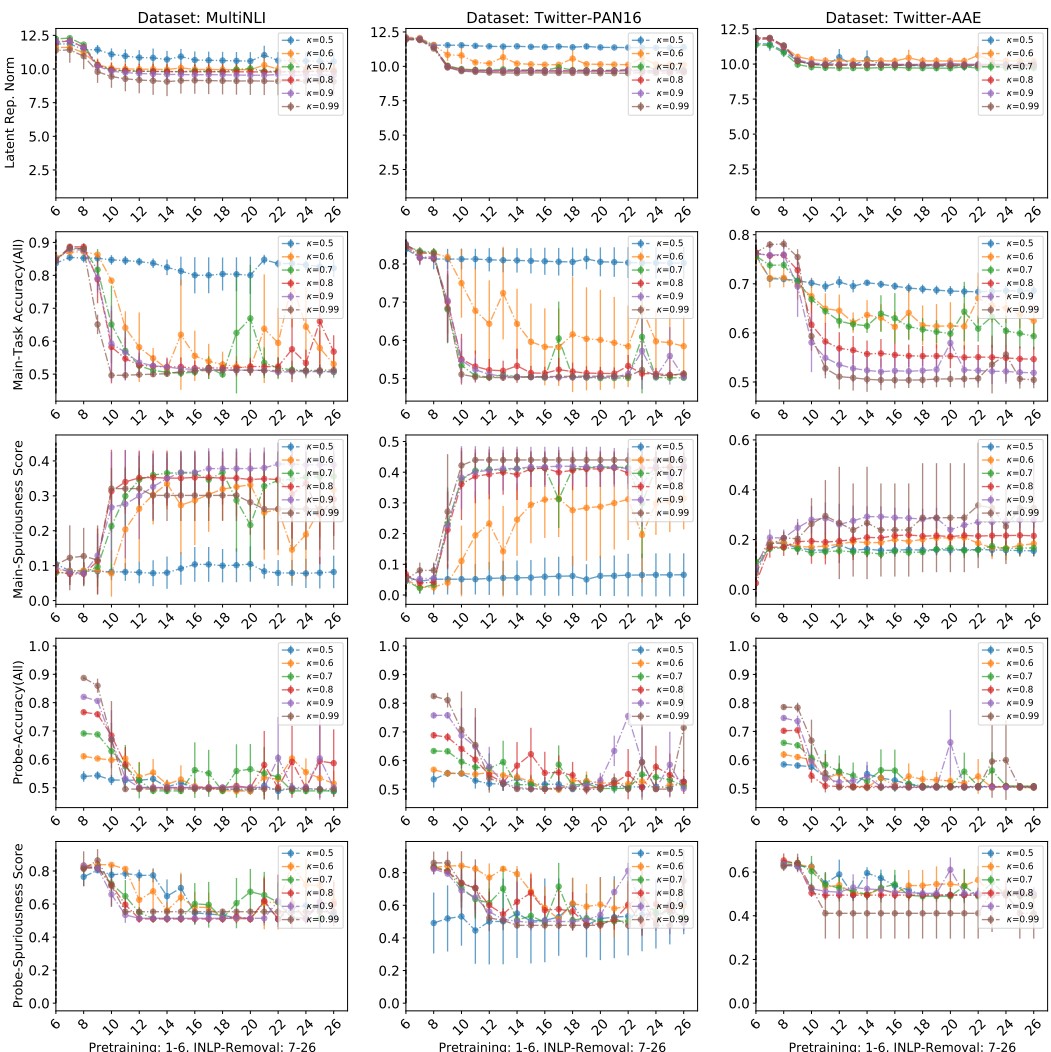

Figure 6: **Failure of Null Space Removal when using RoBERTa as encoder:** Different columns of the figure are for three different real datasets — MultiNLI, Twitter-PAN16, and Twitter-AAE respectively. The x-axis from steps 8-26 is different INLP removal steps. The y-axis shows different metrics to evaluate the main task and probing classifier. Different colored lines show the spurious correlation ($\kappa$) in the probing dataset used by INLP for removal of spurious-concept. The pretrained classifier is clean i.e. doesn't use the spurious concept-causal feature, hence INLP shouldn't have any effect on main classifier when removing concept-causal feature from the latent space. Against our expectation, the second row shows that the main-task classifier's accuracy is decreasing even when it is not using the concept-feature. The main reason for this failure to learn a clean concept-probing classifier. This can be verified from the last row which shows that the concept-probing classifier has a high spuriousness score thus implying that it is using the main-task feature for concept label prediction and hence during the removal step, wrongly removing the main-task feature which leads to a drop in main-task accuracy. For more discussion see §F.2.

cannot completely remove it. The reason for this failure can be attributed to probing classifier. Even when AR has successfully removed the unwanted concept feature, the accuracy of concept-probing classifier will be proportion to $\kappa$ due to presence of correlated main-task feature in the latent space. This can be seen in the third row of Fig. 10. Thus we cannot be sure if the unwanted concept-causal feature has been completely removed from the latent space or just became noisy enough to have accuracy proportional to $\kappa$ after AR converges. In Fig. 10, for each dataset and encoder, we manually choose the hyperparameter described $\lambda$ described in §E.7 which reduces the spuriousness score most for the main-task classifier while not hampering the main-task classifier accuracy. In Fig. 11, we

show the trend in spuriousness score is similar for all choices of hyperparameter $\lambda$ in our search. No value of $\lambda$ is able to completely reduce the spuriousness score to zero.

**Adversarial removal makes a classifier clean.**    Fig. 12 shows that when the adversarial classifier is initialized with a clean main-task classifier that doesn't use unwanted-concept causal features, it makes matters worse by making the main-task classifier use the unwanted-concept feature. For the Synthetic-Text dataset, since the word embeddings are non-trainable, one single hidden layer is applied after the nBOW encoder so that AR methods could remove the unwanted-concept feature from the new latent representation. We create a *clean* Synthetic-Text dataset by training a classifier (iteration 1-20) on dataset with predictive correlation $\kappa = 0.5$ between the main-task and concept label. $\kappa = 0.5$ which implies there is no correlation between the labels thus we can expect the main-task classifier to not use the concept-causal feature. This can be seen from Main classifier spuriousness score in Fig. 12a (2nd row) which is close to 0. We chose this method to create a clean classifier since this allows us to measure the spuriousness score for the main-task classifier. If we would have followed method described in [35], then we would have had only a single value of concept label ($y_p$) in the dataset and couldn't have defined the majority and minority group required for calculation of spuriousness score (see Def 3.1). For all our experiments on Synthetic-Text dataset we use noise =0.3 and trained the main-task and probing classifier with 1 hidden layer. Similarly for training a clean classifier for MultiNLI dataset (iteration 1-6) we again use a dataset with predictive correlation $\kappa = 0.5$. Post training the clean classifier the AR method is initialized with these clean classifiers for removal of concept-causal features. Since AR is initialized with clean classifier which doesn't use concept-causal feature, we expect AR to have no effect on the classifier. In contrast we observe that the spuriousness score of main-classifier for both Synthetic-Text and MultiNLI dataset increases (2nd row in Fig. 12a and 12b) which shows that AR when initialized *clean*/fair classifier could make them unclean/unfair.

### F.4    Synthetic-Text dataset Ablations

**Adversarial Removal Failure in Synthetic-Text dataset:**    Figure 13 shows the failure of AR on the synthetic dataset as we vary the noise in the main-task label and unwanted concept-label. For the experiment, since the word embeddings are non-trainable, one single hidden layer is applied after the nBOW encoder so that AR methods could remove the unwanted-concept feature from the new latent representation.

**Dropout Regularization Helps AR method:**    Continuing on observation from Fig. 14a, 14b and 14c shows the $\Delta$-Prob of the main-task classifier after we apply the AR on Synthetic-Text dataset (with noise=0.3) and how they changes as we increase the dropout regularization. As we increase the dropout (drate in the figure), the $\Delta$-Prob of the main classifier decreases showing that the regularization methods could help improve the removal methods.

## G    Comparison between Spuriousness Score and $\Delta$Prob

In this section we compare the Spuriousness Score proposed in §3.4 for measuring a classifier's use of a binary spurious feature with the ideal, ground-truth metric, $\Delta$Probability ($\Delta$Prob for short) defined in §E.8. $\Delta$Prob measures the reliance on a spurious feature by changing the spurious feature in the input space (when possible) and measuring the change in the prediction probability of the given classifier. Hence $\Delta$Prob is a direct and intuitive measure of spuriousness in a given classifier. But changing the spurious feature is difficult in the input space for real-world data, thus we only evaluate this metric on the Synthetic-Text dataset.

To do so, we use the result from Fig. 13 that showed failure of the adversarial removal method on the Synthetic-Text dataset under various noise settings (refer §F.4 for details). For the setting with noise $n = 0.0$, both Spuriousness Score and $\Delta$Prob curve for Adversarial Removal (marked as ADV in Fig. 13) are identical (close to 0 for all values of $\kappa$ with mean $= 0.0$ and standard-deviation $= 0.0$). For the other settings with non-zero noise, we compute the Pearson correlation between the Spuriousness score and $\Delta$Prob for the ADV curve. As Table 1 shows, we observe high Pearson correlation of 0.83 and 0.95 for the noise setting, $n = 0.1$ and $n = 0.3$ respectively. The third column in the table shows p-value ($< 0.05$) assuming a null hypothesis that the two metrics are uncorrelated.

Table 1: **Correlation between Spuriousness Score and ΔProb on Synthetic-Text dataset:** Pearson-correlation between Spuriousness score and ΔProb; the two metrics for quantifying the dependence of a classifier on a spurious feature. We measure the correlation for adversarial-removal experiment over two different noise setting on Synthetic-Text dataset. For more details, see §F.4. The first column shows different experimental settings and the second column shows the Pearson correlation between the two metrics. The third column shows the p-value under the null hypothesis that the two metrics are uncorrelated. Both correlations are statistically significant since p-value for both the case is < 0.05.

|                      | Pearson Correlation | p-value |
| -------------------- | ------------------- | ------- |
| Synthetic-Text + n=0.1 | 0.83              | 0.0403  |
| Synthetic-Text + n=0.3 | 0.95              | 0.0033  |

These results suggest that Spuriousness-Score can be a good approximation for the ideal ΔProb metric.

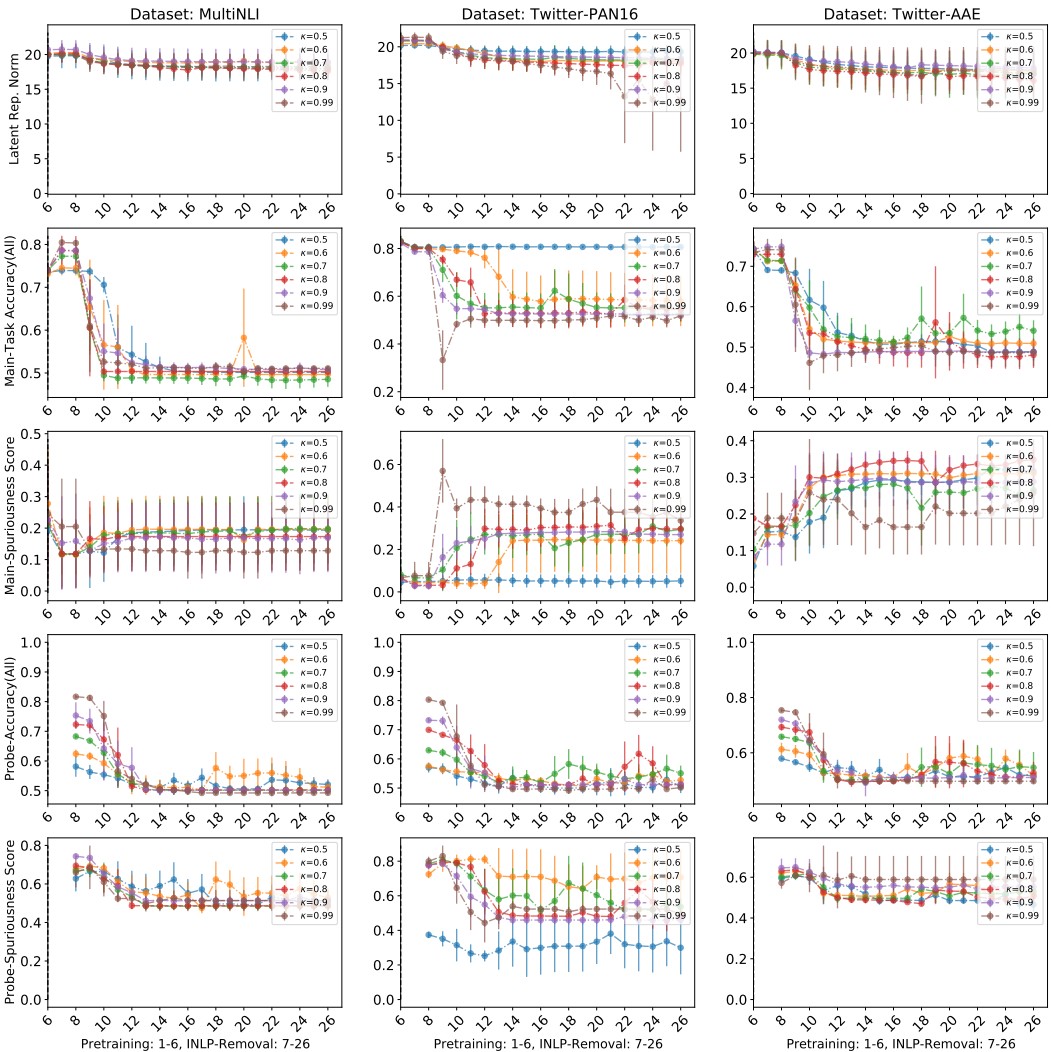

Figure 7: **Failure of Null Space Removal when using BERT as encoder:** The observation is similar to the case when RoBERTa was used as encoder (see Fig. 6) . Different columns of the figure are for three different real datasets — MultiNLI, Twitter-PAN16, and Twitter-AAE respectively. The x-axis from steps 8-26 is different INLP removal steps. The y-axis shows different metrics to evaluate the main task and probing classifier. Different colored lines show the spurious correlation ($\kappa$) in the probing dataset used by INLP for removal of spurious-concept. The pretrained classifier is clean i.e. doesn't use the spurious concept-causal feature, hence INLP shouldn't have any effect on main-classifier when removing concept-causal feature from the latent space. Against our expectation, the second row shows that the main-task classifier's accuracy is decreasing even when it is not using the concept-feature. The main reason for this failure to learn a clean concept-probing classifier. This can be verified from the last row which shows that the concept-probing classifier has high spuriousness score thus implying that it is using the main-task feature for concept label prediction and hence during the removal step, wrongly removing the main-task feature which leads to a drop in main-task accuracy. For more discussion see §F.2.

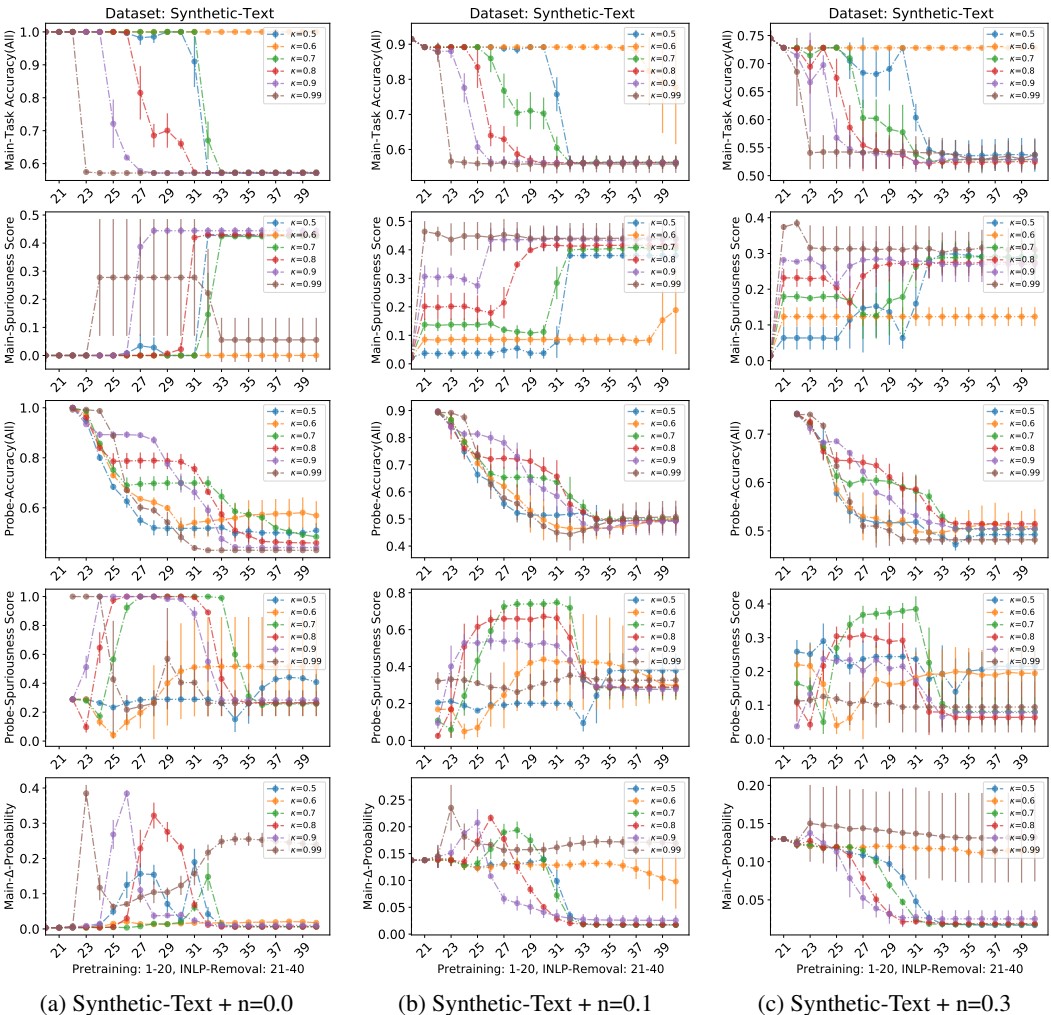

(a) Synthetic-Text + n=0.0     (b) Synthetic-Text + n=0.1     (c) Synthetic-Text + n=0.3

Figure 8: **Failure Mode of INLP in Synthetic-Text dataset**: Different columns of the figure are Synthetic-Text dataset with different levels of noise in the main task and probing task label. Here, n=0.0 means there is 0% noise and n=0.3 means there is 30% noise in the labels. The x-axis from steps 22-40 is different INLP removal steps. The y-axis shows different metrics to evaluate the main task and probing classifier. Different colored lines show the spurious correlation ($\kappa$) in the probing dataset used by INLP for the removal of spurious-concept. The pretrained classifier is clean i.e. doesn't use the spurious concept-causal feature, hence INLP shouldn't have any effect on main classifier when removing concept-causal feature from the latent space. Contrary to our expectation, the first row shows main-task classifier accuracy drops as the INLP progresses. Higher the correlation between the main-task and concept label, faster the drop in the main task accuracy. The last row shows the change in prediction probability ($\Delta$-Prob) of main-task classifier when we change the input corresponding to concept-label. This shows, how much sensitive the main task classifier is wrt. to concept feature. We observe that the $\Delta$-Prob increases in the middle of INLP showing that the main-classifier which was not using the concept initially (as in iteration 21), started using the sensitive concept because of INLP removal. Thus stopping INLP prematurely could lead to a more *unclean* classifier than before whereas running INLP longer removes all the correlated features and could make the classifier useless. For more discussion see §F.2.

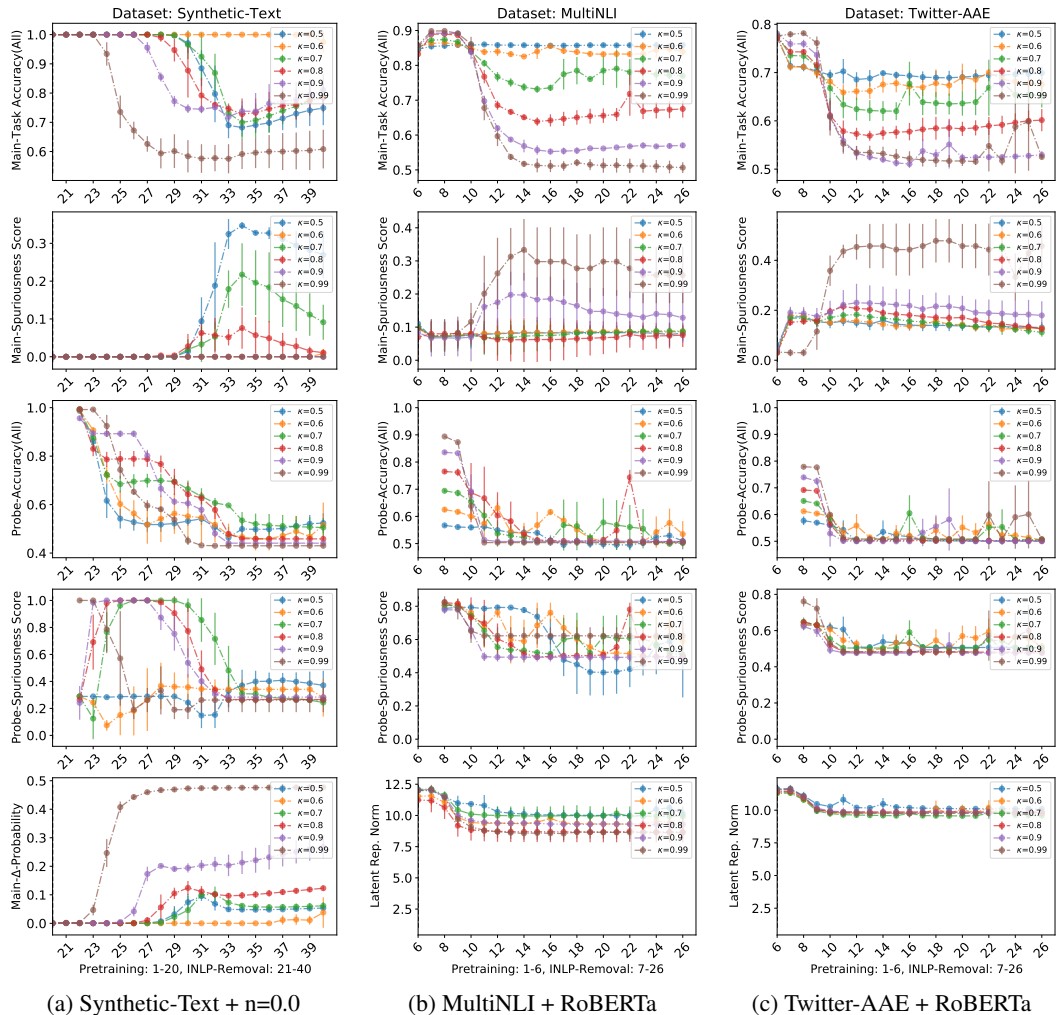

(a) Synthetic-Text + n=0.0

(b) MultiNLI + RoBERTa

(c) Twitter-AAE + RoBERTa

Figure 9: **Failure Mode of INLP + Main Task Classifier head retraining**: Given a pretrained encoder and the main task classifier as input to INLP for spurious concept removal, in this experiment, we retrain the main task classifier after every step of null-space projection by INLP. All the other experiment configurations for these experiments are kept the same as the case when we don't retrain the main-task classifier. The first, second, and third columns show the results for Synthetic-Text, MultiNLI, and Twitter-AAE datasets respectively. We observe a similar trend as the case when the main task classifier was not trained after each projection step (see Fig. 6, 7 and 8). The main task classifier's accuracy drops as the null-space removal proceeds (iteration 21-40 for Synthetic-Text and iteration 7-26 for MultiNLI and Twitter-AAE datasets). Though the drop is not as severe as in the previous setting (when we didn't train the main task classifier), it is significant enough to impact the practical utility of the model (greater than 20% drop in the accuracy when $\kappa > 0.8$ for all the datasets above). Similar to previous setting, early-stopping of INLP removal may lead to a classifier that has a higher reliance on the spurious concept than it had before the INLP removal. For example, for $\kappa = 0.8$ in Synthetic-Text dataset, the main-task classifier's performance drops for the first time at iteration 29 (a valid heuristic for early stopping), but it has high $\Delta$Prob $\approx 10\%$ as shown in the last row of the Synthetic-Text dataset column of this figure. For discussion of the case when the main task classifier is not trained after every projection step, see §F.2 and §4.2.

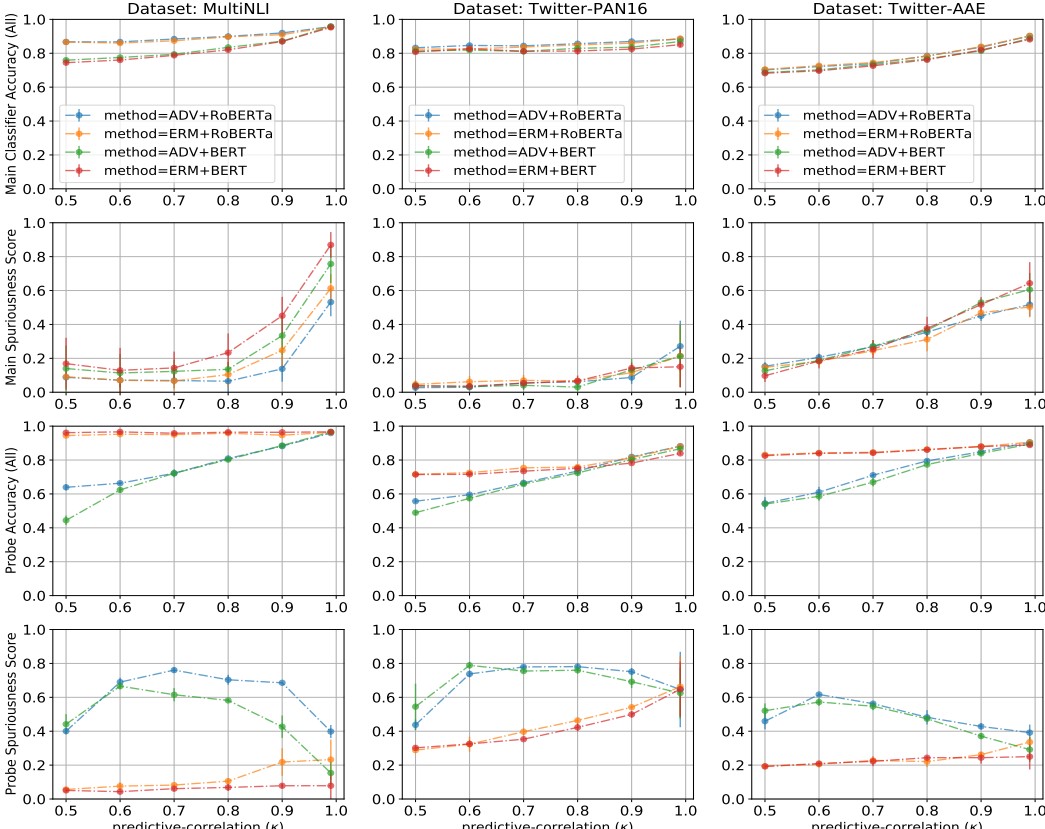

Figure 10: **Failure Mode of Adversarial removal on real-dataset:** Different column shows the result on three different real datasets —MultiNLI, Twitter-PAN16, and Twitter-AAE respectively. The second row shows the accuracy of spuriousness score of the main-task classifier after AR when the dataset contains different levels of spurious correlation between the main-task and unwanted-concept label, denoted by $\kappa$ in the x-axis. When using RoBERTa as the encoder, the orange curve in second row shows the spuriousness score of the main-task classifier when trained using the ERM loss. The spuriousness score describes how much unwanted concept-causal feature the main task classifier is using. The blue curve shows that the AR method reduces the spuriousness of main-task though cannot completely remove it. When using BERT as encoder, the observation is same i.e green curve in second row shows AR is able to reduce the spuriousness of main classifier than the red curve which is trained using ERM, but is not able to completely remove the spurious feature. For more discussion see §F.3.

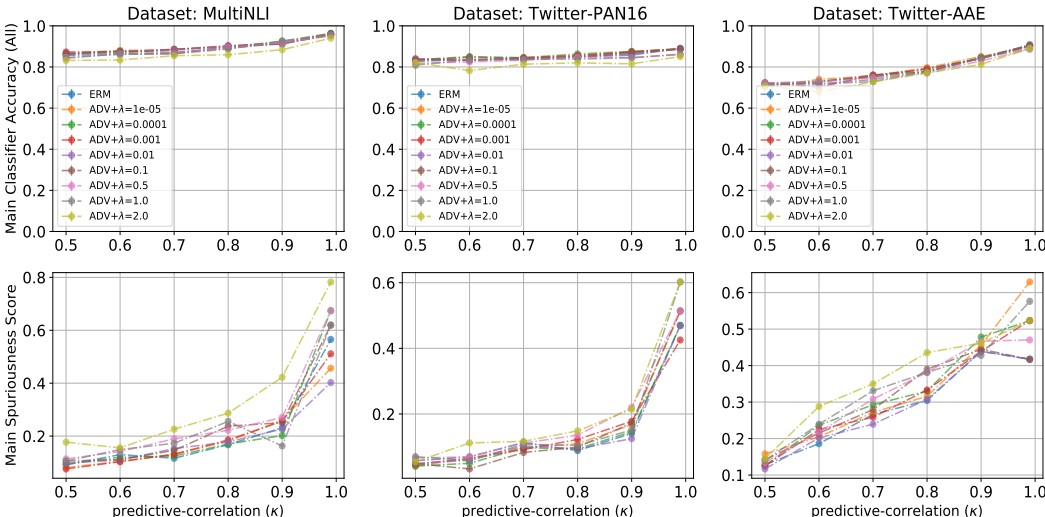

Figure 11: **Choice of Adversarial Strength Parameter** $\lambda$: The second plot shows that trend in spuriousness score after AR is similar for all the choices of hyperparameter $\lambda$ we have taken in our search. None of the settings of $\lambda$ is able to completely reduce the spuriousness score to zero. For more discussion see §F.3.

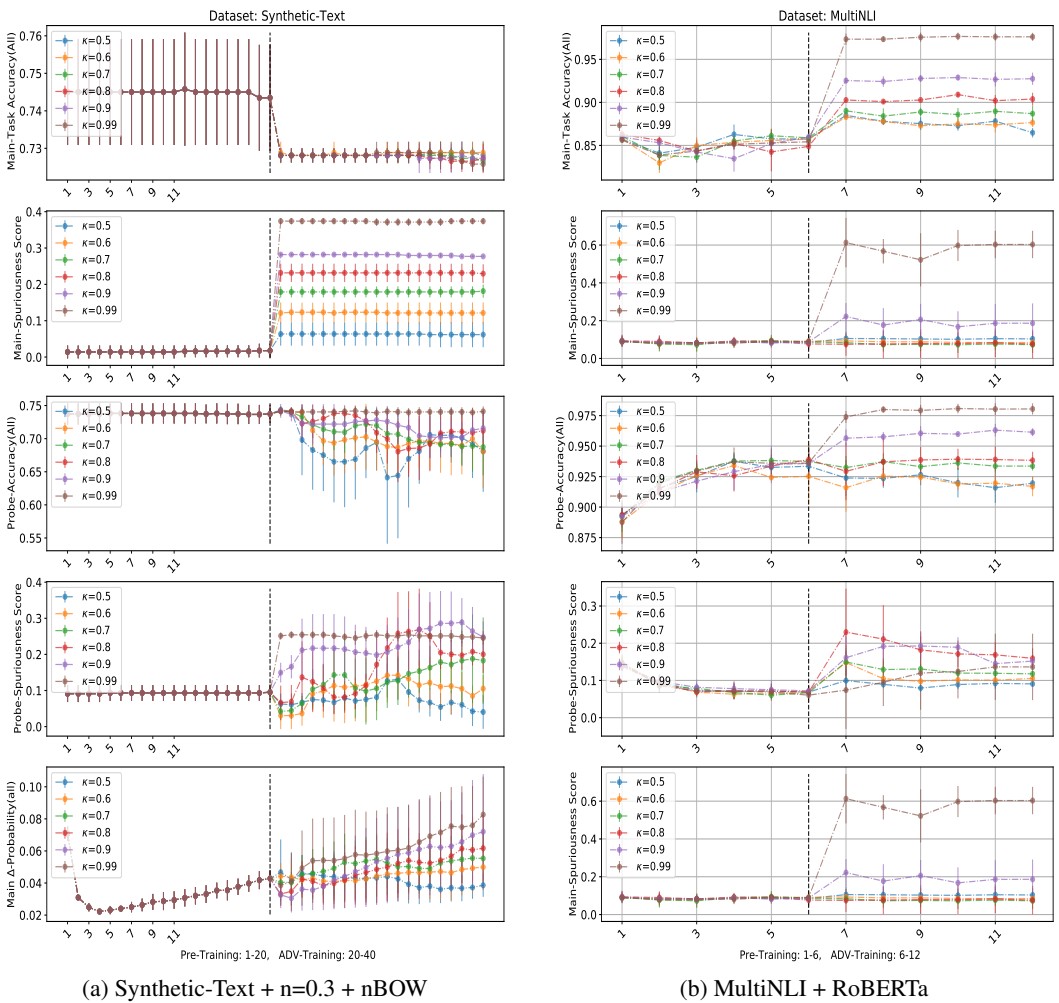

(a) Synthetic-Text + n=0.3 + nBOW          (b) MultiNLI + RoBERTa

Figure 12: **Adversarial Removal Makes a classifier unclean**: We test if the AR method increases the spuriousness of a main-task classifier if initialized with a *clean* classifier. In 12a, from iteration 1-20 in x-axis, a clean classifier is trained on Synthetic-Text dataset (with 30% noise i.e n=0.3 in main-task and probing labels) such that it doesn't uses the unwanted concept-causal feature by training on a dataset with $\kappa = 0.5$ (see §F.3 for details). Then the classifier is given to AR method for removing the unwanted concept feature which makes the initially clean classifier unclean. This can be seen from the second row of the 12a which shows the spuriousness score of main-classifier is $0$ during 1-20 iteration but increases after the AR start from 21-40. Also, the last row shows the $\Delta$-Prob of the main-task classifier on changing the unwanted-concept in input which increases for datasets which have large $\kappa$ i.e correlation between the main and concept label. A similar result can be seen for the MultiNLI dataset where a clean classifier is trained in iterations 1-6 (using a dataset with $\kappa = 0.5$) which is made unclean by AR. Second row again shows that spuriousness score of main-task classifier increases after AR starts in iteration 7-12. For more discussion see §F.3.

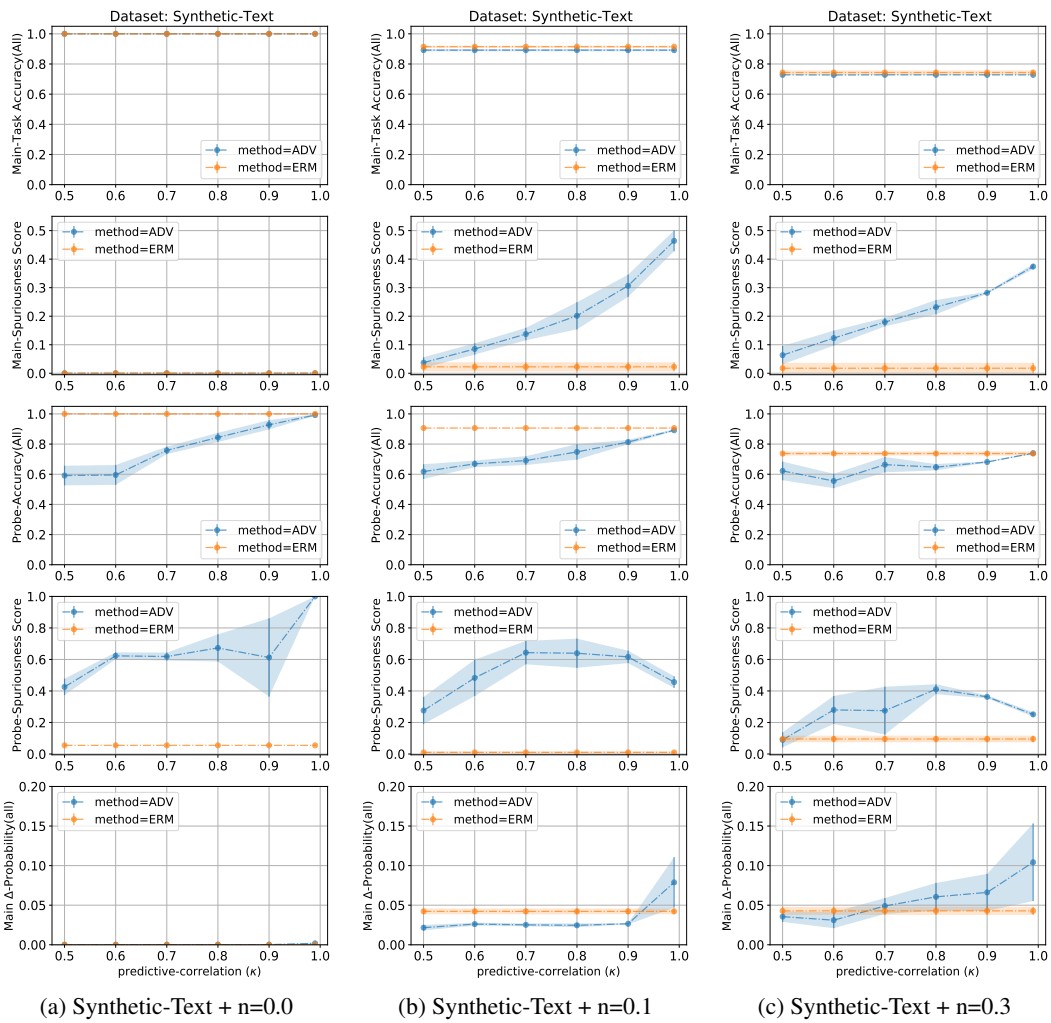

(a) Synthetic-Text + n=0.0      (b) Synthetic-Text + n=0.1      (c) Synthetic-Text + n=0.3

Figure 13: **Failure of Adversarial Removal method on Synthetic-Text dataset:** Different columns show the adversarial removal method on Synthetic-Text dataset with different levels of noise in the main-task and concept label. When there is no noise, from the second row in Fig. 13a, we see that both the classifier trained by ERM and AR has zero-spuriousness score. But as we increase the noise to 10% in Fig. 13b, we observe that the spuriousness score increases when AR is applied in contrast to classifier trained by ERM which stays at 0. Also, higher the predictive correlation $\kappa$, higher the increase in spuriousness. This observation augments the observation in Fig. 12 which shows that using AR makes a clean classifier unclean. Similarly in Fig. 13c when we increase the noise to 30% we observe in second row, AR is increased the spuriousness, unlike ERM which is at 0. For discussion see §F.4

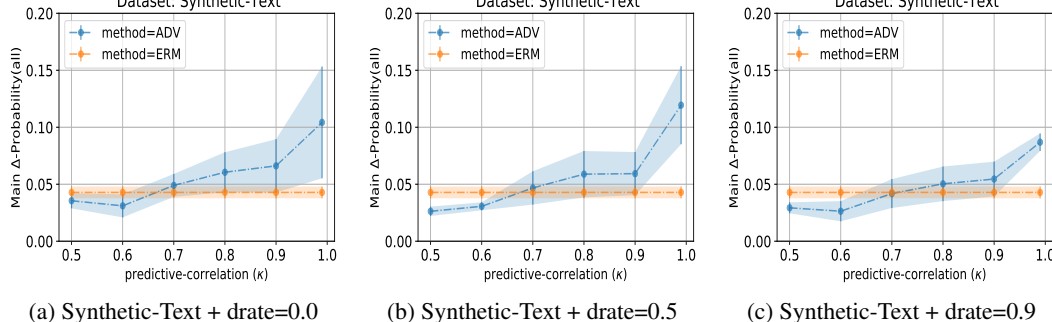

(a) Synthetic-Text + drate=0.0    (b) Synthetic-Text + drate=0.5    (c) Synthetic-Text + drate=0.9

Figure 14: **Dropout Regularization helps in Adversarial Removal:** $\Delta$-Prob of the main-task classifier after we apply the AR on Synthetic-Text dataset (with noise=0.3) decreases as we increase the dropout regularization from 0.0 to 0.9. For discussion see §F.4.