# OpenReview forum: "Probing Classifiers are Unreliable for Concept Removal and Detection"
_NeurIPS.cc/2022/Conference — NeurIPS 2022 Accept_

### Official Review · Reviewer_PTq2 · 2022-07-06

**Rating:** 5
**Confidence:** 4
**Soundness:** 2 fair
**Presentation:** 3 good
**Contribution:** 3 good

**Summary:**

The authors propose that probing-based concept removal and detection methods are unreliable given that the representation space may be corrupted, or the resulting model may still reply to those seemingly removed concepts. The authors try to ground their hypothesis on theoretical based on toy examples in 2D space, and then provide pieces of evidence using a set of naturalistic and synthetic benchmarks where the correlations between the concept and the sequence label are controllable. Although the topic is interesting and the research question is worth pursuing further, I found a set of opening questions that remain unresolved in the paper.

**Questions:**

- L146, says ``w.r.t. task label y_p (Assumption 3.3)``, should it be the concept label instead of the task label?

- Fig 1. (a) is confusing, if not wrong. For instance, for the Gender Probing Classifier (annotated as the blue dashed line), with accuracy = 100%, it is clearly mislabeling multiple examples.

- I would suggest moving at least the essence of the *Proof Sketch* to the main text as it is essential to see formal proofs to back up the authors' main hypotheses. Otherwise, it feels ungrounded.

- In Sec. 4.2, the authors train the model using a special subset of training data where all examples are ``with the same value of the property`` (L305), could you clarify a bit what is the property? and how to have the same value?

- Can authors define ERM before using it as a term in L338?

**Ethics Review Area:**

["I don’t know"]

**Limitations:**

The authors adequately addressed the limitations and potential negative societal impact

**Strengths And Weaknesses:**

**Strengths**:
- This paper aims to address a pressing issue in building debiasing methods of neural networks.
- The toy examples provided in the paper are intriguing.

**Weaknesses**:
- **Is the failure mode of INLP really a valid failure mode?**

The main argument for disqualifying INLP from being a sound method is that INLP will corrupt the representation even though we know there are no concept-level relations with respect to the final label in the task. In Sec. 4.2, the authors train the model using a special subset of training data where all examples are ``with the same value of the property`` (L305). I assume the authors only allow the model to see training examples with the same concept label y_{p}. With this, the authors claim that INLP should have a null effect in removing information in the representation space. I find this argument unconvincing, if not wrong. First of all, INLP in this case will first train a probe classifier with simply 100% accuracy, since it only needs to predict the same label for all examples. For the iterative project process, it needs to remove information to decrease the accuracy from 100% to 0% (if possible). It is obvious that INLP will definitely remove sequence-related information from the embedding no matter what, which eventually will just randomly corrupt the representations. As a result, the results in Fig. 2 are indeed expected. These results are not because INLP is effective, the experiment setups naturally lead to these results.

- **Is the failure mode of the adversarial removal methods really a valid failure mode?**

All experiments conducted in Sec. 4.3 relies on the predictive correlation (κ). Let us say we have κ=1.0. As we are training a probing classifier to have an adversarial loss to remove concept-related information, there is no way to guarantee this will happen, especially in this case where the main-task label is 100% correlated with the concept label. The probing classifier may just be another main-task classifier that is trained to lower the main-task accuracy. In this case, it is not actually a failure mode about the method again, it is a tricky design of the dataset that fools the method. In fact, when the main-task label is 100% correlated with the concept label, there is no way we can ensure we are removing concept-related information if only with classification-driven losses.

I found the topic interesting, but my main concern is those two real-world study cases are not really exposing weaknesses of those concept removal methods, and are rather just some corrupted datasets that will trick any method.

[Update]: Thanks for the clarification on the INLP, as the description was a little confusing to me. Maybe considering updating the text. In the case study of the adversarial removal methods, I am still not convinced yet that building spurious correlations within the dataset is a good way to say the failure mode of these methods. I am raising my score to 5 as I am leaning towards acceptance.

---

> ### Author Response · Authors · 2022-08-02
> **Official Response to Reviewer PTq2 (Part 2/2)**
>
> >**Q3:** L146, says w.r.t. task label y_p (Assumption 3.3), should it be the concept label instead of the task label?
>
> **A3:** Thanks for pointing it out. Yes, it’s a typo and we have made the changes in L148 of the revised version (annotated with blue color).
>
>
>
>
>
>
> >**Q4:** Fig 1. (a) is confusing, if not wrong. For instance, for the Gender Probing Classifier (annotated as the blue dashed line), with accuracy = 100%, it is clearly mislabeling multiple examples.
>
> **A4:** We apologize for the confusion. The gender labels are denoted by the shape of the points (triangle represents Gender1 and circle represents Gender2). As mentioned in L174 of revised version of paper, we call the *projection direction* of the gender classifier as the “Gender Probing Classifier” following the convention from the INLP paper (Ravfogel et. al. [27]). Thus, the decision boundary is orthogonal to the Gender Classifier in Fig 1(a). Specifically, in Fig 1(a), the slanted colored line shows the projection direction of the Gender Classifier, and the Decision boundary of the Gender Classifier is shown by the black slanted line. On one side of the decision boundary both the points are triangles (Gender1), and on the other side both the points are circles (Gender2). Note that the points on the y-axis are *projections* of the actual inputs on the  Profession classifier and thus should not be considered. Hence, this slanted probing classifier achieves 100% accuracy in classifying gender.
>
> For greater interpretability, we add a new Appendix J where we break down Fig. 1a and show the two classifiers separately. Fig. 13a shows the profession classifier and its projected points on the classifier line. Fig 13b. shows the slanted gender classifier, the input points, and its projections on the gender classifier.
>
>
>
>
>
> >**Q5:** I  would suggest moving at least the essence of the Proof Sketch to the main text as it is essential to see formal proofs to back up the authors' main hypotheses. Otherwise, it feels ungrounded.
>
> **A5:** That’s a great suggestion. We will incorporate this change in the camera-ready version (utilizing the extra page).
>
>
>
> >**Q6:** In Sec. 4.2, the authors train the model using a special subset of training data where all examples are with the same value of the property (L305), could you clarify a bit what is the property? and how to have the same value?
>
> **A6:** We apologize for the confusion. To train a “clean” main-task classifier that doesn't use the spurious feature, we subsample the dataset with one particular value of spurious concept label (mentioned as property), as done by Ravichander et. al. [30]. We have made the changes in L305 of the revised version (annotated with blue color).
>
>
> >**Q7:** Can authors define ERM before using it as a term in L338?
>
> **A7:** Thanks for pointing it out. We have made the changes in L340 of revised version (annotated with blue color).

---

> ### Author Response · Authors · 2022-08-02
> **Official Response to Reviewer PTq2 (Part 1/2)**
>
> We thank Reviewer PTq2 for their thoughtful review. We address all the comments below in detail:
>
> >**Q1:** Is the failure mode of INLP really a valid failure mode?
>
> **A1:** There seems to be a misunderstanding in the experimental setup for the null-space removal experiment from Section 4.2. We construct two separate datasets. The first is a subsampled dataset for training a “clean/fair” main-task classifier that contains only a single value of the spurious concept label. This is not available to the INLP method, thus simulating a real-world scenario where only the main-task classifier may be provided. The second dataset is the full dataset containing all values of the spurious concept label that is provided to the INLP method as input. Specifically, we have,
>
> 1.  **a subsampled dataset** with the same value of spurious concept label to train a “clean” main-task classifier. The main-task classifier is trained on this dataset. Since the subsampled dataset contains only a single value of the spurious concept label the main-task classifier will not use the spurious feature (Ravichander et. al. [30]). Next, this clean main-task classifier is given to INLP (wherein its latent representation is frozen).
>
> 2.  **the full dataset** which is used by INLP for the removal of spurious features from the main-task classifier. This dataset contains both values of the binary spurious concept label in the same proportion, but with different levels of predictive correlation with the main-task label. This setup is exactly the same as proposed in the INLP paper (Ravfogel et. al. [27]).
>
> Therefore, **we would like to point out a misunderstanding:  the probing classifier is not trained on the subset of the dataset which has only one spurious label.**
> Our goal with this experimental setup is to show that when an already “clean/fair” main-task classifier is given as input to INLP for spurious feature removal, it could corrupt it and make it “unclean/unfair” (full details of the setup are in Appendix F.6). Specifically, we vary the predictive correlation ($\kappa$) between the main task and spurious concept labels (shown by different colored lines in Figure 2) and evaluate the performance of INLP using different metrics. We observe that for all values of predictive correlation, the accuracy of INLP goes to random guess as predicted by statement 2 of Theorem 3.2. Hence, **the drop in accuracy of the main-task classifier is not because the probing classifier is being trained on a data subset with the same label, but because the main-task label is correlated with the spurious concept which gets removed with repeated application of INLP**, thus demonstrating the failure mode of INLP.
>
>
> >**Q2:** Is the failure mode of the adversarial removal methods really a valid failure mode?
>
> **A2:** $\kappa=1$ is not the focus of this work. Instead, our results characterize the failure of adversarial removal methods for a range of realistic predictive-correlation $\kappa$ values. Therefore, we clarify that your comment does not apply to our key results.
>
> More specifically, while we agree with your counter-example that for $\kappa=1.0$, it is impossible to distinguish between the main-task label and spurious concept label, we are not concluding the failure of the adversarial removal method based on $\kappa=1$. Rather, as the x-axis of Figure 3 shows, we consider values of $\kappa$ from set $\textbraceleft  0.6, 0.7, 0.8, 0.9, 0.99 \textbraceright$  and show that the adversarial removal method is unable to remove the spurious feature in all cases,  even though for any $\kappa<1.0$, there does exist information to distinguish between the main-task feature and spurious feature.
>
> Note that by introducing predictive-correlation, we are not trying to create a “tricky design” or a  harder setting for the removal method: correlation between the main-task and spurious features is almost always observed in real-world datasets, especially in fairness-related scenarios across demographic groups for which the removal methods have been proposed. E.g., consider the correlation between spurious feature (gender) and the main-task label (profession) as reported in [Webster et al., A] (Figure 1), and between race and labeled positive sentiment in the Twitter-AAE dataset (Appendix Section F.1).
>
> Therefore, failure of the adversarial removal method on our datasets suggests its failure on real-world datasets too (which exhibit predictive-correlation between 0.5 to 1).
>
> \
> \
> **References**
>
> [A] Webster, Kellie, et al. "Measuring and reducing gendered correlations in pre-trained models." arXiv preprint arXiv:2010.06032 (2020).

---

### Official Review · Reviewer_PP9U · 2022-07-11

**Rating:** 8
**Confidence:** 2
**Soundness:** 4 excellent
**Presentation:** 3 good
**Contribution:** 3 good

**Summary:**

This paper discusses the problem of spurious correlations in natural language processing (NLP). The authors discuss the well-known issue that machine learning-based NLP models tend to learn representations that encode spurious correlations between input feature attributes (the authors refer to these as “concepts”) and class labels, thereby causing models to fail to generalize to out-of-distribution data. Specifically, this paper investigates the limitations of two classes of methods proposed to remove spurious correlations, namely null space projection and adversarial training. The authors provide theoretical and empirical evidence that the aforementioned methods fail to alleviate the problem of spurious correlations in NLP by showing that they cannot successfully remove undesirable concepts. For the latter, empirical experiments are conducted on a synthetic as well as three real-world datasets: MultiNLI, Twitter-PAN16 and Twitter-AAE.

**Questions:**

* For the “Ablations” paragraph in Section 4: would it be possible to briefly elaborate on the results with other classifiers and model design choices in the manuscript (not only in the Appendix)? I appreciate that the experiments have been conducted under various model types and configurations, yet it would be beneficial to the reader if the main findings for this could be mentioned in the manuscript.

**Limitations:**

The authors do not explicitly mention the limitations of their work in the paper (a discussion beyond the results is missing) and the paper would benefit from a few sentences on that topic as well as future work based on the main findings.

**Strengths And Weaknesses:**

Strengths:
This paper addresses the highly relevant problem of spurious correlations in NLP and shows that existing methods to tackle that problem are limited at doing so. The paper is well-written, the theoretical analysis and the experimental efforts for empirical validation are well-documented and easy to follow, and the obtained results for the latter are convincing. Furthermore, the paper’s main objectives and findings are well differentiated from the existing literature (as stated in Section 5).

Weaknesses:
My main concern with this work is a lack of discussion on potential ways forward. The paper successfully shows the limitations of existing methods for the removal of spurious correlations. However, the authors do not extensively discuss the implications of their findings and show little effort in discussing potential ways forward and/or alternative approaches to improve existing methods. The manuscript contains various spelling mistakes and missing words (e.g., lines 182, 190, 201, 280, 342), which I encourage the authors to correct for better readability. Finally, some crucial information (e.g., around the predictive correlation; also see my question below) are moved to the Appendix. This is understandable due to limited space but can disrupt the flow for the reader.

---

> ### Author Response · Authors · 2022-08-02
> **Official Response to Reviewer PP9U**
>
> We thank Reviewer PP9U for their thoughtful and encouraging review. We address all the comments below.
>
> >**Q1:** However, the authors do not extensively discuss the implications of their findings and show little effort in discussing potential ways forward and/or alternative approaches to improve existing methods.
>
> **A1:** Thank you for pointing this out. We have added a discussion on the implications and future directions in Appendix H. For a summary, see the “Reply to all reviewers”.
>
>
>
> >**Q2:** The authors do not explicitly mention the limitations of their work in the paper (a discussion beyond the results is missing)
>
> **A2:** Due to space constraints, we had to move the Broader Impact and Limitation sections to Appendix A and B respectively. Later, we plan to move it back to the main paper in the camera-ready submission (utilizing the extra page). We have now summarized the limitations in the “Reply to all reviewers”.
>
>
>
> >**Q3:** Moving finer details like “predictive-correlation” and “Ablation” results to the main paper
>
> **A3:** Thanks for this suggestion. We will do our best to move them back to the main paper in the camera-ready version if accepted.

---

> > ### Comment · Reviewer_PP9U · 2022-08-08
> > **Thank you for the detailed response**
> >
> > Thank you for answering my questions and further elaborating on the limitations of your work.

---

### Official Review · Reviewer_og88 · 2022-07-12

**Rating:** 5
**Confidence:** 3
**Soundness:** 3 good
**Presentation:** 3 good
**Contribution:** 3 good

**Summary:**

This paper analyzes the potential issues with previously proposed methods for sensitive concept removal from text representations. The authors mainly study two branches of methods, null-space removal and adversarial removal, which both use probing classifiers operated on frozen representations. The authors present theoretical analyses to demonstrate that both types of methods cannot remove the sensitive attributes completely and may distort main-task features that hurt the target task performance. The empirical studies are conducted on a set of synthetic and real datasets to support the claims.

**Questions:**

I would like to see some insights/ideas derived from the analyses regarding how the previous methods may be improved. Usually, good theoretical analyses not only reveal the challenges/limitations, but also shed lights on the potential directions to overcome them.

**Limitations:**

I hope the authors would discuss how the assumptions made could be relaxed (e.g., the spurious features may not be assumed to be removed entirely, but to some extent).

**Strengths And Weaknesses:**

Pros:
* The problem studied is interesting and important -- how to effectively remove sensitive features from text representations is of great significance in mitigating ethical concerns about machine learning models.
* The paper presents both theoretical and empirical analyses to support the claims.

Cons:
* The major contribution of the paper does not appear to be that significant to me. While it's important to know the potential issues with previous concept removal methods, the main arguments in this paper (that both null-space removal and adversarial removal methods cannot entirely remove the sensitive features) are more or less already mentioned in the original papers proposing those methods. In this regard, the paper is reiterating some previous observations by backing them up with more theoretical analyses (however, these theoretical analyses appear to be based on very strong assumptions, see below).
* Some assumptions made in the paper are too strong to be meaningful in practice. The authors assume the sensitive features need to be removed entirely from the representations (i.e., a perfect separation between main-task features and spurious ones). However, the concept removal methods are usually not expected to be producing a perfectly clean representation in the first place; they are meant to mitigate the influence of the spurious features. Those methods should be considered useful as long as they can (to some extent) remove the sensitive concepts. Therefore, the "perfect removal" assumption appears too strong--I doubt whether there even exists any method that could satisfy such an assumption. If the assumptions are made in a way that the requirements are essentially impossible to satisfy, the subsequent theoretical analyses that show how previous methods may not work will appear much less meaningful.
* Although the authors claim the spuriousness score as one of the contributions, there are no further evaluations on how accurately this metric corresponds to its expected purpose.

**Post-Rebuttal**: The authors have addressed my concerns raised above, so I am increasing my rating.

---

> ### Author Response · Authors · 2022-08-02
> **Official Response to Reviewer og88 (Part 2/2)**
>
> >**Q2:** Some assumptions made in the paper are too strong to be meaningful in practice. The authors assume the sensitive features need to be removed entirely from the representations (i.e., a perfect separation between main-task features and spurious ones). However, the concept removal methods are usually not expected to be producing a perfectly clean representation in the first place; they are meant to mitigate the influence of spurious features.
>
> **A2:** Our motivation was to simulate a setup that is the most favorable for a probing classifier-based removal method (see L120 of revised version). That is why we assumed that the main-task and spurious features are disentangled in latent representation space (see Assumption 3.1) and individually fully predictive of the main task and the probing task respectively (see Assumption 3.2 and C.1). Under such favorable and simple assumptions, any removal method is expected to completely remove the spurious features  (see L120 of revised version). Thus, our assumptions are ideal *by design* and we do not expect them to be true in practice. If we can show that probing-based methods do not work (cannot remove the spurious concept) even under the case where the spurious and task-relevant features are easily separable in representation space, then it is less likely that they will work under the harder, more realistic settings.
>
> That said, we do agree that in practice, partial removal of spurious features is acceptable. In the revised submission, we have edited the text in Section 2, L81 to reflect this. However, as we state in our response above, **it is still unacceptable to remove or corrupt the main-task features in the process of removing the spurious feature. This is the key criterion** that we apply in this paper and we find that both removal methods end up removing or corrupting main task-relevant features even from a “clean” classifier that does not use spurious features (see Theorem 3.2 and Figures 2, 7, and 10).
>
> >**Q3:** Although the authors claim the spuriousness score as one of the contributions, there are no further evaluations on how accurately this metric corresponds to its expected purpose.
>
> **A3:** Given both main-task and spurious feature labels are binary, our spuriousness score gives us a way to measure the reliance of a given classifier on the spurious features. The motivation for using “Minority-Group-Accuracy” i.e accuracy of the classifier on a subset of data where spurious correlation breaks as a metric for quantifying the degree of spuriousness comes from Sagawa et. al. [33]. This metric is widely used in the Domain-Generalization community to measure the spuriousness of a classifier (Sagawa et. al. [33] and Wang et. al. [A] ). We adapted it by normalizing it with the expected minority group accuracy under a “clean/fair” classifier.
>
> Based on your suggestion, we have added Appendix I to the paper where we compare the degree of spuriousness with the ideal spuriousness metric “$\Delta$Probability” (defined in L323 of the revised version and Appendix F.8) which directly implements Def. 2.1 and thus can be considered as “ground-truth”. “$\Delta$Probability” measures the reliance on a spurious feature by changing the spurious feature in the input space and measuring the change in the prediction probability of the given classifier. Since changing the spurious feature is difficult in the input space we only conduct this evaluation on the synthetic dataset (described in Appendix F.1 in the paper). We observe a strong Pearson correlation (>0.83) between Spuriousness Score and $\Delta$Probability in multiple experiments with p-value<0.05  assuming the null hypothesis that the two metrics are uncorrelated.  (see Table 1 in Appendix).
>
> >**Q4:** I would like to see some insights/ideas derived from the analyses regarding how the previous methods may be improved.
>
> **A4:** Thank you for this suggestion! We have added a discussion on the implications for current methods and future directions in Appendix H. For a summary, see the “Reply to all reviewers”.
>
>
> \
> \
> **References**
>
> [A] Zhao Wang and Aron Culotta; Robustness to Spurious Correlations in Text Classification via Automatically Generated Counterfactuals; Advancement of Artificial Intelligence, 2021

---

> > ### Comment · Reviewer_og88 · 2022-08-08
> > **Response to Authors**
> >
> > Thanks for the response and updates! I like the added discussions in Appendices I and H. I also like the clarification provided in **A1** which makes the contribution of the paper clearer -- please consider incorporating these clarifications in the next version of the paper. Overall, I feel that the authors have provided satisfactory explanations and updates that address my concerns. Hence, I am increasing my rating.

---

> ### Author Response · Authors · 2022-08-02
> **Official Response to Reviewer og88 (Part 1/2)**
>
> We appreciate Reviewer og88 for their thoughtful comments. Below we address some of the concerns.
>
> >**Q1:** The major contribution of the paper does not appear to be that significant to me. While it's important to know the potential issues with previous concept removal methods, the main arguments in this paper (that both null-space removal and adversarial removal methods cannot entirely remove the sensitive features) are more or less already mentioned in the original papers proposing those methods. In this regard, the paper is reiterating some previous observations by backing them up with more theoretical analyses
>
> **A1:** The main argument of our paper is not limited to stating that null-space removal or adversarial methods fail to entirely remove the spurious feature. Instead, **we prove something stronger: even under the most favorable setting where main-task features and spurious features are disentangled in representation space (see Assumption 3.1) and individually fully predictive of the main-task and the probing task respectively (see Assumption 3.2 and C.1), these methods fail to remove the spurious feature**. Further, we show that methods based on null-space removal have **harmful side effects that can lead to corruption (see Statement 1 in Theorem 3.2) or complete removal of the main-task feature along with the sensitive feature (see Statement 2 in Theorem 3.2)**, which is novel to the best of our knowledge. In other words, we show that the method violates a basic principle that any removal method should follow: when the main-task feature is simply correlated with the sensitive feature, it is not justifiable to remove the main-task feature. In addition, our empirical results go beyond simply showing that these methods do not fully remove the spurious feature. **For both null-space and adversarial removal methods, we show that they can worsen the model’s use of spurious features**: even if we start with a “clean/fair” classifier that does not use spurious features, probing-based removal methods will lead to a final classifier that does use those spurious features (for details, see Figures 2, 7, and 10).
>
> With respect to claims already known, for Null-Space Removal (INLP), the authors of the paper (Ravfogel et. al. [27]) have shown that with an increasing number of INLP iterations they observe a drop in performance of the main-task classifier. But they don't provide any explanation for this phenomenon. In Theorem 3.2, we show that this drop in performance could be because of two reasons: a) Unintended mixing/corruption of main-task and spurious features (Statement 1 of Theorem 3.2), and b) Unintended removal of the main-task features (Statement 2 of Theorem 3.2). The authors propose early-stopping criteria based on the tradeoff between the drop in main-task accuracy and an increase in spurious feature removal. But our experiments show that finding such a tradeoff is difficult in practice. E.g., we show that when INLP has been given a “clean/fair” main task classifier for removal,  early stopping could corrupt the initially “clean” classifier by making it use the spurious features (see “Early stopping increases reliance on spurious features” paragraph in Section 4.2); while more iterations will lead to removal of all task-relevant features (see Section 4.2).
>
> In the adversarial removal paper (Elazar et. al. [11]), the authors empirically showed that the method will fail to remove the sensitive concept fully under a very specific setting when the attacker has access to a balanced dataset (i.e., no correlation between the main task and spurious features, $\kappa=0.5$). In our empirical work, we extend these results by showing the failure with varying degrees of predictive-correlation between main-task and spurious features ($\kappa \in [0.5,1]$). We observe that as the predictive correlation increases, adversarial removal performs worse (see Section 4.3). In addition, we show a more severe failure: when adversarial removal methods are initialized with a “clean” main-task classifier they end up making it “unclean” (see Figure 10 and “Adversarial Removal makes a classifier unfair” paragraph in Appendix Section G).

---

### Official Review · Reviewer_WqdQ · 2022-07-13

**Rating:** 8
**Confidence:** 4
**Soundness:** 4 excellent
**Presentation:** 4 excellent
**Contribution:** 4 excellent

**Summary:**

This paper presents a theoretical analysis of the methods of subspace removal using probing classifiers. The main argument is that even the features are perfectly separable, the probing classifier will learn something about the task-related features, in addition to the concept related features. Then, based on this argument, this paper claims that the existing probing-classifier-based methods are not sufficient to remove concept-related feature/information.

**Questions:**

No further question

**Ethics Review Area:**

["I don’t know"]

**Limitations:**

No limitation

**Strengths And Weaknesses:**

**Strengths**

Overall, I think this paper is solid in both theoretical analysis and empirical study.

First of all, with some mild assumptions, this work presents a theoretical analysis of why probing-classifier-based method is not reliable in removing concept-related information. To some extent, the assumption is too ideal; for example, it assumes disentangled representation, which I am not sure it is true for most cases. However, this ideal case demonstrates the concept.

In addition to the theoretical analysis, it also presents some empirical studies, which further strengthens this work. Compared to the existing works that only demonstrate the algorithm with empirical results, I like the combination of theory and empirical results.

---

> ### Author Response · Authors · 2022-08-02
> **Official Response to Reviewer WqdQ**
>
> We thank reviewer WqdQ for their positive comments.
>
>
> > **Q1:** To some extent, the assumption is too ideal; for example, it assumes disentangled representation, which I am not sure it is true for most cases. However, this ideal case demonstrates the concept.
>
> **A1:** Our motivation was to simulate a setup that is the most favorable for a probing classifier-based removal method (see L120 of revised version). That is why we made the disentangled representation assumption. It is ideal *by design* and not expected to be true in practice. If we can show that probing-based methods do not work (cannot separate the spurious concept) even under the case where the spurious and task-relevant features are easily separable in representation space, then it is less likely that they will work under the harder, more realistic settings.
>
> Our results show that even under these favorable, simplifying assumptions, probing-based removal methods do not work well: they either remove the correlated main-task feature along with the sensitive feature, fail to remove the sensitive feature completely, or both (see Theorem 3.2 and 3.3).

---

> > ### Comment · Reviewer_WqdQ · 2022-08-08
> > **Thanks for the further clarification**
> >
> > as well as the revision.

---

### Author Response · Authors · 2022-08-02
**Reply to all Reviewers**

We thank all the reviewers for appreciating the importance of our contribution on removing spurious correlation in NLP models, both through theoretical and empirical results. Below we address some common issues. We have also uploaded a revised paper with changes marked in blue.


>**Discussion on implications for current methods and possible remedies** (*og88*, *PP9U*)

Our work provides 1) metrics for evaluating and debugging probing-based removal methods; and 2) a  criterion for removal methods that motivate new directions beyond probing. We have added a new section (Appendix H) discussing these and will add it to main paper for camera-ready (using extra page allowed). Below we provide a condensed version.

Based on our results, before using a probing-based removal method, we recommend that practitioners validate it using the metrics we presented. Specifically, we provided two useful tests. First, we provided a sanity-check test: **any reasonable removal method should not modify an already “clean/fair” classifier that doesn't use any spurious features to produce a final classifier that does use those features.** To validate this, we adapt a method from Ravichander et. al. [30] for generating a “clean/fair” classifier by restricting the training set only for initial training of the main task classifier; and provide two metrics: Spuriousness Score (defined in Section 3.4) and $\Delta$Probability (defined in L323 of revised version and Appendix F.8) to measure spurious features captured by the model (see Figures 7 and 11).  Second, the proposed spuriousness score can also be used to detect and debug any spuriousness in the probing classifier (see Figure 5 and 6). As a future direction, we encourage the community to develop more such benchmarks and sanity-check tests to evaluate proposed removal methods.

Overall, our theoretical and experimental evaluations indicate that it is difficult to create a probing-based removal method due to the fundamental limitation of learning a “clean” probing classifier (see Lemma 3.1, Theorem 3.2 and 3.3). While we acknowledge that full removal of a spurious concept may not be possible, **we specified a simpler selection criteria on which probing-based removal methods failed: a spurious concept removal method should not remove the task-relevant features which are correlated with the spurious concept (and not caused by it)**(see updates in revised Sec.2, L81). Given the sensitive nature of many feature-removal tasks (e.g., fairness on demographic groups) and risk of counterproductive harms, we therefore question whether probing-based methods are right way forward.
Instead, we point attention to other approaches that may provide better guarantees. An example is counterfactual data augmentation [Balashankar et al, A] where given an input, we obtain a new input that only changes spurious concept and then regularize to have such input pairs have the same representation (thus directly implementing Def. 2.1 from our paper).  That said, a limitation is that it may require manual effort and removal quality will depend on diversity of the counterfactual examples generated. An alternative direction is to take inspiration from algorithmic fairness literature [Mehrabi et al., B] and focus on the predictions of the classifier rather than the representation. Compared to removal in latent space, enforcing certain fairness properties on model predictions is a more well-formed task,  more interpretable, and definitely more relevant if the final goal is fair decision-making.


>**Discussion on limitations of our work** (*og88*,*PP9U*)

We had provided limitations of our work in Appendix B. We plan to move it back to main paper in the camera-ready version (utilizing extra page). We summarize the limitations below
1. *Assuming frozen or non-trainable latent representation in theoretical results*: We partially address this limitation in our empirical work where we do not make such assumptions.
2. *A general result for any probing-based classifier*: Our work addresses failure modes of probing-based removal on two popular methods, null-space removal (Sec 3.2 and 4.2), and adversarial removal (Sec 3.3 and 4.3). It would be interesting to show that any removal method based on a probing classifier will fail.
3. *Spuriousness score is an approximation*: We proposed spuriousness score metric because the ideal metric, $\Delta$Prob (defined in L323 of revised version and Appendix F.8), requires human labeling for real-world data. While we validated spuriousness score on synthetic datasets (Appendix I), it will be useful to collect augmented data using human experiments on real-world datasets to directly measure $\Delta$Prob and validate Spuriousness score metric.

\
\
**References**

[A] Balashankar et al. Can We Improve Model Robustness through Secondary Attribute Counterfactuals? Proc. EMNLP, 2021

[B] Mehrabi et al. A Survey of Bias and Fairness in Machine Learning ACM Comp. Surv. Journal, 2021

---

### Author Response · Authors · 2022-08-09
**Thanks!**

We thank all the reviewers for engaging with our rebuttal. We believe we have answered all the concerns. Please let us know if any comments or question remains.

---

### Meta-Review · Area_Chair_mbvX · 2022-08-25

**Recommendation:** Accept
**Confidence:** Certain

**Metareview:**

This papers analyzes failure modes of methods that aim to remove spurious features from the representation. The key finding is that since the spurious features are correlated with the core features, such methods will inevitably also remove core features during the process, thus hurting performance. Both the theoretical results and the empirical findings are important for understanding concept-removal methods which are widely used in domain adaptation and robust learning. All reviewers agree that the contribution is significant. The authors may strengthen the paper by discussion ways forward.

**Award:**

No

---

### Decision · Program_Chairs · 2022-09-14

Accept